# Properties of surface water masses in the Laptev and the East-Siberian Seas in summer 2018 from in situ and satellite data

Anastasiia Tarasenko[1, 2], Alexandre Supply[3], Nikita Kusse-Tiuz[1], Vladimir Ivanov[1], Mikhail Makhotin[1], Jean Tournadre[2], Bertrand Chapron[2], Jacqueline Boutin[3], Nicolas Kolodziejczyk[2], and Gilles Reverdin[3]

[1]Arctic and Antarctic Research Institute, Saint-Petersburg, Russia
[2]Univ. Brest, CNRS, IRD, Ifremer, Laboratoire d'Océanographie Physique et Spatiale (LOPS), IUEM, Brest 29280, France
[3]Sorbonne Université, CNRS, IRD, MNHN, Laboratoire d'Océanographie et du Climat, Expériments et Approches Numériques (LOCEAN), Paris, France

**Correspondence:** Anastasiia Tarasenko (tad.ocean@gmail.com)

**Abstract.** Variability of surface water masses of the Laptev and the East-Siberian Seas in August-September 2018 is studied using *in situ* and satellite data. *In situ* data were collected during the ARKTIKA-2018 expedition and then complemented with satellite-derived sea surface temperature (SST), salinity (SSS), sea surface height, wind speed, and sea ice concentration. The estimation of SSS fields is challenging in high latitude regions, and the precision of Soil Moisture and Ocean Salinity (SMOS) SSS retrieval is improved by applying a threshold on SSS weekly error. For the first time in this region, the validity of DMI (Danish Meteorological Institute) SST and SMOS SSS products is thoroughly studied using ARKTIKA-2018 expedition continuous thermosalinograph measurements and CTD casts. They are found adequate to describe large surface gradients in this region. Surface gradients and mixing of the river and the sea water in the ice-free and ice-covered areas are described with a special attention to the marginal ice zone at a synoptic scale. We suggest that the freshwater is pushed northward, close to MIZ and under the sea ice, which is confirmed by the oxygen isotopes analysis. The SST-SSS diagram based on satellite estimates show a possibility to investigate the surface water masses transformation at a synoptic scale and reveal a presence of the river water on the shelf of the East-Siberian Sea. The Ekman transport is calculated to better understand the pathway of surface water displacement on the shelf and beyond.

## 1 Introduction

The eastern part of the Eurasian Arctic remains one of the less studied areas of the Arctic Ocean. Carmack et al. (2016) described this region as an "interior shelf" (the Kara, the Laptev, the East-Siberian, together with the Beaufort seas), where 80% of the Arctic basin river discharge is released. Armitage et al. (2016) estimated the annual river water input as 2000 $m^3$. The Arctic Ocean stores 11% of a global river discharge, so its role in a planetary water budget deserves a special attention. The surface stratification and the freshwater content are regarded as key parameters that have to be followed to better understand the changing state of a "New Arctic" climate (Carmack et al. (2016)). Johnson and Polyakov (2001) discussed the salinification of the Laptev sea since 1989 up to 1997, explaining it by the eastward freshwater displacement and an excessive brine release in the sea ice leads. A more recent study reports that a 20% increase in the Eurasian river runoff is observed over the last 40 years

(Charette et al. (2020)). Overall, a freshening of the American basin of the Arctic Ocean was reported in 2000-2010 (Carmack et al. (2016)), and at the same time, a decrease in a freshwater content of about 180 $km^3$ between 2003 and 2014 was calculated from altimetry measurements by Armitage et al. (2016) over the Siberian shelf. The importance of shelf seas for the freshwater content storage and distribution was outlined in several recent studies (Haine et al. (2015), Armitage et al. (2016), Carmack

et al. (2016)). The importance of the exchange between the shelf seas and the deep basin is large: 500 $km^3$ for the Laptev and the East-Siberian Seas with anticyclonic atmospheric vorticity "on quasi-decadal timescales", calculated from 1920-2005 hydrographic measurements by Dmitrenko et al. (2008). Previously, the Arctic shelf was considered as a "short-term buffer" (3.5±2 years) storing the river water before it enters into the deeper central part and is transported by the Transpolar drift (9-20 years) to the North Atlantic through the Fram Strait (Schlosser et al. (1994)). A recent study of Charette et al. (2020) shows that

the "intensification of the hydrologic cycle" will speed up the transport of the freshwater, carbon, nutrients and trace elements from the shelf to the central Arctic and further: the trace elements and isotopes move from the shelf edge to the Transpolar drift stream over 3-18 months, and the Transpolar drift takes 1-3 years.

Processes taking place on the eastern Eurasian Arctic shelf are important for the redistribution of the freshwater arriving there and its further path, while the amount of freshwater is expected to increase (Carmack et al. (2016), Charette et al. (2020)).

A complex topography, several sources of fresh and saline water masses, unstable atmospheric conditions and ocean processes, such as mesoscale activity and tidal currents, can alter the direction of the freshwater distribution. Close to the coast the riverine water from several sources is expected to propagate eastward as a "narrow (1-20 km) and shallow (10-20 m) feature" (Carmack et al. (2016), Lentz (2004)), but its transformation and mixing with a saline seawater and sea ice melting and freezing are less studied. The Laptev and the East-Siberian shelf areas were described as a substantial region of sea ice production for the

central Arctic (Ricker et al. (2017)), and to better estimate the impact of the incoming freshwater on the sea ice formation, the freshwater pathways in the Arctic should be better understood. Despite several studies on the freshwater in the Eastern Arctic (e.g., Semiletov et al. (2005), Dmitrenko et al. (2012), Osadchiev et al. (2017), Bauch and Cherniavskaia (2018)), to the best of our knowledge, no study has shown yet the evolution of the water masses on a synoptic scale in the Laptev Sea, exept the very recent one by Osadchiev et al (2020), which has been done in parallel with this study, but on the basis of other *in situ* data.

In this paper, we look at the information accessible with satellite salinity. The salinity provides precious information about the fate of the freshwater river input. While this information is restricted to the top sea surface, the regular and synoptic monitoring of sea surface salinity from space allows to document its spatio-temporal variability in great details not accessible with any other means, providing a new tool for analyzing some of the processes at play.

The Laptev Sea is shallow in its southern and central parts (less than 100 m) with a very deep opening in the north (3000 m)

(Fig. 1). Several water masses are mixed in the Laptev Sea. The Lena, Khatanga, Anabar, Olenyok, and Yana rivers discharge fresh water in the shallowest part of the Laptev Sea in the south. The Kara Sea water enters via the Vilkitskiy and the Shokalskiy straits, the Atlantic Water (AW) propagates along the continental slope to the north of the Severnaya Zemlya Archipelago and further eastward, the Arctic Water is found in its northern part (Rudels et al. (2004), Janout et al. (2017), Pnyushkov et al. (2015)). A direction of the surface freshwater circulation is supposed to correspond to a general displacement of the

intermediate Atlantic Water: mainly eastward following the coastline (Carmack et al. (2016)). This eastward transport brings

the water masses of the Laptev Sea over the shelf of the East-Siberian Sea where they meet Pacific-origin waters (Lenn et al. (2009), Semiletov et al. (2005)).

In the Arctic region, a strong seasonality of air-sea heat flux, sea ice melting and freezing modify the temperature and the salinity in the upper layer, and therefore, result in a vertical structure of the water column with fronts at the surface and "modified layers" in the interior (Rudels et al. (2004), Pfirman et al. (1994), Timmermans et al. (2012)). The most common concept of the upper ocean layer is a "mixed layer" concept: between an ocean surface being in contact with the atmosphere and a certain depth, the temperature and the salinity are homogeneous. A mixed layer extends until a specified vertical gradient in density and/or temperature (de Boyer Montégut et al. (2004), Timokhov and Chernyavskaya (2009)), or a maximum of Brunt-Väisälä frequency (Vivier et al. (2016)). In the Arctic, the reported mixed layer depth (MLD) varies between 5 and 50 m depending on region, time, and open water or under the ice measurements (10 m in the Laptev and the East-Siberian seas and 5 m in the Central Arctic ocean and Northern Barents Sea in summer, Timokhov and Chernyavskaya (2009); 10-15 m in the Beaufort Sea close to MIZ in summer, Castro et al. (2017); 20 m in the Barents Sea in late summer, Pfirman et al. (1994), 40-50 m under the ice close to the North Pole in winter, Vivier et al. (2016)).

At the same time, Timmermans et al. (2012) proposed to use a term "surface layer" instead of "mixed layer" for the Arctic Ocean, because a water layer lying between the sea surface and the Arctic main halocline can be weakly stratified even though the halocline hampers an active exchange of matter and energy. The main halocline is situated at 50-100 m depth in the Eastern Arctic (Dmitrenko et al. (2012)), and at 100-200 m depth in the Western Arctic Ocean (Timmermans et al. (2012)). Using concept of the "surface layer", the processes in that layer can be discussed separately from the ones in the deeper layer. The freshwater is expected to be delivered to the central (European) Arctic from the Siberian shelf, roughly along the Lomonosov Ridge and to the western Arctic, partly along the continental slope (Charette et al. (2020)).

The position of the pycnocline in the Arctic is mostly defined by salinity. One of the first studies of Aagaard and Carmack (1989) devoted to the freshwater content was using 34.80 as a reference salinity value separating the "fresh" and the "saline" water; 34.80 was considered a mean Arctic Ocean salinity at that time. This value is used as well in more recent overviews (e.g. Haine et al. (2015), Carmack et al. (2016)) and helps to define the "Atlantic Water" as saltier than this value. Rabe et al. (2011) used a 34-isohaline depth to estimate a liquid freshwater content in the Arctic Ocean. Carmack et al. (2016) considered a depth of a "near-freezing freshwater mixed layer" in the Eurasian Arctic Ocean to be 5-10 m. Cherniavskaia et al. (2018) reported an overall salinity in the upper 5-50 m layer within the range from 30.8 to 33 based on *in situ* data in the Laptev Sea during 1950-1993 and 2007-2012. Between the very surface layer and the Atlantic Water, Dmitrenko et al. (2012) found the Modified "Lower Halocline" Water with typical characteristics of salinity (between 33 and 34.2) and a negative temperature (below -1.5°C); in 2002-2009 this layer was situated at 50-110 m depth. The study of Polyakov et al. (2008) on the Arctic Ocean freshening defined the upper ocean layer to be between 0 m and a depth of a density layer $\sigma_\theta = 27.35 \, kg \cdot m^{-3}$. This isopycnal is often located at 140–150 m depth, "slightly above the Atlantic Water upper boundary defined by the 0°C isotherm".

The stable vertical stratification is modified by mixing. Mixing can be induced by winds generating surface-intensified Ekman currents, mesoscale dynamics (eddies), a shear in tidal and other currents (Carmack et al. (2016), Lenn et al. (2009), Rippeth et al. (2015)). Tidal currents and internal waves amplified over the shelf edge are associated with the mixing in the

interior of the water column, below or in the main Arctic pycnocline (Rippeth et al. (2015), Lenn et al. (2009), Lenn et al. (2011)).

Temperature and salinity fronts separate well-mixed water masses. Dmitrenko et al. (2005) and Bauch and Cherniavskaia (2018) showed that interannual changes of a river discharge and wind patterns define the position of oceanographic fronts in the central part of the Laptev Sea. Based on model results, Johnson and Polyakov (2001) showed that in 1989-1997 the freshwater was driven eastward under the influence of winds associated with a "strong cyclonic vorticity over the Arctic". The same study demonstrated that the associated salinification of the central Arctic Ocean weakened the vertical stratification of the water column. The anticyclonic regime is considered to increase the salinity of the shelf seas (Armitage et al. (2016)). Armitage et al. (2017) discuss the importance of sea ice, as it creates a surface drag and establish the Ekman transport of the freshwater in the surface layer, which, in turn, impacts the dynamical ocean topography and geostrophic currents in the Arctic Ocean. Armitage et al. (2018) further mention that alongshore winds correlated with AO (Arctic Oscillation) index create the onshore Ekman transport, changing water properties over the shelf.

In the seasonal cycle, the summer season is of a particular interest for all Arctic studies. The sea ice melting usually starts in June and ends in August-September, while the sea ice formation can start already in September, and by November the Laptev Sea is already completely sea-ice covered. The East-Siberian Sea is usually covered by the sea ice most of the year, and is exposed to the air-sea interaction for a shorter period of time (in August-September) over a smaller ice-free surface than the Laptev Sea. August and September are two summer months that are very important for the heat exchange between the open ocean and the atmosphere over the Laptev Sea. In a recent study, Ivanov et al. (2019) reported that during this time period when the sea ice is melting and the ocean is opening, the net radiative balance at the sea surface changes from 100 $W/m^2$ to zero values, following the seasonal cycle of shortwave radiation (meaning the flux from the atmosphere to the ocean). The sea level anomalies over the Eastern Arctic shallow seas are positive and largest in summer (up to 10 cm at 75°N, down to 3 cm at 80°N, as it was reported by Andersen and Johannessen (2017)). The seasonal peak of the maximum freshwater content over the shelf is found in summertime when the river discharge is the highest, while the freshwater content minimum (following export of this accumulated freshwater) occurs in March, when the freshwater captured by sea ice is advected away from the shelf (Armitage et al. (2016), Ricker et al. (2017)).

The Laptev Sea is not at all sampled by Argo products, so the recent ARKTIKA-2018 expedition measurements combined with novel satellite sea surface salinity and other satellite-derived parameters provide an unprecedented documentation of the temporal evolution of the surface water properties in the Laptev and East-Siberian Seas during the summer 2018. In this study, we propose to follow the upper ocean water displacement and to discuss what causes it on a daily basis.

## 2   Data and Methods

To analyse the upper-ocean processes, we will focus on the surface layer with satellite data and on the upper 250 m layer with the CTD (conductivity, temperature, depth) casts, providing the isohaline and isopycnal positions. Such an approach to the upper layer is required to estimate the upper limit of the Atlantic Water, which is one of the key contributors to the water mass

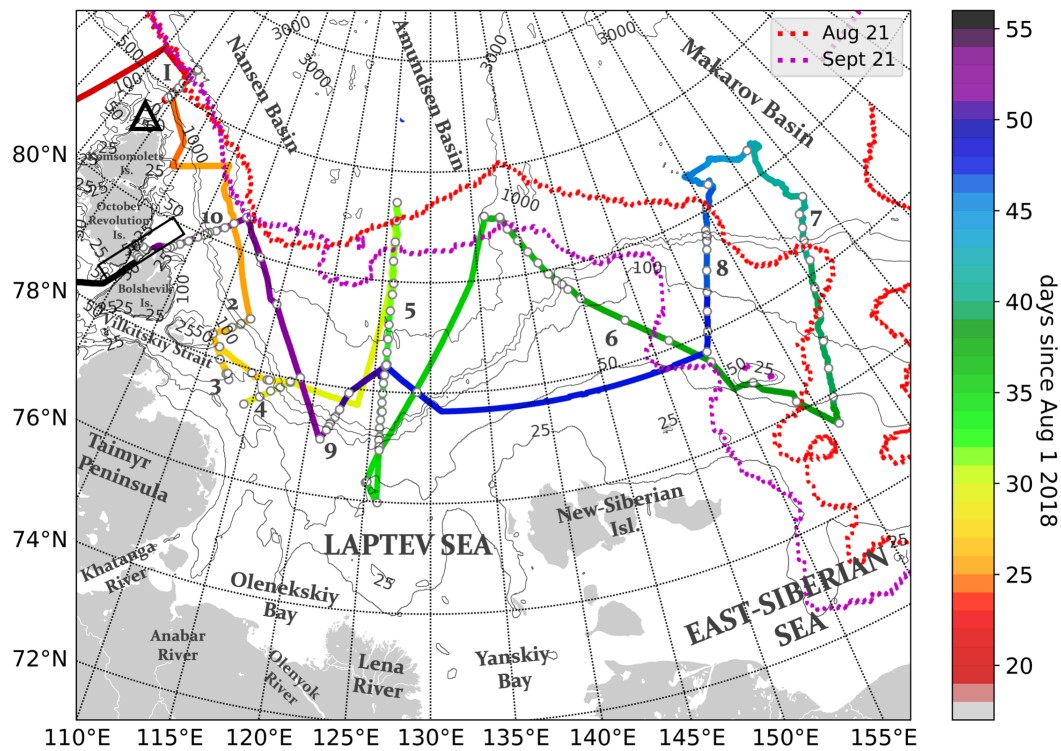

**Figure 1.** Legs and stations of the ARKTIKA-2018 expedition overlaid on the bathymetry from ETOPO1 "1 Arc-Minute Global Relief Model" (Amante and Eakins (2009)). CTD stations are shown with white dots. The colour indicates the number of days since August 1, 2018. The sea ice edge position is indicated with a red dashed line for the beginning (August 21) and with the purple dashed line for the end of the expedition (September 21). The ice edge is based on the sea ice mask provided in the DMI SST product. Numbers indicate positions of 10 oceanographic transects discussed below. The black triangle in the north of the Komsomolets Island shows the Arkticheskiy Cape. The Severnaya Zemlya Archipelago consists mainly of the Komsomolets, the October Revolution, and the Bolshevik Islands (with smaller islands not shown here). The black box indicates the Shokalskiy Strait between the October Revolution and the Bolshevik Islands. The Yana river estuary (outside the map area) is south of Yanskiy Bay

transformation. The surface water evolution of the Laptev and the East-Siberian Seas is described and discussed with respect to changes in wind speed and direction during the ARKTIKA-2018 expedition.

## 2.1 In situ measurements during the ARKTIKA-2018 expedition

Oceanographic measurements during the ARKTIKA-2018 expedition on board RV Akademik Tryoshnikov started on August 5 21, 2018 and ended on September 24, 2018 (Fig.1). Oceanographic sections were organized to take into account the require-ments of different scientific expeditions on board, NABOS (Nansen and Amundsen Basin Observational System) and CATS (Changing Arctic Transpolar System) to observe shallow and continental slope processes. NABOS sections were mostly cross-

shelf (1, 5, 6-8, 10), and CATS sections were shallower (2-4, 9). Sections 3 and 10 were made in the straits between the Kara and the Laptev Sea: section 3 in the Vilkitskiy Strait southward to the Bolshevik Island, with depths from 70 to 200 m opening into the deep central part of the Laptev Sea (more than 1000 m) and section 10 in the narrow and rather shallow (250 m) Shokalskiy Strait between the Bolshevik and the October Revolution Islands. Some measurements were carried out in marginal ice zone (MIZ) and ice-covered area (see the sea ice edge positions at the beginning and the end of the cruise in Fig.1). In this study we define MIZ as an area with 0-30% sea ice concentration close to the ice edge. Standard oceanographic stations (145 in total) were conducted with SeaBird SBE911plus CTD instrument equipped with additional sensors. For this study, we use mainly the CTD measurements of potential temperature and practical salinity, but also the results of oxygen isotope analysis from the first (surface) bottle samples (Alkire and Rember (2019)). All CTD data were processed and quality checked. The cruise data can be found at https://arcticdata.io/catalog/data (Polyakov and Rember (2019)) and Ivanov (2019).

The ship was equipped with an underway measurement system Aqualine Ferrybox, widely known as a thermosalinograph, TSG. The instrument had a temperature and a conductivity (MiniPack CTG, CTD-F) sensors and a CTG UniLux fluorometer installed; thus, continuous temperature, salinity and chlorophyll-a estimations were obtained along the ship's trajectory. The inflow is situated at 6.5 m below the surface (the inflow hole is on the ship's hull). All data were processed and filtered for random noise and bad quality measurements, and then compared and calibrated with CTD measurements. When calculating a linear regression between CTD measurements at 6.5 meters depth and TSG measurements, we obtain a good correlation for both temperature and salinity (correlation coefficient equal to 0.979 and 0.966, respectively, not shown). The standard error is 0.023 for temperature and 0.025 for salinity, and the standard deviation for the difference of measurements (CTD minus TSG) was $STD_{temp} = 0.413°C$, and $STD_{sal} = 0.423$. To adjust the continuous TSG measurements to the more precise CTD measurements, we applied the obtained linear regression equation to TSG data. We only use these adjusted temperature and salinity data.

The vertical profiles of the conservative temperature and practical salinity in the upper layer are presented in Fig.2. To investigate if the TSG measurements can be used to study the surface layer in a highly stratified Laptev sea, we calculated a summer mixed layer depth following de Boyer Montégut et al. (2004) method based on density and temperature gradient thresholds (Fig.2, a, c). The MLD is found at a depth of the first maximum temperature gradient below a depth of defined (by given threshold) density gradient (see de Boyer Montégut et al. (2004) for details). Using the same approach, we computed MLD with density and salinity vertical profiles. The threshold chosen for practical density gradient was 0.3 $kg/m^3$ per 1 m, and 0.2 salinity units per 1 meter for conservative temperature and practical salinity gradients. Regarding the MLD calculated from salinity ($MLD_{sal}$), most of the measured vertical profiles (75.17%) had the $MLD_{sal}$ below 7 m depth with the median of $MLD_{sal}$ 11.99 m. As for the temperature ($MLD_{temp}$), 81.37% of the measured profiles had the MLD below 7 m depth with a median of $MLD_{temp} = 13.50m$. Thus, in most cases the upper 12 m of the surface layer was homogeneous, and the CTD and TSG measurements can be used for the validation of satellite data. The median vertical profiles of temperature and salinity in the upper 5-100 m are presented as well as the associated STD in Fig.2, b, d). We observe rather cold (0.5°C) and fresh (30.5) water at 5 m, followed by a smooth thermo- and halocline down to 30 m depth (with a temperature of -1.3°C and salinity of 33.8). Below 30 m the temperature is slightly rising to -1 °C, and salinity stays close to 34.5. The STD of

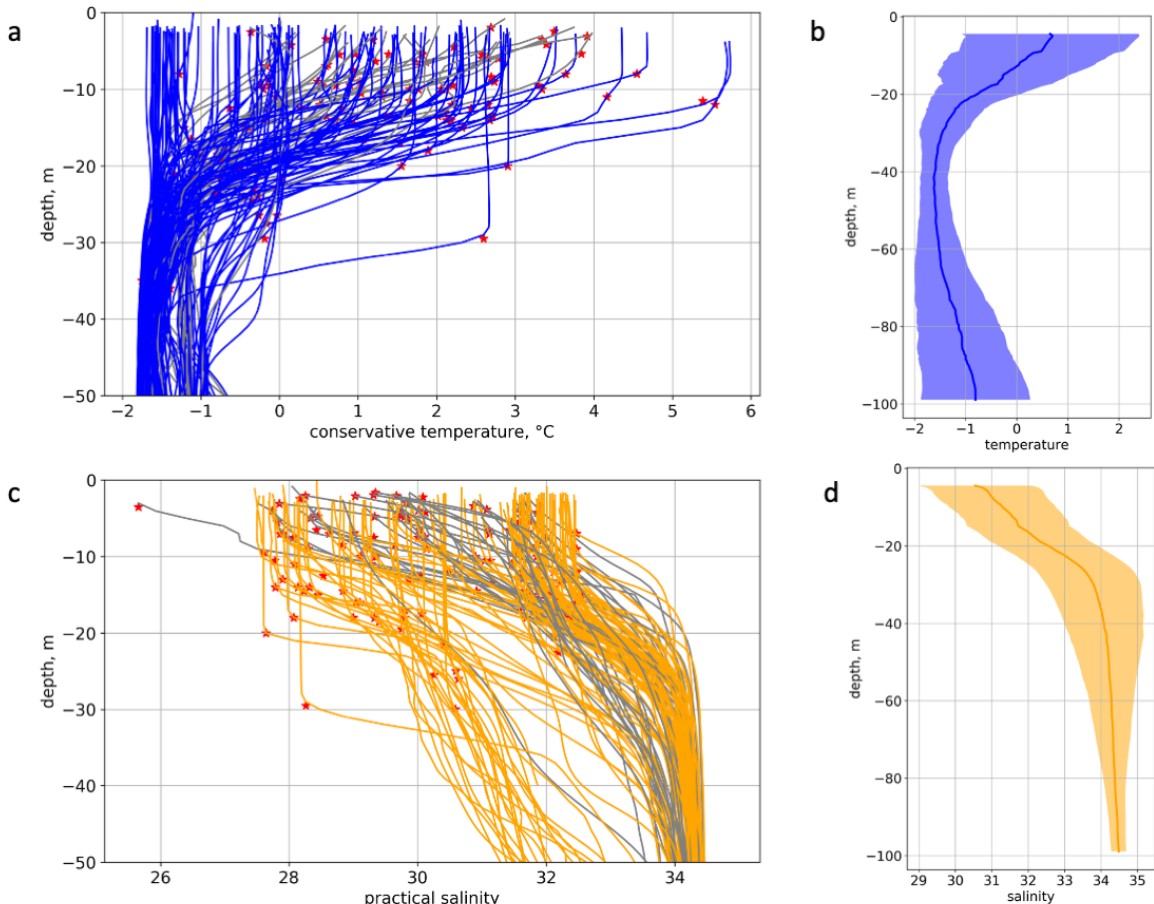

**Figure 2.** Vertical profiles of conservative temperature (a, b) and practical salinity (c, d) from CTD measurements in the upper ocean layer. Figures (a) and (c) show all vertical profiles in the upper 50 m, calculated using de Boyer Montégut et al. (2004) method (see details in the text), red stars indicate the mixed layer depth; coloured profiles show the cases, when the MLD is below 7 m depth and gray profiles indicate when the MLD is above 7 m depth. Figures (b) and (d) show the median vertical profiles in the 5-100 m layer of temperature and salinity, respectively, where the shaded area shows the associated STD.

conservative temperature is the largest at the surface (1.55°C) and smallest at 40 m depth (0.27°C). The STD of salinity is also the largest at the surface, 1.50, but is diminishing with depth to 0.20 at 100 m. Nevertheless, it is clear that at the end of a summer season in the region with very different water origins, these median profiles are not representative for all water masses. Additionally, we did an important number of CTD casts in very shallow areas with depths between 30 and 50 m, so the calculated averaged (median) vertical profile is composed of "shallow" and "deep" vertical profiles. We do not include the very surface measurements above 5 m, because we only had 45 CTD measurements at 2 m depth among 146 possible, and taking them into account would bias the median profiles as well.

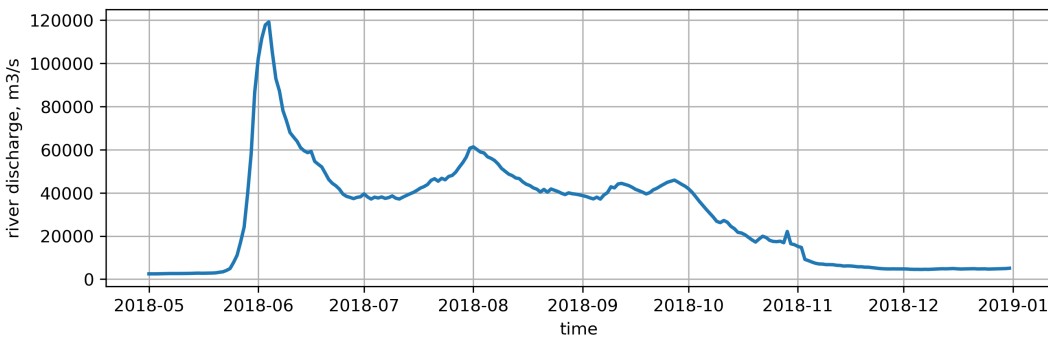

**Figure 3.** The Lena River discharge in 2018, data from Arctic GRO dataset (https://www.arcticrivers.org/data)

### 2.1.1 River discharge

To illustrate the amount and temporal variability of the river discharge in 2018, we used daily measurements of the Lena River discharge from the Arctic Great Rivers Observatory (GRO) dataset (https://www.arcticrivers.org/data)). In Fig.3 we present a time series of the Lena River discharge from May to November 2018. The river stayed under the ice with a very small discharge

up to the end of May. The main peak of the Lena River discharge occurred in the beginning of the Arctic summer in June and corresponds to the snow and ice melting over the river basin in Siberia. In two weeks, the discharge changed from 2 500 to 120 000 $m^3/s$. The second, smaller peak of the river discharge occurred at the beginning of August (60 000 $m^3/s$), which might be associated with the summer precipitation. During August 2018 the river discharge decreased from 60 000 to 40 000 $m^3/s$, and in September it varied very little staying close to 40 000 $m^3/s$. A significant diminution of the river discharge started in

the beginning of October and continued up to the beginning of November. After the beginning of November the river discharge was very weak and close to its minimum values (4500 $m^3/s$).

The described seasonal dynamics is typical for the Lena River and consistent with existing results, e.g. demonstrated in Janout et al. (2015). It can be complemented by the results of Papa et al. (2008) study of the large Siberian rivers using satellite data. Papa et al. (2008) showed that the maximum of precipitation over the basins of the Lena, the Ob' and the Yenisey Rivers

occurs in July, and the mean monthly air temperature is maximum at that time.

### 2.2 Satellite data

Satellite data provide information on the surface distribution of geophysical characteristics over the whole study area together with their temporal evolution.

All products listed below are considered from August 1, 2018 to September 25, 2018 (the last day of ARKTIKA-2018 expe-

dition). For consistency, when not specifically indicated all products are linearly interpolated on a regular grid within the box 74-85N 90-170E, with 0.01-degree step in latitude, and 0.05 degree in longitude. The spatial resolution of the selected grid roughly corresponds to 1 km.

### 2.2.1 Sea surface temperature

The SST-retrieving instruments with the highest resolution, such as AVHRR (Advanced Very High Resolution Radiometer), MODIS (Moderate Resolution Imaging Spectroradiometer) and VIIRS (Visible Infrared Imaging Radiometer Suite) work in Near Infrared (NIR) and Infrared (IR) bands and strongly depend on atmospheric conditions (providing measurements only for clear sky without clouds). For lower resolution microwave instruments, such as AMSR2 (Advanced Microwave Scanning Radiometer 2), the clouds are transparent, but the SST retrievals may still be hampered by high wind speed and precipitation events. As satellite measurements in IR and NIR ranges are sparse because of the frequent cloudiness over the Arctic Ocean, we used a blended product. In this paper we use the Danish Meteorological Institute Arctic Sea and Ice Surface Temperature product (hereafter referred as "DMI SST"). DMI SST is a Level 4 daily product provided by the Copernicus Marine service ("Level 4 product" means that several swath measurements were interpolated to achieve a regular resolution in time and space). Daily surface temperatures over the sea and ice are derived on a 5 km spatial grid from several instruments: AVHRR, VIIRS for SST and AMSR2 for sea ice concentration, using optimal interpolation (Høyer et al. (2014)).

Besides the full coverage over the studied area, the advantage of the blended DMI SST product is that it takes into account the ice temperature, so the marginal ice zone (MIZ) is better represented and not masked out. The total number of SST measurements ingested over the studied area from August 1 to September 25, 2018 varies from 1000 to 2500 measurements per pixel.

### 2.2.2 Validation of DMI SST

The first step of the DMI SST validation was its value-by-value comparison with a collocated *in situ* dataset (nearest neighbour DMI SST pixel). For this analysis, we co-located DMI SST with the *in situ* potential temperature measurements in the upper 6.5 m layer: all available CTD measurements averaged every half a metre above 6.5 m depth and all TSG measurements at 6.5 m depth averaged every 30 minutes. The median depth of the collocated CTD measurements is 5.25 m. As for the TSG, the ship was moving with a median speed of 8 knots during the cruise, so an average of 30-minutes TSG measurements is an average over approximately 7.5 km. Thus, 30-minutes TSG average is comparable with one DMI SST pixel (10 km). There were 1707 collocated points in the analysis.

Although satellite SST estimates may differ from the *in situ* temperature measurements in the upper 6.5 m, we expect an overall consistency between the datasets. Studies carried out by Castro et al. (2017) devoted to the validation of MODIS SST in the MIZ, and by Vivier et al. (2016), which described *in situ* measurements in the iced-covered area, reported that the first 7-10 m layer below the surface was mostly homogeneous. As is shown in Fig.2, most of our measurements (more than 75%) were homogeneous in the upper 12 m (and were done in the ice-free areas). Nevertheless, a diurnal warming and a local vertical mixing can affect the vertical temperature distribution in the very surface layer. The SST diurnal amplitude can reach more than 3 K in the Arctic Ocean (Eastwood et al. (2011)). To create DMI SST L4 product, only the observations between 21:00 and 7:00 local time are used (Høyer et al. (2014)), thus local diurnal variations of SST are supposed to be filtered out. Diurnal

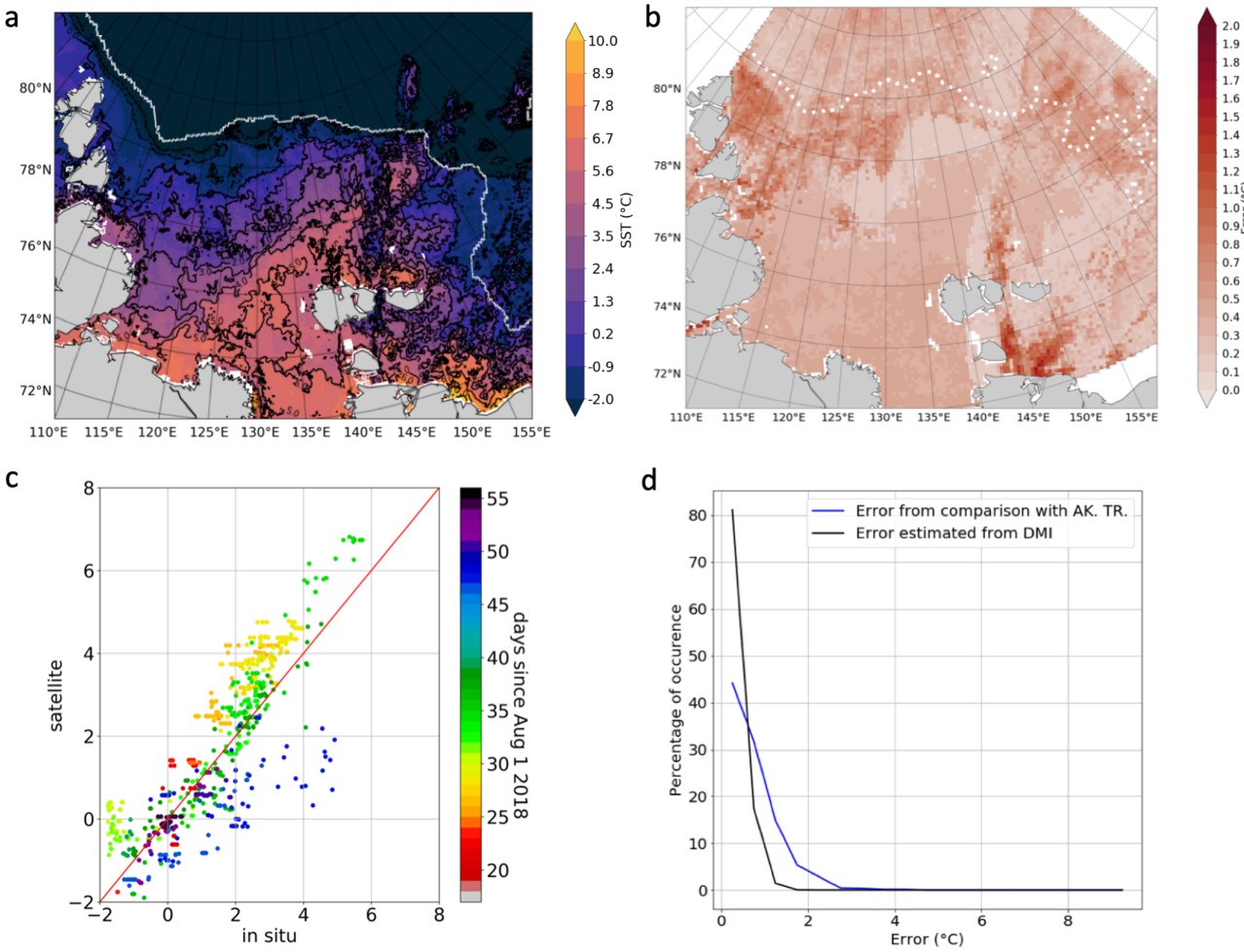

**Figure 4.** Sea surface temperature validation: example of DMI SST L4 image for September 13 (a) with the error estimates (b); comparison of collocated SST and *in situ* data (CTD and TSG) in the upper 6.5 m (c) and distribution of uncertainty provided by DMI and absolute difference derived from comparison with *in situ* data (d).

variation of temperature might be present in real *in situ* measurements during strong diurnal warming events, but no particular observations allowing to investigate this question were done during the cruise.

To illustrate the consistency of SST and *in situ* temperature datasets, September 13, 2018 was considered as it was one of the rare days in summer 2018, when the central part of the Laptev Sea was cloud-free, which is especially important for DMI SST.

The DMI SST product for September 13, presented in Fig. 4 (a) shows a rather complex pattern with a pronounced gradient associated with warm river water in the central part of the Laptev Sea. The uncertainty in SST estimates provided by DMI, is shown in Fig.4 (b, d). The percentage of occurrence is computed in temperatures classes by 0.5°C classes starting at 0°C(Fig.4,

d). The highest uncertainty (up to 2.5°C), was observed over some open sea areas that were partially cloudy, but was mostly associated with the sea ice due to its heterogeneity (Fig.4 (b)). Over most of the southern and the central part of the ice-free Laptev and the East-Siberian Seas, the uncertainty is below 0.5°C and over the eastern part, it is below 1°C.

Comparison of the DMI SST and *in situ* surface-layer temperature (Fig. 4, c) shows a good agreement almost independent on area and time during the ARKTIKA-2018 expedition. The correlation coefficient is 0.89, and the RMS difference is 0.77°C. The difference between mean DMI and mean *in situ* surface temperature data is 0.19°C. This excess average DMI SST seems to be possible, based on CTD measurements, indicating that the 0-3 m water layer is on average 0.3°C warmer than the 3-6.5 m layer (not shown). The largest deviations are observed when the ship is in the MIZ or a more compact sea ice, so they might be associated with either imperfect sea ice flagging of some stages of sea ice in the DMI SST product or a noise introduced after re-interpolation of data on a regular grid. This noise together with the different sampling of *in situ* potential temperature measurements and DMI SST product lead to a distribution of the absolute differences between *in situ* and DMI SST slightly wider than the one of uncertainties provided in the DMI SST product (Fig. 4, d). Nevertheless, this comparison should be taken only as indicative of a reasonable order of magnitude of the uncertainties given the limited number of *in situ* measurements for each uncertainty range. Overall, DMI SST agrees rather well with *in situ* data, and it captures a small-scale spatial variability of the SST in the ice-free areas (Fig. 4, a) well above SST uncertainties, so we use this product for the following analysis of SST time-series.

### 2.2.3 Sea surface salinity

Soil Moisture and Ocean Salinity (SMOS) is the first satellite mission carrying an L-band (1.41 GHz) interferometric microwave radiometer, which measurements are used to retrieve the sea surface salinity (SSS) in the first top centimeter. With the recent processing, the standard deviation of the differences between 18-day SMOS SSS and 100-km averaged TSG surface salinity measurements is 0.20 in the open ocean between 45°N and 45°S (Boutin et al. (2018)). However, the precision degrades in cold water as the sensitivity of L-band radiometer signal to SSS decreases when SST decreases, even though this effect on temporally averaged maps is partly compensated by the increased number of satellite measurements at high latitude (Supply et al. (2020)). A possibility of using SSS estimates in cold regions derived from L-Band radiometry has been recently demonstrated by several working groups (Tang et al. (2018), Grodsky et al. (2018), Olmedo et al. (2018)). However, existing L3 ("Level 3" means a product resampled at a uniform time-spatial grid, different from swath grid) SSS products: SMAP CAP/JPL (Soil Moisture Active Passive satellite, a product created using the Combined Active Passive algorithm by Jet Propulsion Laboratory) SSS or SMOS BEC (Barcelona Expert Center) SSS, are spatially averaged from 60 km to more than 100 km. SMAP REMSS (Remote Sensing Systems) SSS L3 v3 provides a 40 km resolution version, but do not provide a sufficient coverage in the Laptev Sea. The methodology developed in this study to retrieve SMOS SSS aims at maintaining SMOS original spatial resolution and at retrieving SSS as close as possible to the ice edge.

A new product, hereafter SMOS SSS "A" ("A" for the Arctic Ocean) L3, investigated in this study was computed using SMOS L2 ("Level 2" product means that a geophysical parameter, eg. SSS, was computed at the swath grid) SSS from the ESA (European Space Agency) last processing (v662, Arias and Laboratories (2017)), (Fig. 5, a). SMOS L2 SSS are available

on the ESA SMOS Online Dissemination website. SMOS SSS are representative of SSS integrated over about 50x50 $km^2$ given the footprint of SMOS radiometric measurements involved in the SSS retrievals. The SMOS ESA L2 SSS products are oversampled over an Icosahedral Snyder Equal Area (ISEA) grid at 15 km resolution. The oversampling on a 15 km grid is possible owing to the image reconstruction of the SMOS interferometric data, but in this processing we don't make any spatial average for SSS fields.

SMOS "A" SSS was obtained as described below. Seven-day running means were computed for each day and each pixel of the ISEA grid, with a temporal Gaussian weighting function with a standard deviation of 3 days. The full width of SMOS ascending and descending orbits swaths was considered in order to take advantage of better temporal and spatial sampling over the Arctic Ocean and to decrease the uncertainty with temporal averaging. In order to eliminate the SSS at very low and high wind speeds because of higher uncertainties, SMOS ESA L2 SSS was considered only if the associated ECMWF (European Centre for Medium-Range Weather Forecasts) wind speed was between 3 and 12 m/s. SMOS ESA L2 SSS measurements were also weighted relative to the uncertainty of the SSS measurement (as in (Yin et al. (2013), equation A7). This uncertainty was derived from information provided with the SMOS L2 products, the SSS "theoretical error", derived from the uncertainty of all the parameters used for retrieving SMOS SSS, multiplied by the normalized $\chi^2$ cost function of the SSS retrieval. Dinnat et al. (2019) showed that the Klein and Swift (1977) dielectric constant model was inaccurate at low SST. In order to mitigate this effect, a SST-dependent correction derived from Fig. 16 of Dinnat et al. (2019) (blue-circle line) was applied:

$$SSS_{SMOS-"A"} = SSS_{SMOS-ESA-L2} - (-5 \cdot 10^{-4} \cdot SST^3_{ECMWF} + 0.02 \cdot SST^2_{ECMWF} - 0.23 \cdot SST_{ECMWF} + 0.69).$$

Finally, a criterion on a SMOS-retrieved pseudo-dielectric constant (ACARD parameter, defined in Waldteufel et al. (2004))) was applied to discard SMOS measurements affected by sea ice (discarded when $ACARD < 45$). The uncertainty of SMOS SSS "A" was derived from the propagation of the uncertainty on individual SMOS ESA L2 SSS pixel during 7 days. The uncertainty strongly increases in the vicinity of sea ice (Fig. 5, b). For this reason, in the following study, above 75°N, all pixels with an SSS weekly uncertainty larger than 0.8 were not considered. South of 75°N, a higher threshold was used (1.5) allowing to maintain some measurements closer to fresh river water from the Lena and the Khatanga Rivers near the coast. In this area, the $\chi^2$ may increase due to the strong heterogeneity of SSS within SMOS multi-angular brightness temperatures footprints, and the number of measurements is low due to the presence of the coast and islands even without sea ice. The theoretical uncertainty of SMOS SSS "A" field is below 0.5 in the center of the Laptev Sea and up to 2 and higher close to the coastline and MIZ.

### 2.2.4 Validation of SMOS "A" SSS

In this section, we compared the SMOS SSS "A" relative to *in situ* measurements. Figure 5 presents the SMOS SSS "A" on September 13, 2018, the same day as the DMI SST in Fig.4. We co-located of SMOS SSS "A" and *in situ* measurements of salinity in the upper 6.5 m layer in a following manner: the averaging of the TSG salinity was done over one hour period (equal to ∼ 15 km distance, contrary to DMI SST validation) in order to be closer to SMOS SSS "A" spatial resolution. We used 985 collocated points.

Comparison between the *in situ* practical salinity and SMOS SSS "A" shows a very good agreement, not yet demonstrated

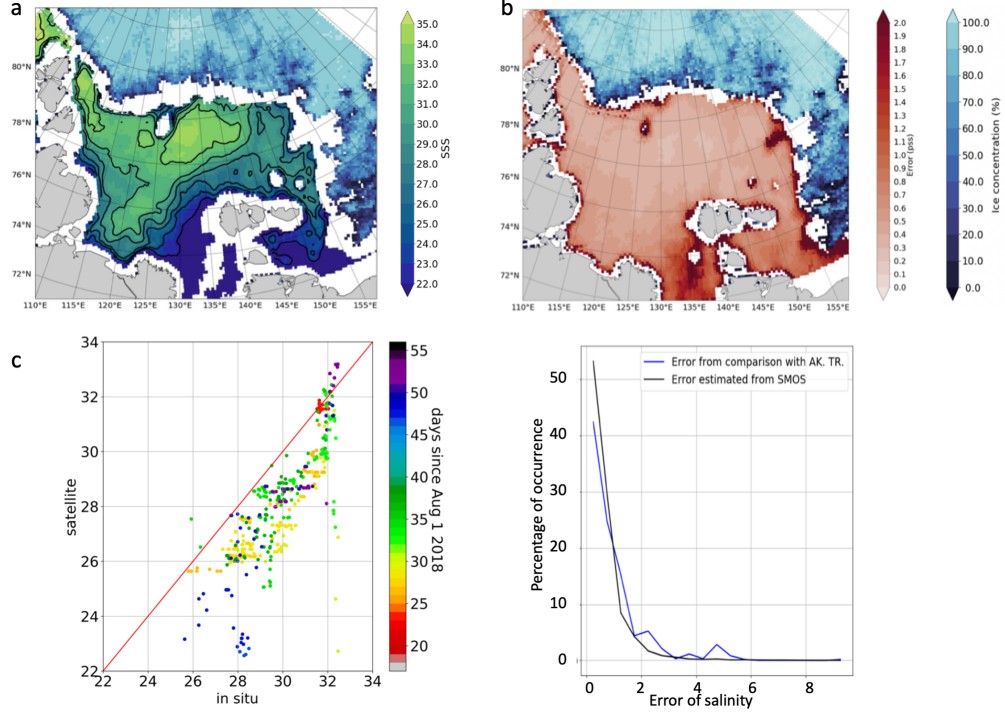

**Figure 5.** Sea surface salinity validation: example of SMOS SSS "A" for September 13, 2018 (a), and associated uncertainties (b), scatter plot of co-located SSS and *in situ* data in the upper 6.5 m (c) and statistical distribution of provided SMOS L2 uncertainties and measured absolute difference from comparison with *in situ* data (d). Sea ice concentration from AMSR2 is indicated with blue shading on the upper panels

before by any other salinity product in the Laptev Sea. The correlation coefficient is 0.86 with a RMS = 0.86. The mean difference is 2.06, SMOS SSS being lower than *in situ* surface salinity. This underestimate could be related to the presence of land in the very wide SMOS field of view as observed at lower latitudes by Kolodziejczyk et al. (2016), and this difference appears to be, at first order, systematic (Fig.5, c). In what follows, we subtract this mean difference from the entire SMOS SSS

5   dataset. The standard deviation of SMOS SSS with respect to *in situ* SSS does not vary with the depth of the *in situ* salinity measurements above 6.5 m, either because *in situ* salinity was homogeneous vertically or because comparisons were too noisy to detect these small variations (not shown). Although SMOS SSS "A" shows a good agreement most of the time, some larger uncertainties occur close to the sea ice margin or when pixels are contaminated by small ice pattern not detected by AMSR2 sea ice concentration algorithm (as at 80°N 125°E in Fig. 5, a).

10   Comparison between SMOS uncertainties and error based on comparison with *in situ* salinity measurements is presented in Fig. 5 (d). The percentage of occurrence is computed in salinity classes with a size of 0.5 that starts at 0. It shows a rather good agreement between the distribution of SMOS SSS "A" uncertainties estimated from retrieval process and the distribution of

error obtained from comparison with *in situ* salinity measurements. The uncertainties are in 85% of cases less than 1.2, which is relatively small compared to spatial gradients shown on Fig.5, a. These results allow us to use the SMOS SSS "A" error with confidence for this analysis. Using error filtering, the points too close to the ice edge were excluded.

### 2.2.5 Sea ice concentration and ice masks

Sea ice masks were obtained from AMSR2 sea ice concentrations products provided by the University of Bremen (Spreen et al. (2008)): they are weather-independent, thus, continuous for the whole period. The highest available spatial resolution is 3.125 km. The AMSR2 ice masks were used in addition to the masks provided with every satellite product discussed (DMI SST, SMOS SSS "A", ASCAT (Advanced SCATterometer) winds L3 (see its description below)). A continuous erroneous presence of ice along the Siberian coast was observed and had to be filtered: images in optical band and the ice charts from the Arctic and Antarctic Research Institute (AARI) were used as a reference (can be found at http://www.aari.ru/odata/_d0004.php). As it was detailed above, an additional filtering was applied to SMOS SSS "A" as the L-Band measurements are sensitive to ice thicknesses less than 50 cm contrary to AMSR2 measurements.

The sea ice opening starts relatively late in the Laptev Sea: a coastal polynya appeared in the southern-central part of the Laptev Sea at the beginning of June in 2018 and by the beginning of August, the sea was ice-free only south of 79°N. The Laptev Sea was completely covered by the beginning of November in 2018. For this study, we define the sea ice edge with the position of 1% sea ice concentration and MIZ as 0-30%.

### 2.2.6 Wind speed

To investigate the wind speed pattern, we use ASCAT scatterometer daily C-2015 L3 data produced by Remote Sensing Systems. Data are available at www.remss.com.

### 2.3 Reanalysis data

Reanalysis data are used to include some additional parameters not available from satellite and *in situ* data. Atmospheric forcing fields: sea level pressure, SLP, and air temperature, are obtained from the ERA5 reanalysis (Bertino et al. (2008)). The latest reanalysis of ERA5 still has relatively crude spatial grid of 0.5°for the SLP and 0.25°for air temperature.

### 2.4 Ekman transport

To investigate the role of the wind forcing, we compute mean monthly wind fields and the Ekman transport for August and September 2018. Horizontal Ekman transport, $m^2/s$, is calculated as:

$$
\begin{aligned}
u_{Ekm} &= \frac{\tau_v}{\rho_w * f} \\
v_{Ekm} &= -\frac{\tau_u}{\rho_w * f}
\end{aligned}
\tag{1}
$$

where $u_{ekm}$ and $v_{ekm}$ are horizontal components of the Ekman transport, $\tau$ is a wind stress, calculated from ASCAT winds $(u_{wind}, v_{wind})$ using ERA5 air density $\rho_{air}$: $\tau_u = C_D * (u_{wind}) * u_{wind} * \rho_{air}$; $\rho_w$ is a surface density, calculated from SST and SSS with TEOS-2010 (McDougall et al. (2009)); $C_D$ is surface drag coefficient calculated from wind speed according to Foreman and Emeis (2010): for the wind speed $U_w$ below 10 m/s $u_{star} = 0.051 * U_w - 0.14$ and for the stronger winds:
5  $u_{star} = 0.051 * (U_w - 8) + 0.27$; $f$ is the Coriolis parameter.

## 3  Results

### 3.1  Overview of SST and SSS in the Laptev and East-Siberian Seas in August-September 2018

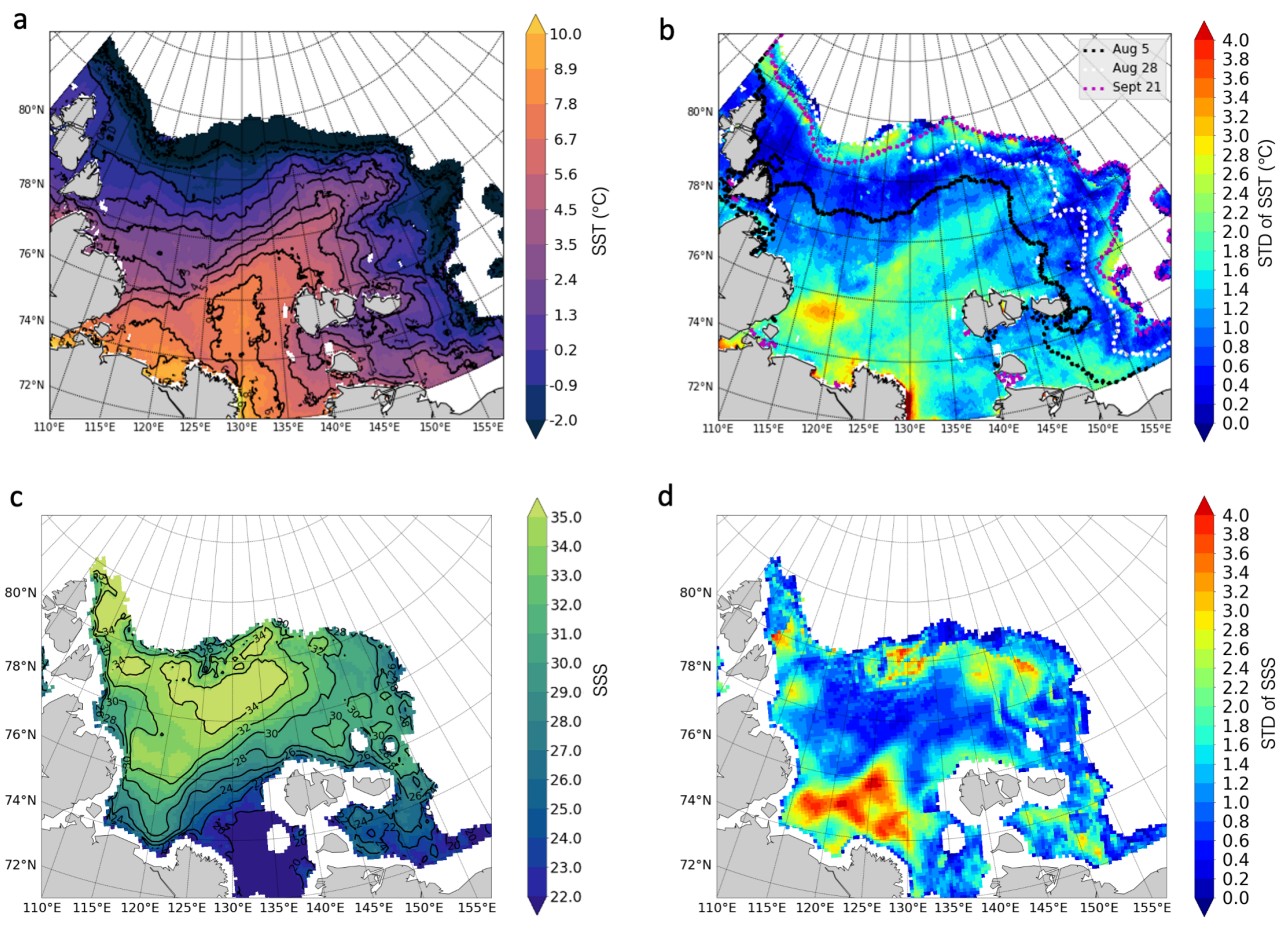

**Figure 6.** Mean DMI SST (a) with its STD (b) and mean SMOS SSS (c) with its STD (d) for August-September 2018. The dotted lines in Figure (b) show the position of sea ice edge at different moments of time before and during the ARKTIKA-2018 cruise

The mean SST during the 2 summer months is 2.18°C in the Laptev Sea (between the Severnaya Zemlya Archipelago and the New Siberian Islands), and 1.13 °C in the part of the East-Siberian Sea investigated (Fig. 6). The highest temperatures (above 6°C, up to 9°C) were observed close to the Lena River delta in the Yanskiy Bay and in the Olenekskiy Bay in front of the Khatanga River. A warm water pool associated with the river plume between 125°E and 135 °E progressively propagates north-eastward and warms up this part of the sea: 0°C isotherm at 140°E meridian is situated 100 km northward compared to its position at 120°E. The studied part of western East-Siberian Sea was not completely ice-free in August-September 2018. Negative temperatures are observed near the ice edge at a distance of 50-100 km of the ice edge almost everywhere, except for a small area at 80°N 160°E, where warm river water meets the sea ice with no open water with negative temperatures. The strongest gradients are observed along the sea ice edge and the river water plume (up to 0.05° C/km). Standard deviation of SST in Fig.6 is the largest in the Olenekskiy Bay (over 2.5°C), along the coastline close to the Khatanga estuary (2.5-3°C), the Lena River delta (about 4°C) and in marginal ice zone (mostly over 1.5°C). The remarkable variation of SST in the central part of the Laptev Sea should be associated with the thermal fronts (largest SST gradients) displacement.

The averaged SSS is 28.75 (with uncertainty of 0.10) in the Laptev Sea and 27.74 (with uncertainty of 0.20) in the western East-Siberian Sea (Fig. 6). The spatial distribution of mean salinity for August-September 2018 shows the freshest water (salinity below 20) within the river plume northeast of the Lena River delta and within the southern part of the East-Siberian Sea. Water with salinity below 28 reach the sea ice edge in the northeast Laptev Sea. Additional fresher water from the Kara Sea enters via the Vilkitskiy and Shokalskiy straits in the west (salinity of 28-30) and is also observed along the sea ice edge, where it could be associated with ice melting. The most saline water (salinity above 34) is located in the central part of the Laptev Sea near 78-80°N 120-140°E, and in the northwest, along the Severnaya Zemlya Archipelago. As also observed in SST, SSS in the Olenekskiy Bay is highly variable, which can be explained by the variation of the freshwater discharge during the 2 months. Nevertheless, large SSS variability is also observed all along the sea ice edge: at 78-80°N in the north and northwest and at the boundary between the Laptev and East-Siberian Seas. This large variability can be explained in two ways: physical (haline fronts related to sea ice melting) and instrumental (remaining ice contaminated pixels, lower sensitivity of L-band in cold water). At 78-80°N 125°E, free-floating patches of broken ice detached from compact sea ice edge are observed during several weeks in August-September 2018. Random pieces of broken sea ice are not always recognized by ice-mask filters, so can artificially increase SSS variability. At the same time, this is the area where river water encounters sea ice, which induces natural variability.

## 3.2   Observed surface water masses of the Laptev Sea and their transformation

To generalise our understanding of vertical structure of the studied area, we use the classical TS-analysis, first based on CTD measurements. Fig. 7 shows the temperature-salinity distributions in the upper 200 m, coloured as a function of depth. The most prominent feature on the diagram is the transformed Atlantic Water mass with salinity close to 34.5-35, temperatures from -0.5 to 2.5 °C lying at a depth of 100-200 m. The water mass overlying the Atlantic Water (between 50 and 100 m depth) is the lower halocline water, described by Dmitrenko et al. (2012) as having salinity in a range of salinity 33-34.5, and negative temperatures starting from the lowest values presented in Fig. 7, -1.7 to 2.5°C. The surface water observed in the upper 50

metres is in general less saline (salinity below 34), but we can clearly observe two separate branches with negative and positive temperatures. The two upper-layer branches are (1) warmer ($[-1; 6]°C$) and low-saline (below 34) surface water of the ice-free Laptev Sea and (2) colder ($[-2; 0]°C$) and low-saline waters of the ice-covered East-Siberian Sea. The latter correspond to the measurements from the sections 7 and 8 eastward of 150°E.

5    It should be remembered that a T-S diagram based only on CTD measurements does not provide an instantaneous view on the ocean state, but is a collection of conditions encountered in different regions at different times (from the end of August to the end of September 2018). During the summer months, the surface water of the Arctic Ocean quickly evolves, and the synoptic satellite data provide an additional information to the point-wise *in situ* measurements.

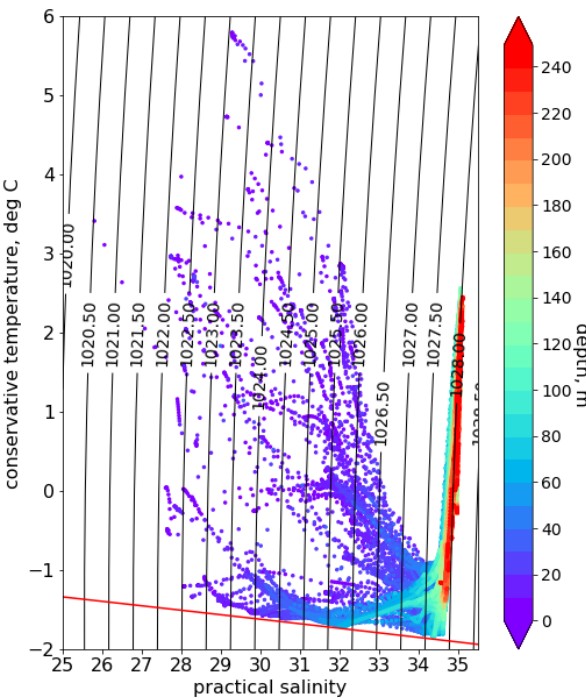

**Figure 7.** T-S diagram based on the CTD data in the upper 250 m, colour-coded by depth. The red line shows the freezing line.

Using DMI SST and SMOS SSS weekly estimates, we plotted T-S diagrams similar to the one in Fig.7, but only for surface
10    satellite measurements for several reference days: Aug 1, Aug 15, Aug 30, Sept 4, Sept 13, and Sept 30, 2018 (Fig.8). On the lower row, we present all *in situ* measurements in the upper 6.5 m and the differences between satellite-derived sea surface temperature and salinity for the selected day. The range of variation in SSS and SST values covered by the satellite measurements (first row of Fig.8) is an order of magnitude wider than the one covered by *in situ* measurements (Fig.8, first column,

bottom row). The difference in T-S diagram covered by each type of measurement cannot be explained the errors of satellite measurements (RMS difference with respect to *in situ* measurements of 0.77°C and 0.8 in temperature and salinity, respectively), nor by the uncertainties associated with each satellite product (Fig.4, b and Fig.5, b). It primarily reflects the much more extensive spatio-temporal monitoring of different water masses by satellite measurements: e.g. the in situ measurements miss the southern Laptev sea, close to the origin of the riverine water. The DMI SST increases only up to the end of August with the maximum temperatures from 8 to 11.5°C in some cases, and then decreases to 4.5°C by the end of September. The temperature is changing by 0.5 - 1°C per week (while increasing and decreasing).

Based on the Fig. 8 visual analysis, we propose to identify 6 surface water masses in the Laptev and East-Siberian Seas (Tab.1) to follow the transformation of surface waters during 2 summer months. The number and the limits of water masses were arbitrarily chosen based on the temperature-salinity scatter plot for September 4, 2018, as this day allows to separate the cores of surface waters into groups in the best way based on the density of points. The temperature and salinity ranges of variation of each class are also well above the T and S uncertainties.

The main surface water masses are warm and fresh (WF) river water and cold and saline (CS) open sea water. All other water masses show either different stages of transformation of these two water masses, or are advected from other regions. It should be noted that satellite-derived data have a larger range of temperature and salinity than the near-surface (upper 6.5 m) *in situ* measurements, which enables this detailed classification. The locations of the different water masses for specific days are shown in Fig.9 together with the percentage distribution of water masses (the whole studied area is 100%, and sea ice occupies some part of it).

On August 1, the sea ice still covers more than 80% of the studied area and extends on average to 78°N in the Laptev Sea, while the East-Siberian Sea is almost completely covered by ice. Warm and Fresh (WF) river water is well observable in the southern parts between 74 and 76°N. It occupies almost the same amount of surface as the Cold and Saline sea Water (CS), the rest of the open area is occupied by a transformed river water (Warm and Medium Salinity, WMS, Cold and Medium Salinity, CMS), that already formed a recognisable river plume front: its signature is continuous from 115°E to 150°E up to the northern position of sea ice edge.

During the next two weeks the ice cover retreats, and a Cold and Fresh Water (CF) mass appears in the south-west East-Siberian Sea. The amount of this Water increased progressively in this area during the remaining period. We suggest that this water mass represents the river water trapped under the ice and then exposed (see results of geochemical analysis below in Section 3.3.4).

On the 15th of August, a water mass CMS also appears close to the Vilkitskiy Strait. It is less pronounced by the end of August, but a thin stream of cooled and transformed river water from the Kara Sea extends along the Taimyr peninsula in September. The Lena River water mixing and cooling happens as well close to the sea ice edge in the north-east Kara Sea. As a whole, the surface occupied by this water mass is steadily growing during the observed period to reach nearly 10% of the surface by the end of September. We suggest that water mass CMS is a transformed version of water mass CF.

**Table 1.** The temperature and salinity of six defined surface water masses of the Laptev Sea using satellite data (see the text for the explanation of water masses names)

| Water mass | WF | WMS | CF | CMS | WS | CS |
|---|---|---|---|---|---|---|
| T | $> 3°C$ | $> 3°C$ | $< 3°C$ | $< 3°C$ | $> 3°C$ | $< 3°C$ |
| S | $< 25$ | $25 - 29$ | $< 25$ | $25 - 29$ | $> 29$ | $> 29$ |

The end of August is warmer as seen in Fig. 9 with the amount of saline water with temperatures above $3°C$ (water mass WS, Warm and Saline) occupying the central and the western part of the Laptev Sea (almost 10% of the studied area). This water mass disappears by the end of September with the seasonal decrease of temperature.

By September 13, the SST and SSS variability diminishes. The water mass CF in the north-east Laptev Sea consisting of cold fresh water becomes saltier (transforms into the water mass CMS). The freshwater cools south of the New Siberian island and by September 25 occupies all the ice-free area. The river plumes signature shifts as well to the New Siberian island as well (Fig. 9). Cold and saline water dominates the surface of the Laptev Sea. Finally, by September 25, the T-S diagram shows that most of the SSS/SST points lay between 25 and 35 and -1°C and 4°C, with a main core within a salinity range 25-35 and temperature between -1 to 1°C, and the second one within the salinity range 22.5-30 and temperature of 3-4°C. The Laptev and the East-Siberian Seas start then to refreeze, the most rapidly in the areas with cold and fresh river water.

### 3.3 Freshwater variability in the Laptev Sea

To evaluate the distribution of freshwater input in the Laptev Sea in August-September 2018, we consider zonal and meridional transects along 78°N and 126°E, respectively, and plot the temporal evolution of DMI SST, SMOS SSS "A", wind speed and SLP in Hovmöller diagrams. The freshwater can be defined by comparison to the saline "marine water" (typically, 34.80 as in Aagaard and Carmack (1989) or 34.92, as in Bauch and Cherniavskaia (2018)). As a 0-salinity river water quickly mixes with a saltier marine water, in reality the "freshwater" is more "brackish" than "fresh". Nevertheless, for simplicity assuming a river plume front at the 29 isohaline, the "freshwater" corresponds to all water masses with the salinity lower than 29, as we referred to it in section 3.2.

### 3.3.1 Water from the Lena River plume

The zonal transect helps to investigate the mean stream position of the river plume away from the coast, in the central part of the Laptev Sea with more complex topography (Fig. 10). This virtual section does not correspond to any real CTD-section, apart from some TGS profiles following the ship's route (see the position of virtual section on the SST and SSS maps in Fig.10, f-g). In the western part (up to 130 °E), the transect is located roughly above the continental slope and then over the shelf (Fig.10, e). The river water displacement roughly follows that of sea ice edge in the east and is bounded by the shelf break in the west. Overall, temperatures are higher in August than in September: a warm pool with SST over 6 °C is observed during the first 30 days at 78°N, 130-147 °E, with highest temperatures on August 26. These coordinates define the position of the river plume

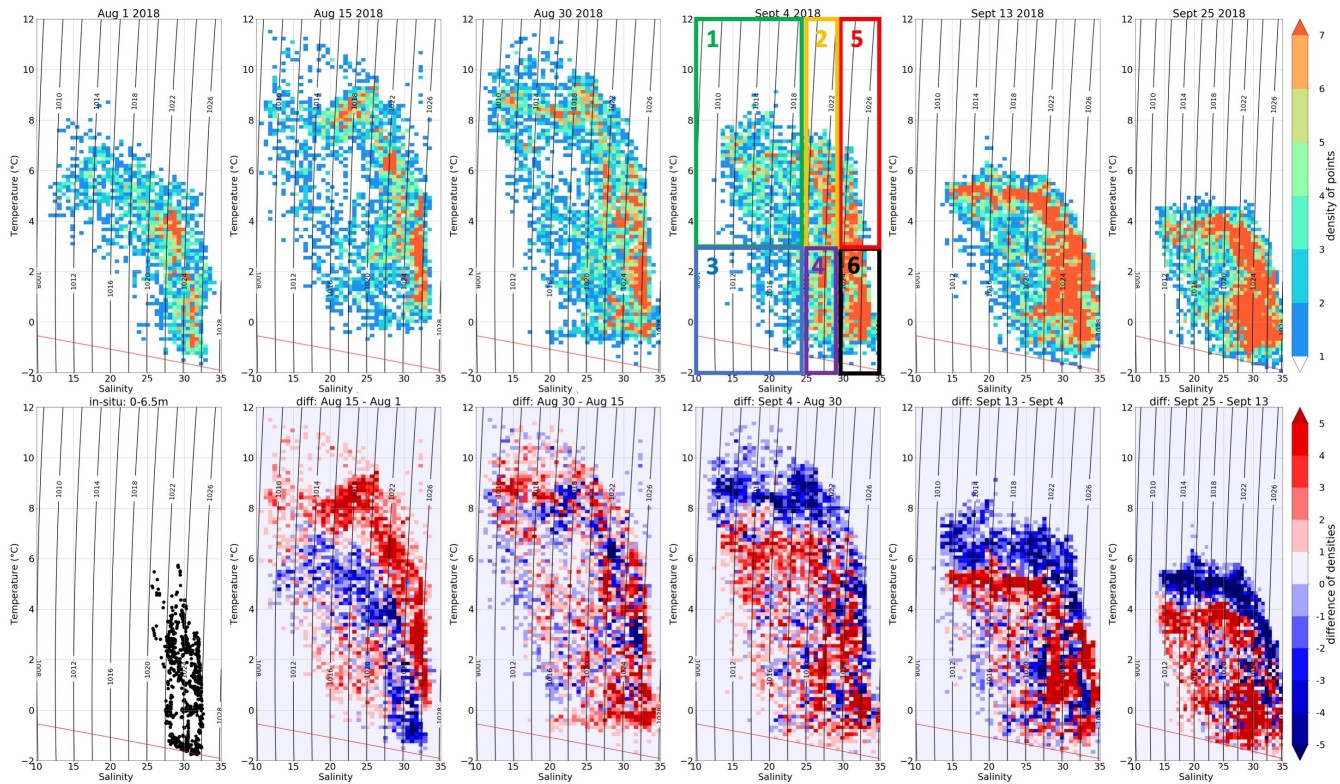

**Figure 8.** Temporal evolution of surface water masses in August-September 2018 for the following reference days (upper row): Aug 1, Aug 15, Aug 30, Sept 4, Sept 13, and Sept 30, 2018. colour represents the density of points (number of observations with this temperature and salinity). The red line shows the freezing point temperature for different salinity. The boxes show the cores of 6 water masses described in text: 1 - WF, 2 - WMS, 3 - CF, 4 - CMS, 5 - WS, 6 - CS. Lower row: column 1, the T-S diagram based on CTD measurements in the upper 6.5 m only, and from column 2 to 6, the differences (in density points) between the reference days.

at 78°N latitude, as can be clearly seen in the salinity values varying in a range of 27-30. Relatively strong daily winds (10-12 m/s) observed during the first 10 days of September were associated with a series of cyclones, which strongly impacted the surface layer: the median temperature over the zonal transect decreased from 3°C to almost 0°C, and salinity increased by 1. As the amount of incoming solar radiation diminishes in September, the maximum SST values did not exceed 3°C anymore.

5  Nevertheless, at the end of September, a new freshwater patch was observed at 140°E (less visible in SST field) indicating that the "upstream" surface mixed layer (in the southern part of the Laptev Sea) contained a sufficient amount of freshwater to restore its previous state after a mixing event induced by the wind. Another possible explanation is that a small peak observed in the Lena River discharge in the first days of September (Fig.3) introduced an additional portion of freshwater that reached 78°N several weeks later.

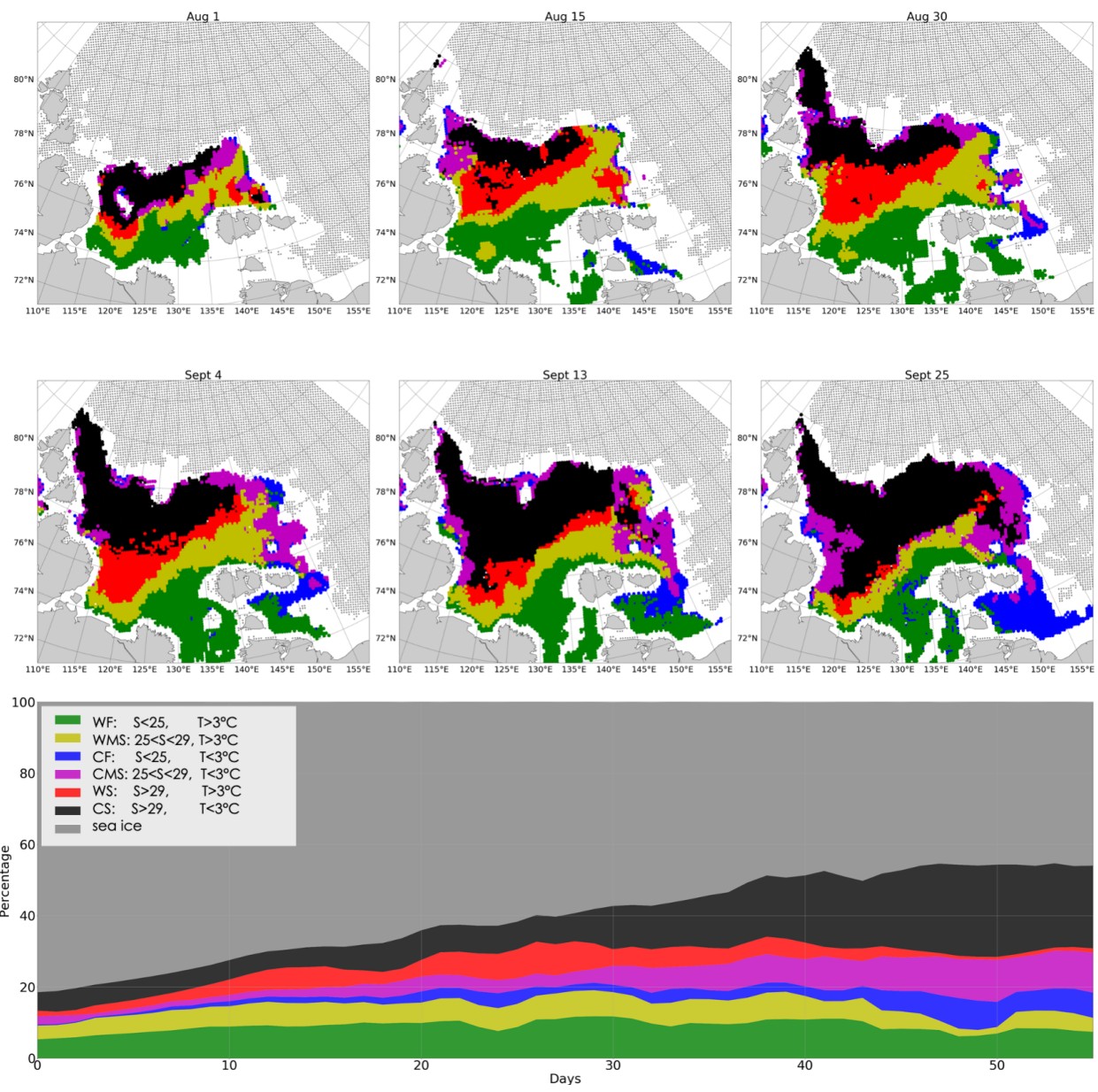

**Figure 9.** Spatial distribution of surface water masses in August-September 2018: upper row - Aug 1, Aug 15, Aug 30; middle row - Sept 4, Sept 13, Sept 30. Sea ice cover from AMSR2 is plotted as dashed area. The lowest panel show temporal evolution of surfaces occupied by each water mass or sea ice cover in the Laptev Sea (in % of the Laptev Sea surface).

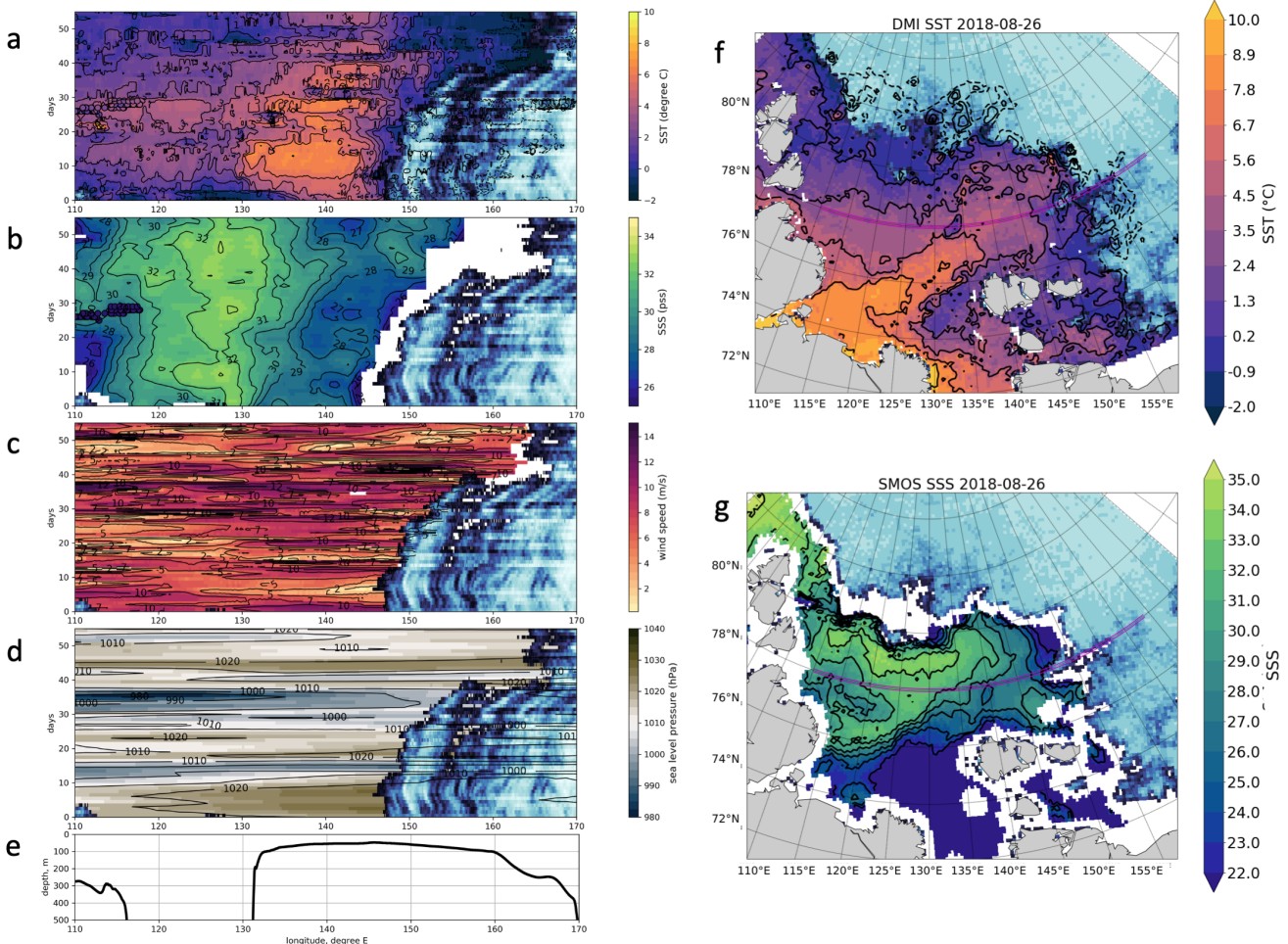

**Figure 10.** Hovmöller diagram of DMI SST (a), SMOS SSS "A" (b), ASCAT wind speed (c), and ERA5 sea level pressure (d) for the zonal transect at 78°N. Small coloured circles on SST and SSS diagrams (a, b) show *in situ* measurements of temperature and salinity (first CTD or TSG at 6.5 m). Sea ice concentration (AMSR2) is indicated with a blue colour, see Fig.5 for the colour scale. The bathymetry along the virtual transect (e) is extracted from "1 Arc-Minute Global Relief Model" (Amante and Eakins (2009)). The position of a virtual transect is shown on DMI SST and SMOS SSS "A" maps for August 26, 2018 (f, g) with magenta lines.

### 3.3.2 Water from the Kara Sea

The zonal transect allows to see not only the Lena River plume, but as well the Kara water intrusions in the west. The selected zonal transect at 78°N is partly lying in the Vilkitskiy Strait connecting the Kara and the Laptev Seas. Being a reservoir for two other great Siberian Rivers, the Ob' and the Yenisei, the Kara Sea has a low salinity compared to the central Arctic Basin

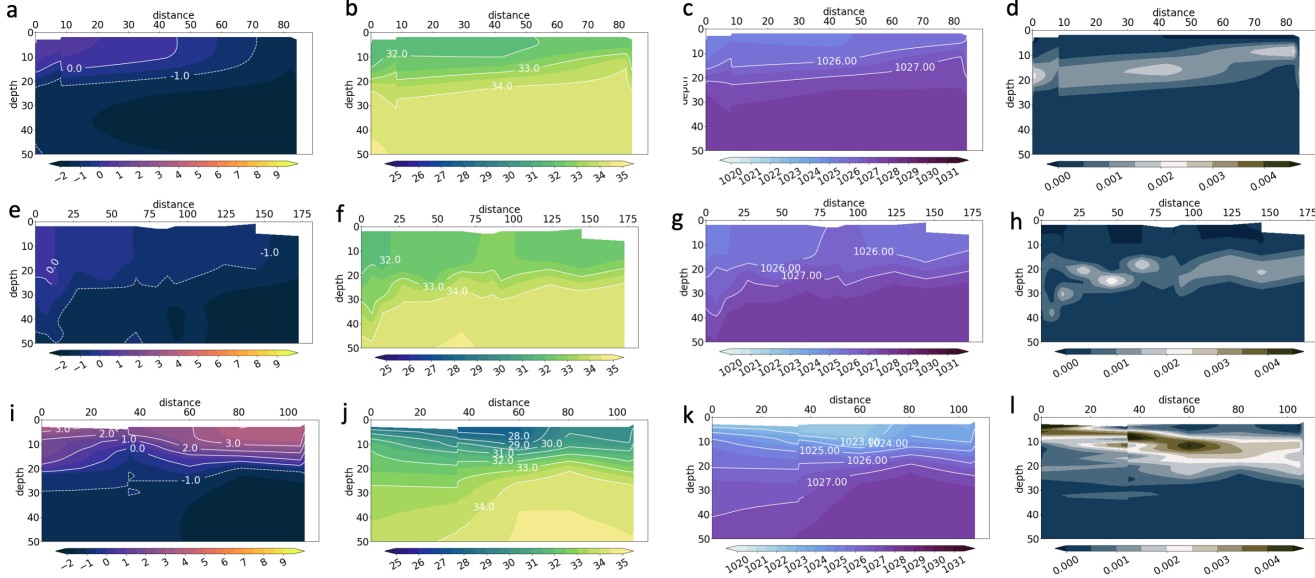

**Figure 11.** Temperature, °C, (left, first column), salinity, (second column), water density, $kg/m^3$ (third column) and buoyancy frequency, $s^{-1}$, (right, fourth column) obtained from CTD measurements in the upper 50 m for section 1 northward of Arkticheskiy Cape (upper row), section 10 across the Shokalskiy Strait (second row), and section 4 across the Vilkitskiy Strait (lower row). See Fig.1 for the section's positions. The zero km is always placed at the southern point of each section

(Janout et al. (2015)). In the absence of significant river sources on the Severnaya Zemlya Archipelago, we considered that the freshwater input close to the Vilkitskiy and the Shokalsky Straits arrived from the Kara Sea.

We observe the freshwater arriving from the Kara Sea at 110-115°E with typical values of 25-28 during the first 20 days of August and at the end of September (Fig.10, b). It is noteworthy that the SST fields do not indicate so clearly the presence of these intrusions. This suggests that fresh and warm water of the Ob' and Yenisei rivers arriving to the Laptev Sea have already lost a significant part of their heat content to the atmosphere, but that the freshwater layer is not completely mixed with the surrounding sea environment.

In Fig. 11, the CTD data justify that the amount of freshwater arriving from the Kara Sea through the Vilkitskiy Strait is significantly greater than freshwater arriving via the narrow and rather shallow (250 m) Shokalskiy Strait between the Bolshevik and the October Revolution Islands or north of the Severnaya Zemlya Archipelago at the traverse near the Arkticheskiy Cape across the continental slope. The temperature of the surface layer is increasing between 0°C and 3.5°C from north to south. Salinity sections indicate freshwater with salinity above 29 only in the Shokalskiy and the Vilkitskiy straits, which suggests very little advection of the Kara-origin freshwater via the north. From the buoyancy cross-sections, we find that the strongest stratification is at 5-20 m depth, which corresponds to the 1024-1025 $kg/m^3$ isopycnals depths. This result argues against

a definition of fresh-water content by the 1027.35 $kg/m^3$ isopycnal of Polyakov et al. (2008), as the surface salinity and temperature in the Siberian shelf seas are lower than in other regions.

### 3.3.3 Meridional transect

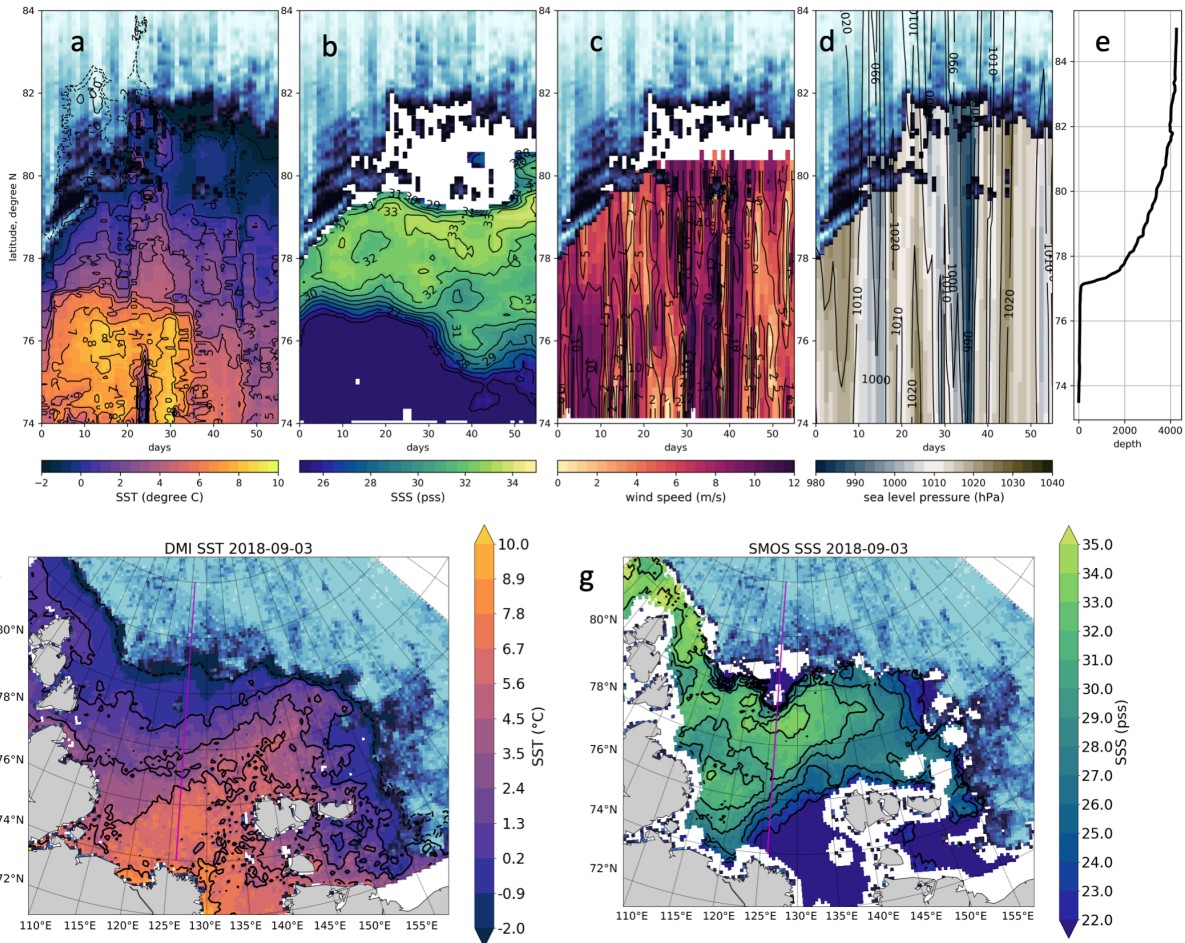

**Figure 12.** Hovmöller diagram of DMI SST (a), SMOS SSS "A" (b), ASCAT wind speed (c) and ERA5 SLP (d) for the virtual meridional transect at 126°E. Sea ice concentration (AMSR2) is indicated with a blue colour, see Fig.5 for the colour scale. The bathymetry along the transect (e) is extracted from "1 Arc-Minute Global Relief Model" (Amante and Eakins (2009)). The position of a virtual transect is shown on SST SMI and SMOS SSS "A" maps for August 26, 2018 (f, g) with magenta lines.

The meridional transect along 126°E (Fig. 12) partly corresponds to the standard oceanographic section 5 carried out during ARKTIKA-2018 expedition on September 1-4, 2018 (Fig.13). This transect helps to understand the northward propagation of the river plume and to evaluate the freshwater content using *in situ* data. The highest SST observed along 126°E longitude is 8°C in August (please note, that a small cold temperature intrusion on days 22-26 probably corresponds to an error in DMI

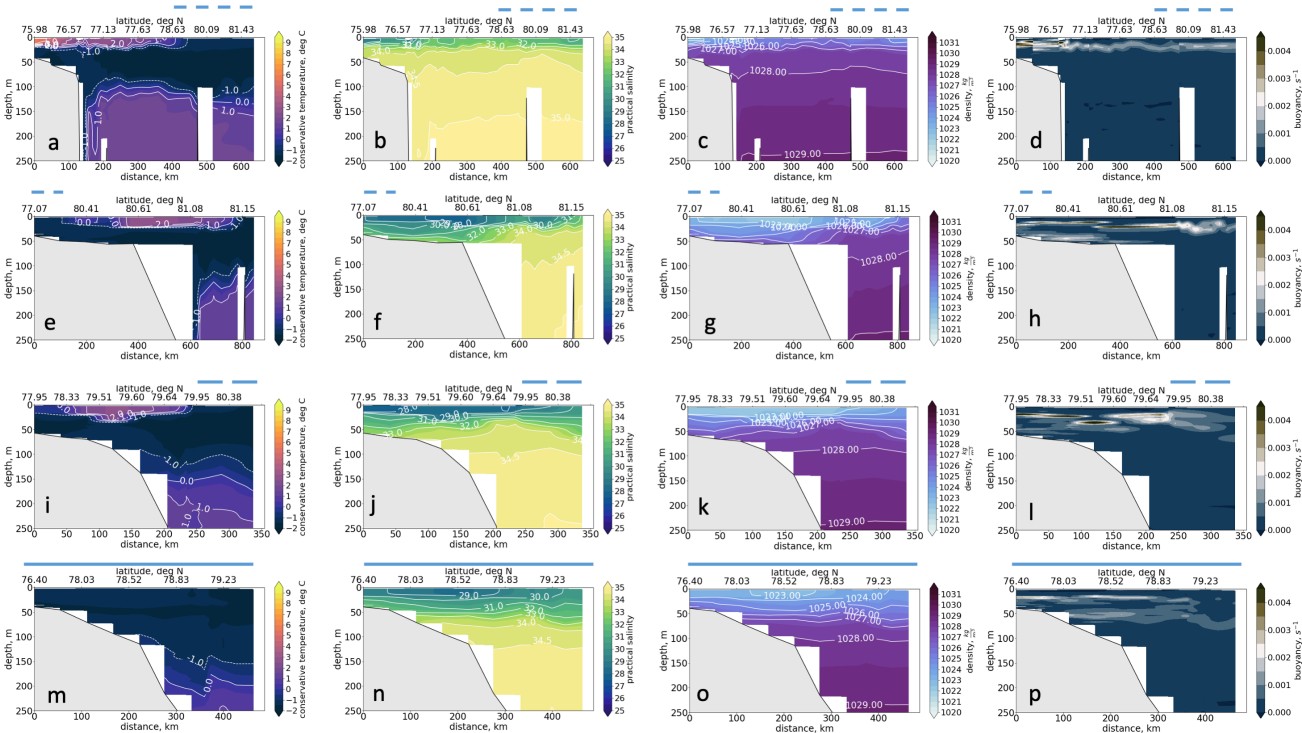

**Figure 13.** Conservative temperature (left, first column), practical salinity (second column), density (third column) and Brünt-Vaisälä frequency (right, last column) in the upper 250 m along oceanographic section 5 (a-d); section 6 (e-h); section 8 (i-l); and section 7 (m-p). See Fig.1 for the section positions. The zero km is always placed at the southern point of each section. The dashed blue line indicates the MIZ for sections 5, 6 and 8 (the rest is ice-free area); the full line indicated that the section 7 was done under the ice.

SST product due to a cyclone passage and thus, either bad cloud masking or strong winds. This is an assumption reinforced when comparing DMI SST to SST AMSR2 microwave data (not shown here). More information on the SST corrections in the Arctic can be found in the work of Høyer et al. (2014)).

The warmest (5-9°C) and freshest (salinity of 20-30) water of river plume occupies the area between 74-77°N in August and
5   progressively retreats in September: SST and SSS gradients become wider and less pronounced, temperature decreases to 3-4 degrees. High wind speed (10-12 m/s) associated with an atmospheric cyclone passage during the first two weeks of September is found both on the meridional and the zonal Hovmöller diagrams and might explain this widening of the surface thermal and haline frontal area. Nevertheless, a point-wise cross-correlation between the time-series of wind speed and temperature or wind speed and salinity does not give statistically significant results: both correlation coefficients are below 0.2 at any time lag (0-10
10  days). Better correlation is observed with sea level pressure (up to 0.6 at some points), but over the 56 days investigated it is not statistically representative as only two passing cyclones were observed.

The oceanographic sections allow to estimate a thickness of the freshwater layer and how far the river water propagates under the ice. Section 5 provides complementary information to the meridional Hovmöller diagram (Fig.13, upper row) as it was done along the same 126°E parallel from 76 to 81.4° N on September 1-4 2018. This date corresponds to the passage of several cyclones over the Laptev Sea, which, in turn, displaced the river front to the south, unfortunately, almost away from this

oceanographic section. Nevertheless, at 76-78° N (first 200 km of the section), low salinity between 29-33 was still observed in the upper 25 meters. A thin upper layer with positive temperatures has the same thickness, but extends further northward, up to 79° N. In the northern part of the section, under the ice, the temperatures are below 0°C and salinity is rather low, below 32. The low salinity under the ice suggests that the remnants of the river water arrived in this area earlier. If the river water were propagating under the ice when the Laptev Sea was not yet completely open, we should assume their further mixing

with sea water when the sea started to open in its central part (mixed water with salinity between 30 and 32 and still positive temperatures). The heat exchange with the sea ice might be more effective than with the atmosphere, so under the ice the temperatures are negative, and the warm river water signal is not observed anymore, contrary to salinity. At the same time, it depends on thermal conductivity in the ice, and its initial temperature profile, so this question needs further attention.

Overall, the first 150 km over the shelf, where the warmest and freshest water were observed, are characterized by the

strongest stratification in the upper 25 m layer. This is the depth of a stable stratification for the whole section, though stratification is less pronounced in the deeper part of the sea than over the shelf. Below the pycnocline, we observe cold (with negative temperatures) and saline (salinity between 33 and 34.5) water mass. The warm (T above 0°C, following Pnyushkov et al. (2018)) and saline (S above 34) Atlantic Water spreading along the continental shelf is best identified in temperature vertical profiles at 100-120 m depth, but is also detected by the instability signal (right column in Fig. 13). The propagation

of the Atlantic Water is beyond the scope of this paper, and though Atlantic Water is observed in all oceanographic sections presented below, it won't be discussed furthermore.

When considering other meridional sections (6, 8, and 7 according to their positions), we follow the eastward propagation of the river water away from the Lena River delta. Section 6 started on September 5 in the vicinity of the marginal ice zone in the deep North-Eastern part of the Laptev Sea and ended in the ice-covered part of the East-Siberian Sea over the shelf on

September 9. This section is not exactly perpendicular to the continental slope, so we cannot estimate the width of the river water plume, but overall the thickness of the upper layer is similar (20-30 m) to that observed with section 5 in the deep part of the section. The waters over the shallowest part (depth smaller than 60 m) were observed under the ice, as is clearly seen in the temperature signal that is negative even close to the surface. At the same time, the main freshwater core with the highest temperature is observed above the shelf break. The second core is observed in the northern part of the section, with lower

salinity than in the north of section 5. The mixing over the shelf was effective enough to stretch the isopycnals between the bottom and the surface. Nevertheless, the depth of the maximum stratification is close to 20 metres as for the shallow part of the section 5. Over the edge of the continental slope, the maximum Brünt-Väisälä frequency moved deeper to 25 m, and over the deep-water part to 30 m depth.

Section 8 started on September 15 in MIZ over the deep part of the East-Siberian Sea and finished by September 17 in the

ice-free area over the shelf. The river signal is still very pronounced both in temperature and in salinity profiles, with an efficient

mixing over the 60 m layer on the shelf and more concentrated isopycnals over the shelf edge. The most eastern section 7 was conducted under the ice. The temperatures are, thus, negative above the Atlantic Water, but salinity profile reveals the river water presence with the freshwater core having salinity below 29. The maximum value of Brünt-Väisälä frequency are less than for other sections and are observed at 20 m depth and at 55 m depth, following 1024 $kg/m^3$ and 1026.5 $kg/m^3$ isopycnals, accordingly, showing the maximum stability of water vertical stratification under the ice.

To summarize, during the summer 2018, we observe a north-eastern displacement of the Lena River water including in the MIZ and ice-covered area. We suggest that the active displacement started in the ice-covered conditions after the maximum of river discharge in June-July (following the Papa et al. (2008) study and the Lena River discharge measurements presented in Fig.3), then, with progressive opening, part of the river water was mixed within the upper sea layer and exchanged heat with the atmosphere. For the water under the ice, the heat flux from the river water to the sea ice resulted in cooling of this water to the ambient negative temperature, but, at the same time, the sea ice protected the freshwater layer from wind-induced mixing, so it conserved a pronounced salinity signal.

### 3.3.4 Tracing surface water origin using oxygen isotopes (delta-O18)

The oxygen isotopes are considered as a "natural tracer of river runoff in the Arctic Ocean" (Ekwurzel et al. (2001)) and are widely used to detect the origin of water masses (Ekwurzel et al. (2001), Serreze et al. (2006), Bauch and Cherniavskaia (2018)). The simplest approach to detect a river water fraction in a water sample is to compute a ratio between the measured salinity and oxygen isotope 18 (delta-O18). As is described in Data and Methods Section, we used only the surface measurements in the upper 3-m layer.

Using a rather simple three-component model to distinguish the marine water, the river water (meteoric water), and the sea ice melt water described in Bauch and Cherniavskaia (2018), we calculated the fractions of each water mass (Fig. 14). In the work of Bauch and Cherniavskaia (2018), authors provide values of end-members of this model (typical salinity for each water mass and typical d-O18 concentrations), so after resolving a simple system of three linear equations using the values of the total (measured) salinity and the measured d-O18 concentration, we found a contribution of each fraction. As done in Bauch and Cherniavskaia (2018), the role of precipitation is neglected in this model, as its amount is insignificant compared to the river water input. The sea ice melt fraction can be negative in case of sea ice formation.

This analysis indicates that the most important fraction of river water is brought over the shelf and the shelf edge of the East-Siberian Sea (Fig.14, a). At the same time, the water samples at the northern part of the 126°E section consist of 10-15 % of the river water and only of 0-5% of the sea ice melt fraction. Knowing that the main maximum of the river discharge occurs in June (Fig.3), this fact supports our hypothesis that a noticeable amount of river water was distributed under the ice far northward into the deep part of the Laptev Sea (north of 80.5°N), where it will enter the central Arctic Basin later.

It is interesting that the areas with the highest sea ice melt fraction (Fig.14, b) (5-10%) very slowly follow the sea edge, so they were observed in the central and western part of the Laptev sea and in the MIZ area in the East-Siberian Sea. The sea ice formation (the negative values of sea ice melt fraction) is found in MIZ and its vicinity at 78-70°N - 150-160°E of the

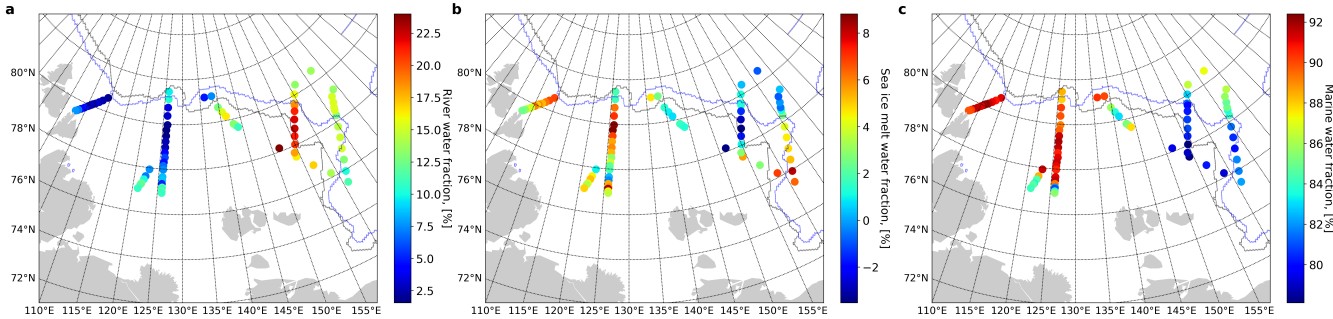

**Figure 14.** Fractions of river water (a), sea ice melt water (b) and marine water (c), calculated using d-O18 measurements and Bauch and Cherniavskaia (2018) 3-components model of freshwater balance. A thin black line shows the position of sea ice edge on August 31, 2018, when the northern stations of the meridional (5) section along 126°E were done in the MIZ, and the blue line shows the sea ice edge on September 16, 2020, when the ARKTIKA-2018 expedition was working in the MIZ of the East-Siberian Sea. Please, note that the colourbar scale is different for each water fraction.

East-Siberian Sea, which is expected as these measurements were done in the second part of September 2018, the beginning of freezing season. The presence of river waters may accelerate the sea ice formation if the air temperature favours it.

The surface water samples of the western and central parts of the Laptev sea consist of large marine water fraction (90-95%), Fig.14, c. The lowest marine water fraction (75-80%) was found over a very shallow ice-free area between the New-Siberian islands and MIZ in the East-Siberian Sea, where both sea ice melt and river water fractions are relatively high (5-10% and 10-25% respectively). Actually, it is the area of the most intense surface mixing that was observed using *in situ* measurements during the ARKTIKA-2018 expedition.

### 3.4 Wind forcing.

A previous study in this region (Dmitrenko et al. (2005)) claimed that the surface fronts displacement is mainly governed by the wind and atmospheric pressure centers. To investigate the role of the wind forcing at the synoptic scales, we compute mean monthly Ekman transport for August and September 2018 (Fig. 15). The calculation is described in the section 2.

The discussed displacement of the river plume extension in August and September is well seen in both SST and SSS mean monthly fields (Fig.15 a, c and b, d, respectively). The most pronounced feature in the SST field is the drop of SST by 3°C in the central and southern Laptev Sea. Salinification of the northern, central, and southwestern part is observed in August-September SSS fields. The average wind speeds are low to moderate during August and September, 3-7 m/s (Fig.15 e, f). The wind field in August is more homogeneous and velocities are slightly higher with an overall south-easterly direction; the Ekman transport pushes the river water out of the central part of the Laptev Sea favouring its propagation under the ice.

In September, the wind changes its main direction to south-westerly, which leads to river water trapped in the Yanskiy bay, still favouring the freshwater flux propagation under the ice, but mostly into the southern part of the East-Siberian Sea. As

it was shown in the work of Lentz and Helfrich (2002), the onshore Ekman transport will generate a coastal downwelling followed by an increase of the salinity over the far-field areas of the river plume and its further offshore entrainment.

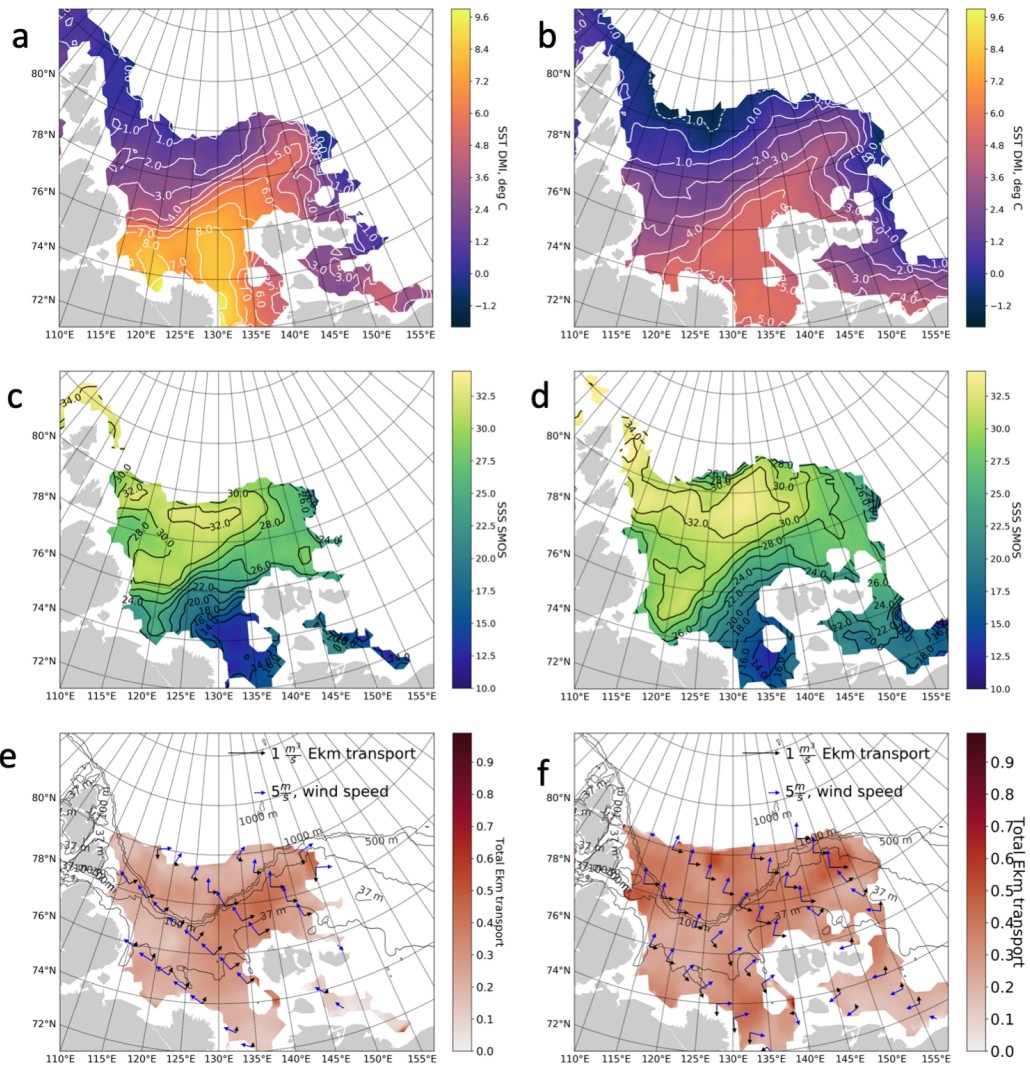

**Figure 15.** August (left column) and September (right column) monthly means of: (a, d) SST, (b, e) SSS, (c, f) wind speed and direction shown with blue arrows, horizontal Ekman transport is shown with black arrows, total horizontal Ekman transport is shown with colour

# 4 Discussion and conclusion

Based on *in situ* and satellite measurements, we document the evolution of water masses during August and September in the Laptev and the East-Siberian Seas. Satellite DMI SST and SMOS SSS "A" estimates are cross-validated with CTD data and continuous TSG measurements rarely done in this region. For the first time, thanks to the new satellite-derived salinity field
(SMOS SSS "A"), a vast range of *in situ* measurements and results of geochemical analysis, we follow how the river water input is distributed and where it is stored in the Laptev and the East-Siberian Seas at a synoptic scale.

To investigate local surface water masses, we use a variation of a classical TS analysis from satellite measurements. It helps to define new surface water masses adapted for the Eastern Arctic Ocean with a typically low salinity and discuss their transformation. As the validity of SMOS SSS was successfully demonstrated and as SMOS measurements are accessible from
2010 to the present, this technique could be applied for future studies of the surface water transformation on different time scales.

The transformation of fresh water mass from river inflow occurs very quickly during the Arctic summer, over a period of typically 1-2 weeks. Based on our observations, we suggest that wind-induced vertical mixing, a weaker river discharge, and a continuously decreasing radiative income impact the variability of surface water characteristics, as it is well observed at the
beginning of September 2018.

Following the classification of coastal buoyancy flows described in Lentz and Helfrich (2002), the displacement of river waters over the Laptev Sea shelf can be regarded as a slope-controlled buoyant gravity current, where the buoyancy forces are acting on the water body together with the wind stress and the bottom friction over a large area of the slightly inclined shelf, and at some moment the buoyancy flow detaches from the bottom at a distance $W_\alpha$. If we suppose that the equilibrium water depth
$h_p$, where the bottom friction becomes 0, is 25 m (based on section 5 southern CTD measurements), in the Laptev Sea it is situated at a distance $W_\alpha$ of 100 km from the Lena River delta, approximately at 75°N, 130°E. The upwelling-favourable winds will induce the offshore Ekman transport and stretch the buoyancy plume over the shelf, while the "downwelling" (onshore) winds will cause a deepening of the isopycnals. A sequence of upwelling-downwelling events can cause a further entrainment and a possible detachment of the buoyant coastal plume to the northern part of the shelf, the continental slope and the deepen
part of the Laptev Sea or over the East-Siberian Sea shelf.

The variability in freshwater and the energy sources therefore partly explains the seasonal variability of the buoyant plume. The first yearly maximum of river discharge occurs in May, after an opening of the sea ice. The second one occurs in the beginning of August. This warm and fresh river water is redistributed and transformed in the surface layer of the Laptev Sea during the month of August. After the passage of several cyclones in the beginning of September, there is no additional source
of heat and fresh water that would maintain the variability of water masses. Overall, in September, the "cold and saline" water mass progressively occupies the ice-free surface of the Laptev Sea instead of other ("transformed") water masses observed there in August. Sea ice growth starts in the end of September whereas sea surface will be fully covered by sea ice only by November 2018. The autumn freezing begins only after the heat accumulated during the summer season is released to the atmosphere, and the water temperature at the surface drops to the freezing point.

An overview of Horner-Devine et al. (2015) describes main mechanisms of the river plume mixing and transport at different distances from the estuary area. Our *in situ* and satellite data make it possible to investigate mostly the "far field" area of the river plume, where the balance between the wind stress and the buoyancy is governing the propagation and mixing of the surface fresh water with the "ambient sea". As it was mentioned, the Ekman transport illustrates an important forcing for the

freshwater displacement. As the theoretical Ekman depth is controlled mainly by the Coriolis parameter $f$ and by viscosity, assuming that the viscosity is homogeneous in the south of the Laptev and the East-Siberian Seas, the Ekman depth exceeds the depth of some shallowest areas (according to Baumann et al. (2018) study in the same region the Ekman depth is $D_{Ekm} = 37$ m: see the position of 37-m isobath in the Laptev Sea in Fig.15). Thus, the calculated Ekman transport should be regarded as a theoretical concept illustrating possible mechanisms of horizontal transport and vertical mixing only in the central and

northern areas, whereas over the shallower seas, the dynamics will be more constrained by mixing and the direction of the Ekman currents relative to bathymetric contours.

During the first 10 days of August, the upwelling-favorable north winds (mostly anticyclonic atmospheric conditions) stretched the freshwater plume from the south to the central Laptev Sea up to 78°N. In the second part of August, the wind changed its direction to the westward and (downwelling favourable), so that a buoyancy plume of river waters displaced paral-

lel to the coastline to the East-Siberian Sea. An entrainment of the buoyancy plume is evidenced in the CTD-measurements of salinity in the Eastern Laptev Sea and in the East-Siberian Sea (sections 6-8).

The wind situation in August was favorable for the extreme propagation of river water into the north-eastern part of the Laptev Sea followed by a propagation into the East-Siberian Sea in the MIZ and under the sea ice. The propagation of river water under the sea ice is apparent in the western part of the East-Siberian Sea, where two branches with warm and fresh cores

have been observed with *in situ* data (section 6). These branches can be understood as a result of previous entrainments of the Lena River water after a sequence of upwelling-downwelling events, pushing and detaching them offshore. After the river water separated from the estuary, it followed the shelf keeping its freshwater core with salinity below 29 up to the East-Siberian Sea.

A pathway of the low salinity Kara water is observed during several days of our study. The Kara water propagates mostly

through the Vilkitskiy Strait and partly through the Shokalskiy Strait, but no freshwater is found northward of the Severnaya Zemlya Archipelago. Propagating along the coastline (as a reminiscent of a Kelvin wave starting from the Ob' and Yenisey estuary), the low-salinity Kara water enters the Oleneksiy Bay where it meets another freshwater flux from the Khatanga and the Lena rivers. The arrival of freshwater via the Vilkitsiy Strait was already studied using *in situ* data by Janout et al. (2015) and Janout et al. (2017). However, this is the first time this event has been observed from satellite salinity data which provides

a unique opportunity for a regular monitoring. The freshwater input from the glaciers and icebergs of the Severnaya Zemlya Archipelago should probably also be taken into account, but this is out of the scope of this study, and we have assumed the volume of this source of freshwater to be negligible compared to the volume of the incoming very fresh Kara Sea water.

At a larger scale, the observed spatial variability can also be explained by a positive (in April-October 2018) Arctic Oscillation index favouring the eastward propagation of fresh water, as it was demonstrated in studies by Morison et al. (2012),

Armitage et al. (2016), and Armitage et al. (2018). An important part of the northward propagation to the shelf edge is not

explained by the positive AO. Based on the oxygen isotopes results, we claim that a similar propagation of river water far northward happened before the observed period (in June-July), when the Laptev Sea was still covered with ice and the Lena River discharge was at its largest (Fig.3). At the northern end of the 126°E section, under the ice, the upper 25-m layer is fresher with a salinity below 33, which supports this hypothesis. There is no evidence that the sea ice melting itself created

such a considerable layer of freshwater. Our isotopes estimates could be further refined using alkalinity to separate the meteoric water estimated with water isotopic analysis (river input from precipitation). A study of Bauch and Cherniavskaia (2018) also supports the influence of river water, as a similar situation was observed in 2011. Unfortunately, the present spatial resolution of satellite-derived SSS and its uncertainty due to the proximity of sea ice makes it difficult to separate river water from the freshening associated with sea ice melt. No accurate satellite measurement of sea ice thickness in the MIZ is available at present

to the best of our knowledge, hence it is not easy to evaluate the freshwater input due to the sea ice melting on the scale of several months. Nevertheless, the existing satellite data have already a great potential for the Arctic studies of fresh water. To improve this evaluation of the freshwater budget in the Arctic Ocean, we suggest that appropriate numerical models assimilate the estimates of river discharge, new satellite-derived sea surface salinity and wind data. An attempt to analyze the new sea level dataset provided by DTU (Danish Technological University) for the Arctic Ocean (Andersen and Knudsen (2009)), and

calculated geostrophic currents, is discussed in the Appendix and enriches the overview of the surface ocean dynamics during selected summer months. Nevertheless, at present it seems that an additional work on the continuous ADT estimates over the shallow water is needed to use this dataset for the studies of coastal flow variability.

**Appendix A:  Altimetry and geostrophic currents**

We calculated two monthly fields of absolute dynamic topography (ADT) and geostrophic currents from sea level anomalies

(SLA) Arctic L4 product and mean dynamic topography (MDT) provided by the Danish Technological University (Fig. A1, Andersen and Knudsen (2009)). Sea level anomalies are available as mean monthly values on a grid adapted to polar regions with 0.25°step for latitudes, and 0.5°step for longitudes. Mean dynamic topography global model with one-minute spatial resolution was used to compute ADT. The resulting monthly absolute dynamic topography ($ADT = MDT + SLA$) was calculated for selected summer months.

25       Overall, the ADT remarkably follows the ocean bottom topography with higher SLA over the shelf and lower SLA over the deep part of the studied area, which corresponds to the study of Armitage et al. (2016). The only exception is negative SLA in the Olenekskiy Bay in August 2018. We suggest that the general northward wind-induced displacement of the water over the very shallow southern part of the sea was compensated only by the river water inflow to the east of 122°E, close to the Lena River delta. Positive SLA were more pronounced in September than in August, though in August the SST was higher over the

southern and central parts of the Laptev Sea, and the salinity was lower in the Olenekskiy Bay. The importance of the steric component in variation of the sea level in August-September is thus doubtful, though several sources of uncertainty can impact the quality of these SLA data: uncertainties in the tidal model, the bathymetry precision, the accuracy of the MDT over the shallow part of the Laptev Sea, etc, as Armitage et al. (2016) noticed in their work. It should be noted that SSH variability of

the Laptev and the East-Siberian Seas presented in the work of Armitage et al. (2016) had the lowest correlation with *in situ* gauge measurements in the Arctic Ocean, because of the "seasonal runoff".

The geostrophic currents were calculated following: $u_g = \frac{-g}{f}\frac{dh}{dy}$, $v_g = \frac{g}{f}\frac{dh}{dx}$, where $h$ is ADT, $x$ and $y$ are the distance in metres. Geostrophic currents presented in Fig.A1 are very weak and demonstrate rather chaotic structures during the summer months of 2018. Among the well-pronounced features, we find an outflow from the Laptev Sea in the Vilkitskiy Strait. Above the continental slope edge, the principal direction of currents is westward with a maximum current speed of 0.5 m/s. Further south, an outflow at 122 °E and 130°E contributes to transport the Lena River Water into the central part of the Laptev Sea. In the Yanskiy Bay a vortex-like system exists in both August and September 2018. Geostrophic currents in the East-Siberian Sea were calculated from the altimetric measurements in MIZ, so should be interpreted with care. Armitage et al. (2017) stated that the SSH measurements there cannot reflect the mesoscale phenomena, because of the small Rossby radius (of order 1 km) and the altimeter along-track resolution of 300 m. At the same time, the same study reported the largest eddy kinetic energy in the shallowest areas.

From our calculations, a cyclonic feature of 150 km in diameter is seen at 79°N, 157°E, and might be topographically induced, as well as a similar cyclonic feature at 78.5°N 135°E. An extended study should be carried out to validate the accuracy of altimetry-derived currents in this region with mooring or vessel mounted ADCP measurements.

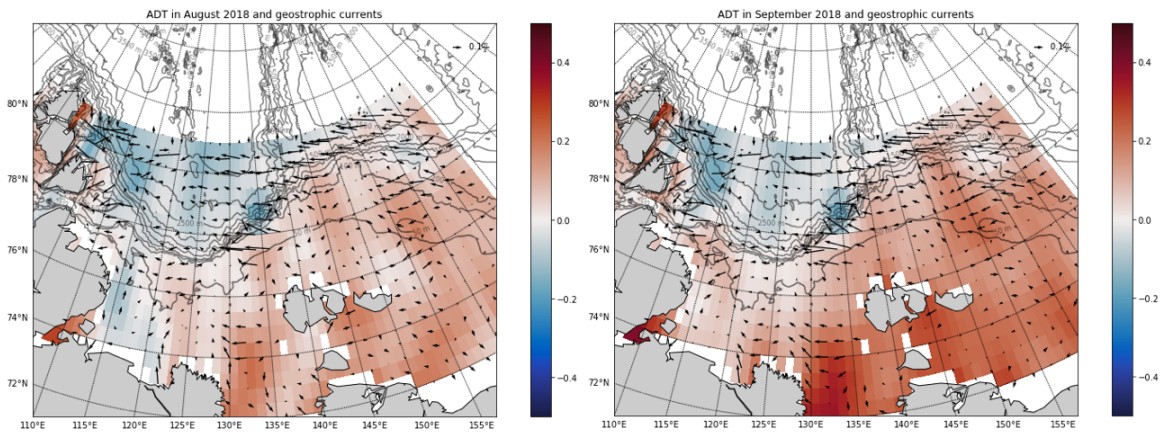

**Figure A1.** Absolute dynamic topography (cm) and geostrophic currents (cm/s) in August and September 2018, calculated from DTU monthly Sea Level anomaly

*Competing interests.* No competing interests are present

*Acknowledgements.* We thank Jean-Luc Vergely for fruitful discussions about SMOS SSS data filtering in the Arctic ocean. We thank Mattew Alkire, Andrey Novikhin, Natalia Vyazigina, all hydrochemistry team of the ARKTIKA-2018 expedition and Ekaterina Chernyavskaya for the collection of water samples for the oxygen isotopes analysis, their analysis and further discussion. We thank all the scientific team of the ARKTIKA-2018 expedition and the crew of RV Akademic Tryoshnikov for their work. This study was supported by the French CNES-TOSCA SMOS-OCEAN project. Anastasiia Tarasenko, Vladimir Ivanov and Nikita Kusse-Tiuz acknowledge financial support from the Ministry of Science and Higher Education of the Russian Federation, project RFMEFI61617X0076.

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
