# Peer review of "Properties of surface water masses in the Laptev and the East-Siberian Seas in summer 2018 from in situ and satellite data"

_Ocean Science, 2019_

## Short Comment (SC1) · 23 Oct 2019

Review of "Surface waters properties in the Laptev and the East-Siberian Seas in summer 2018 from in situ and satellite data" by Anastasiia Tarasenko et al. Submitted to: Ocean Science Manuscript number: 2019-60

Summary: In this paper the authors focus on distribution of surface water masses based on satellite salinity and temperature and in situ measurements at the shelf of the Laptev and East-Siberian seas. They report several features registered by satellite and in situ data in the surface layer, namely, variability of surface salinity and temperature in August-September 2018, inflow of freshwater from the Kara Sea to the Laptev Sea,

seasonal cooling of surface water in autumn. The paper is interesting insofar as the authors focus on the Arctic seas which hydrological structure and dynamics remain largely unstudied. Thus, the topic addressed in this manuscript is of great scientific and practical interest. Despite my enthusiasm for the topic, I don't think this article is ready to be published in Ocean Science due to several drawbacks of this work. Generally, this article seems like a cruise report, it describes structure of SSS and SST in the Kara Sea to the Laptev Sea, but lacks scientific novelty and new insights into processes in the study area. I recommend the authors to improve their study by providing more thorough analysis of in situ data including vertical profiles and to focus on certain processes that occur at the shelf of the Laptev and East-Siberian seas, rather that providing brief description of multiple processes.

General comments: 1. The authors define the plume as water mass with salinity less than 30 (e.g., page 10, line 28). However, the majority of works that deal with river plumes in different World coastal areas define river plumes as relatively shallow surface-advected water masses bounded with large salinity gradient at their border with ambient sea. Existence of this salinity gradient determines significantly different dynamics of river plumes (governed by buoyancy force), as compared to ambient sea, which is the main reason to distinguish river plumes as individual water masses. River plumes formed at the shelf of the Laptev and East-Siberian seas generally have sharp salinity gradient at isohalines of 15-25, while water masses with greater salinity are regarded as ambient shelf water. Thus, I recommend the authors to determine salinity border of the Lena plume based on maximal salinity gradient and to distinguish winddriven dynamics and variability of river plumes and more "typical ocean dynamics" of shelf water mass. 2. In this study you use SSS data from SMOS satellite which spatial resolution is 50 km (page 5, line 18). However, you deal with salinity maps with 15 km spatial resolution (page 5, line 18). Did you reduce spatial resolution only by reprojection? 3. In this study you deal with in situ thermohaline data obtained from the depth of 6.5 m (page 7, line 3). However, salinity at this depth can be significantly different from surface salinity (even more than several units) especially within the river

plumes. Thus, your usage of this data to compare and validate satellite data require additional proof, e.g., based on vertical thermohaline measurements. 4. The results of validation of satellite SSS and in situ salinity obtained from the depth of 6.5 m does not seem convincing, especially at the areas influenced by freshwater discharge (Section 3.1.2 and Figure 3). We see underestimation of salinity by several units for almost all measurements. I recommend authors to deal with salinity gradients rather than absolute salinity values, e.g., to show that satellite SSS data reproduces well shows relative salinity differences if it really does. 5. Ranges of temperature and salinity values used to determine different water masses at the study area are heuristic and are not based on any precise idea (Section 5 and Table 1). What is the reason to select T = 3 °C and S = 25 and 29 as borders between water masses? Why you determine 6 water masses? Why not to determine 5 or 7? 6. Reconstructed circulation in the Laptev and East-Siberian sea based on Ekman theory does not seem convincing, especially the presented patchy distribution of upwelling and downwelling areas (Section 4.1.2 and Figure 9e). These results have to be supported by in situ measurements and/or numerical modelling. 7. Propagation of freshened water from the Kara Sea and its presumed missing with Lena plume in the Olenekskiy bay requires additional proof by in situ measurements and/or numerical modelling (page 19, lines 6-9). The role of the Khatanga plume in this process (as well as the plume of the Olenyok River surprisingly not mentioned here) also should be supported by additional data.

---

## Referee Comment (RC1) · Anonymous Referee #1 · 26 Nov 2019

Overview.

This manuscript uses a variety of resources, including in situ measurements, satellite measurements and reanalysis products, to describe the evolution of the Laptev and East Siberian Seas in late summer of 2018. The manuscript begins with a correct statement: that the region of interest is understudied. I think that the manuscript should be publishable, but only after significant attention has been directed (1) at the introduction (for context) and conclusions (for the meaning of the results in context), and (2) at Ekman fluxes and Ekman pumping. For the Ekman point, I believe the calculations to be invalid in shallow and coastal waters; see specific point 10 below. More on point (1)

[Figure]

follows, because at present, I think that the manuscript reads more like a report than a paper.

The manuscript introduction has some weaknesses. It tells the reader some of what happens in the region, but it does not really say why the region is important, why it matters. If I were making a case as to why a reader should care about the region, I would be looking at freshwater fluxes, specifically the role of the Siberian shelves in modulating the inflow to the Arctic Ocean of the great Siberian rivers, either those that discharge directly into the Laptev and East Siberian Seas, or those west of Severnaya Zemlya, some of whose freshwater runoff enters the region through the Vilkitskiy Strait. The most recent (and very good) review of such issues is the Eddy Carmack paper (JGR 2016).

Also, if the region is understudied (and I agree that it is) then the introduction needs to describe whatever of importance has been published on the region. Item 4 below gives three references led by Tom Armitage. They are pan-Arctic remote sensing papers that have quite a bit to say about the Siberian Shelves, and they are not mentioned. Plus I could add Johnson & Polyakov (GRL 2001), Semiletov et al. (GRL 2005), Lenn et al. (GRL 2009 and JPO 2011), and the Lenn papers remind me that the manuscript talks about currents but does not mention tides, which are important in the shelf seas. There may be more papers, these are just a few that come to mind. I am not confident that the authors have thoroughly reviewed the literature.

Then concerning the final section called Discussion and conclusions; this section is really not much more than a brief restatement of the work reported in the preceding sections. If the introduction does not tell the reader why the region is important, then the conclusions cannot then tell the reader why the new results matter.

Specific comments are in order of occurrence.

1. The first paragraph on pp 1-2 describes the region. Reference should be added to the map of Figure 1; all locations in the text should be labelled, so please add text

"Severnaya Zemlya" to the map. Later you refer to Arkticheskiy Cape, add this as well.

2. In the same first paragraph, it is correctly stated that the region is little-studied, but I think there should be a sentence stating why the authors think the region is important. At the end of para. 1 on the top of p 2, add a sentence "The region is important because ...".

3. When talking about salinity, it is not appropriate to use "PSS". If using the new Absolute Salinity, then you can say parts per thousand, ppt, or use the "per mille" symbol. If not, then just "salinity of nn.n" or "nn.n in salinity" is correct.

4. Recent papers led by Armitage using Envisat, Cryosat and GRACE between 2003-2014 seem not to have been looked at: JGR 2016 is about sea surface height variability, with discussion of Siberian shelf seas; Cryosphere 2017 is about surface geostrophic circulation; GRL 2018 is about sea level & surface circulation response to the Arctic Oscillation. Some the material in these papers is directly relevant, and this omission should be corrected.

5. All figures with multiple panels, please label them a, b, c, etc. and refer to the panels as such in the manuscript text. All captions must state all plotted quantities and their units.

6. Section 3.1.2 on salinity, and cf SMOS text in 2.2.2. I doubt that the spatial resolution is as high as it appears in Figure 3. I understood SMOS to resolve at about 100 km, in 2.2.2 the authors mention sampling at 15 km resolution, but adjacent points are surely not independent. How does this affect their statistics?

7. Section 4 first & second lines, your temperature accuracy cannot be 1 m$^\circ$C, so why are you quoting temperature values to three decimal places?

8. Figures 5 and 7, Hovmoller plots: you are inconsistent. Figure 5, longitude (x-axis), days (y-axis); figure 7, vice-versa. Pick one orientation and stick to it.

9. Figure 8, temperature sections. Improve your presentation, please. Viewing the

PDF, about half of the figure is just black and any temperature structure is obscured. The simplest solution would be to plot contours in white – not black!

10. Section 4.1.2. Please improve your terminology. "Transport" is typically a volume transport, units mˆ3/s. In your eq. 1, stress (N/m2) divided by [ density (kg/m3) * Coriolis (sˆ-1) ] has units m2/s. These units appear in tiny (almost unreadable) notation in figure 9. This is neither a velocity nor a transport. Importantly, though: are your Ekman calculations valid in shallow water? Ekman's assumptions included (1) no boundaries (remote from coasts), and (2) deep water (typically >200 m). What is the Ekman layer depth? Is it not more likely that upwelling / downwelling are dominated by sea surface height changes in shallow water? For instance, a wind from the west will cause surface water movement to the right (the south) in the northern hemisphere. Water "piles up" against the coast, and that induces downwelling (at the coast): see papers by Steven Lentz, for example. Is this in accord with your calculated vertical velocities in figure 9? I would say not.

———————————————

---

## Referee Comment (RC2) · Anonymous Referee #2 · 4 Feb 2020

The submitted manuscript "Surface waters properties in the Laptev and the East-Siberian Seas in summer 2018 from in situ and satellite data" by Tarasenko et al., presents and validates DMI SST L4 and SMOS SSS products (including errors) in a novel way against in situ temperature and salinity data (CTD profiles from the upper 6.5 m and TSG data at 6.5 m depth) obtained during the ARKTIKA-2018 expedition in the Laptev Sea and the East–Siberian Sea in August and September 2018. Since this region is highly under sampled, this study is of high relevance for the scientific community focusing on air-ice-ocean interaction processes in the Arctic Ocean. In the study, they follow the north-eastward spreading of warm and fresh river water from the Lena River Delta towards the central Laptev Sea and further towards and under the sea ice

in the East-Siberian Sea. Also, river water inflow from the Kara Sea to the Laptev Sea is followed. By combining SST and SSS data products with satellite-derived products of sea ice concentration, wind speed (and direction), and ERA5 reanalysis fields of sea level pressure, they investigate wind-induced horizontal displacements of the river water plume due to Ekman transport and change in the strength of the stratification across different shelf-slope transects due to Ekman pumping (downwelling/upwelling). Based on the satellite-derived SST and SSS data, they also suggest different types of surface water masses for the area, and further present their distribution and presence during the studied period.

Even though this study is of high relevance, the submitted manuscript needs a major revision before suited for publication in Ocean Science. First of all, the English language needs to be improved and corrected. There are simply too many grammatical and syntax mistakes and typos for me to point out, so I will only mention some of the most general mistakes below. Secondly, I spent too many hours reading the manuscript due to an unfinished appearance with in some places lack of background information or a logical flow. Hence, I found it a bit hard to get the main objectives, the main results, and the main conclusions from the study. More specific comments follow below.

General on language:

Use definite (the) or indefinite (a/an) articles where suitable.

Remember to use capitol letter S in named seas: Laptev Sea, Kara Sea, East-Siberian Sea etc. This is not used consistently in the text. Same with named rivers, use capitol letter R (Lena River etc.) Same with named archipelagos, use capitol letter A (Severnaya Zemlya Archipelago etc.) Write the Arctic Ocean with capitol letter O. Use capitol letter W in named water masses: Atlantic Water, Arctic Water, etc.

Be consistent with use of present (is) or past (was). Use past and not present when referring to others work.

[Figure]

Avoid plural in surface water, river water, etc. if you don't have several named surface or river waters.

"Ice free areas" instead of "free of ice areas"?

In the north/south/east/west, not on the north/south/east/west.

Practical salinity scale (PSS) or practical salinity unit (PSU) is termed salinity with no unit, while absolute salinity is termed absolute salinity with unit g/kg.

Title

Perhaps change the title to "Properties of surface water masses . . ." To use surface water in plural seems a bit strange to me. You are suggesting six different water masses in the surface, not several surface waters, or?

Introduction

Generally, the introduction lacks objectives and proper background information of the study area. What is known in this region already, what studies have been done in this under-sampled area of the Arctic Ocean, what about river discharges into the study area (the river water plume is the main research focus of the study), and why is this area important?

Page 1, Line 19: "In the Arctic region, a strong seasonal warming and cooling with sea ice melting and freezing modify . . ."

Page 2, Line 5: ". . . the upper ocean water displacement . . ." Not plural.

Page 2, Line 9: " via the energy vertical distribution . . ." What energy?

Page 2, Lines 14-19: Rewrite this paragraph and improve language, for instance:

Line 14: Rewrite to i.e. ". . . proposes to use the term "surface layer" instead of "mixed layer" for the Arctic Ocean, because the water . . ."

Line 15: What is meant with "the water horizon lying between the sea surface and the

Arctic main halocline"? Use another word than horizon?

Line 17: "m depth in the Eastern Arctic Ocean (Dmitrenko et al., 2012) and at 100-200 m depth in the Western Arctic Ocean . . ."

Page 2, Line 23: Rewrite to i.e. ". . . the saline waters, which was considered the mean Arctic Ocean salinity at that time."

Page 2, Line 25: Reorder sentence to "Cherniavskaia et al. (2018) reported an overall salinity in the upper 5-50 m layer to lie within the range from 30.8 to 33 based on in situ data in the Laptev Sea during 1950-1993 and 2007-2012.

Page 2, Line 31: Remember unit on density (kg m-3)

Page 3, Line 8: Define L-band satellites.

Data and Methods

All information regarding the data and methods (presentation of the in situ hydrographic sections, processing and analysis methods) that are described and included in the results section should be moved to the data and method section. More details are given later. It would also be useful to include a link or doi to the downloaded data products (DMI SST L4, SMOS SSS L2 (you are making weekly averages of these, are you using any other SSS products?), AMSR2 SIC (and specify all SIC products used), ASCAT C-2015 L3 for wind speed and direction, ERA5 reanalysis for SLP and air temperature). Several data products are mentioned and utilised in this study, and to better clarify which ones that are actually used, it would be good to start the description of the used data products, why these ones are chosen, and what has been done with the data products in this study (post-processing). Then, other alternative products used in other studies can be mentioned afterwards for comparison? This goes especially for Section 2.2.2, which now appears a bit messy with a lot of mixed information about others products and work and the products used in this study.

Page 3, Lines 19-21: Put all place names on the map in Figure 1.

Page 3, Line 27: Did you find a typical error estimate from the comparison with CTD measurements?

Page 4, Line 2: What interpolation methods were used?

Page 4, Line 6: Define AVHRR, MODIS and VIIRS

Page 4, Line 8: Define AMSR2

Page 4, Line 11: Your write "(hereafter referred to as "SST DMI")", then make sure you do. Several names are used on this product later in the manuscript (DMI SST L4, the SST field from DMI L4 product, blended DMI product?, temperature estimates provided by DMI, and others).

Page 4, Line 11: Define Level 4.

Page 5, Line 7: You write "100-km averaged ship SSS . . ." Is this TSG?

Page 5, Lines 12-13: Define SMAP CAP/JLP and SMOS BEC L3, REMSS

Page 5, Line 16: Define ESA.

Page 5, Line 17: Be consistent with naming of products. Here you use L2 SSS and not SMOS SSS L2.

Page 5, Line 18: What do you mean with individual SMOS SSS? Explain how the SMOS SSS are sampled over an Icosahedral Snyder Equal Area grid at 15 km resolution. Is this the interpolation?

Page 5, Line 23: Define ECMWF.

Page 5, Lines 26-29: Specify the correction with numbers. Also, give the criterion on the ACARD parameter. And what are the typical errors in the weekly SMOS SSS and the individual SMOS SSS?

Page 5, Lines 32-34: Where is the river plume? Define and introduce it. How does this affect the weekly SSS error?

Page 6, Line 4: Define all the ice masks provided.

Page 6, Line 10: Define ASCAT.

Results

Move all method and technical descriptions to Data and Methods. Only results should be presented here. The results section should have more focus on the SST and SSS distribution, the hydrographic sections, and highlight results to be discussed, not so much on technical details, error analysis and discussion, which I recommend you to move to the data and method section. In general when presenting and discussing the results, please be consistent and distinguishing between the parameters SST/SSS and in situ temperature/salinity. Don't use temperature or salinity on both.

Page 6, Line 22: Please define the river waters. Use plural form if there are several types of river water, it not, use singular form (river water).

Page 6, Lines 24-25: Move to Data and Method.

Page 7, Lines 1-7: This paragraph belongs in data and methods.

Page 7, Line 2: What are "basic statistics"?

Page 7, Lines 4-5 & 8-9: How much does the temperature and salinity change in the upper meters? Perhaps show a mean +/- std profile of the upper 10 meters?

Page 7, Lines 10-11: And how is it in ice free conditions?

Page 7, Line 15: In-situ surface layer temperature is then the upper 6.5 m (but not including the uppermost 2? meters typically)?

Page 7, Lines 17-19: This is a complicated sentence, so please rewrite. Can this be shown?

Page 7, Line 21: What grid? Please describe in data and method, and refer to it.

Page 7, Lines 28-29: I guess the 15 km SMOS resolution is the interpolated resolution?

Please be more precise when naming the different SMOS SSS products, which should all be clearly defined and stated in the data and method section.

Page 7, Line 31: Use either ARKTIKA-2018 expedition or Akademik Tryoshnikov measurements. Be consistent with the naming of data both in the text and in the figures.

Page 7, Line 33: Again, be consistent with naming of data products, I guess the vessel SSS is the CTD and TSG data from the upper 6.5 m from the ARKTIKA-2018 expedition?

Page 7, Lines 33-34: This belongs in the method description.

Page 8, Line 1: "SMOS post-processed SSS", what is this product compared to the other named products?

Page 9, Lines 2-3: Is the precision so high that you can use three decimals in SST?

Page 9, Lines 2-4: Put place names on the map in Figure 1.

Page 9, Lines 9-10: Please define river water somewhere (in data and method?). What is the definition of the river water plume?

Page 9, Line 10: Standard deviation of what?

Page 10, Line 1: What about the Katanga River Estuary?

Page 10, Line 2: Define the thermal fronts.

Page 10, Line 15: "at 125 E, ..." what latitude?

Page 10, Line 20: Distribution of freshwater input, is that water with S = 0? Please define the characteristics of this freshwater.

Page 10, Line 21: Please refer to Figure 1. What Section number or longitude/latitude limits?

Page 10, Line 21: Temporal evolution of what? Introduce Figure 6 before Figure 7.

Page 10, Line 25: "shelf break" instead of "edge of the shelf"?

Page 10, Line 29: What was the wind direction during these series of cyclone passages?

Page 10, Line 30: Change to ". . . salinity increased by 1 . . ."

Page 10, Line 34: Change to ". . . mixing event induced by the wind."

Page 10, Lines 30-34: How does this relate to any river discharge data from the same period? Are there any model data to compare with?

Section 4.1.1

Change to something like "Water from the Kara Sea". You have not defined Kara Sea waters anywhere in the text.

Page 11, Lines 6-7: The temperature is decreasing?

Page 11, Line 7: Where and when has the exchange with the atmosphere taken place? Is the atmosphere colder here than in the central Laptev Sea?

Page 13, Line 1: You write ". . . significantly greater than . . ." How can you see relative amount of freshwater from one snap-shot and with different timing on each section occupation?

Page 13, Lines 2-5: Show place names on map in Figure 1. Also, define the hydrographic sections in data and methods with time, wind condition, and air temperature during their occupations. Then stick to the named Sections (Section numbers) later in the text.

Page 13, Line 7: 1024-1027 kg m-3?

Page 13, Line 14: What happens during cyclone passages? Clouds? Have you discussed this effect in Data and Methods? If yes, refer to it.

Page 13, Line 15: Can you provide any T and S limits?

Page 13, Line 17: By depression passage you mean low-pressure or cyclone passage?

Page 13, Lines 18-19 & Page 14 Lines 1-4: What about wind direction? Wind speed alone will not give the complete forcing pattern. Perhaps make some cross-correlation maps between wind speed/direction and SLP with SST and SSS? SLP and wind direction should be highly correlated.

Page 14, Lines 5-7: This belongs in Data and Methods.

Page 14, Line 9: Absolute salinity and conservative temperature are shown in Figure 8. Either change the text to absolute salinity and unit g/kg or change to practical salinity in figure.

Page 14, Line 11: You refer to different observations at specific latitudes in the figure, but the figure is presented with distance in km, so please refer to km as well. This applies for the other sections as well.

Page 14, Lines 12-16: What about melting under the sea ice due to the presence of warmer (above freezing point temperature) river water? This will cool the river water and still keep the water under the ice relatively fresh. Are there any $\delta 18O$ data (or other tracers) to be able to identify the source of this water?

Page 14, Line 15: You write: "The heat exchange with the sea ice might be more effective than with the atmosphere, ..." Why?

Page 14, Line 19: "Below the pycnoclines, ..." to what depth?

Page 15, Line 1: The 34.5 isohaline is not shown in the figure.

Page 15, Lines 2-3: What are the T and S characteristics of AW is this region? I don't agree that AW is seen at 100-120 m depth in Figure 8? Why is it instability in the AW layer?

Page 15, Line 5: Define all hydrographic sections in Data and Method (see earlier comment on this).
Page 15: Please remember to refer to figures.

Page 15, Line 11: You write: "... which is clearly seen in temperature signal that is negative even close to the surface." Is the temperature still above the freezing point, i.e. any melting under the ice?

Page 15, Lines 13-14: What kind of mixing?

Page 15, Line 19: How can you see an efficient mixing from these data?

Page 15, Line 20: Are the temperatures still above freezing?

Page 15, Lines 25-26: The river discharge information should be introduced in the introduction.

Page 15, Lines 27-28: Was the atmosphere colder or warmer than the river water or the mixed upper layer water? What about melting of sea ice from below? Is the river water warmer than the freezing point temperature?

Section 4.1.2

Some more background information is needed in the beginning of this section. It is also an analysis method and should be moved to the data and method section. The surface fronts in question should be defined. To get the Ekman transport, you need to integrate over the Ekman depth, what is the Ekman depth in this region? Assumptions made by Ekman were no boundaries, infinitely deep water, and no geostrophic flow. How are these assumptions met in this region? What happens when you have boundaries (coastlines)?

Page 16, Line 8: Give the reference to TEOS-2010 and show the formula for the drag coefficient CD.

Page 16, Line 10: Use Ekman pumping instead of vertical Ekman speed?

Page 17, Line 12: You write "... mixing of different water masses, ..." What are these

water masses? You should introduce the water masses in the region in the introduction section.

Section 5

Perhaps start the results with this section?

Page 17, Line 17: And below 200 m?

Page 17, Line 23: Specify which of the Arctic Ocean waters that quickly change.

Page 17 Line 24: Add "..., and the synoptic satellite data provide ..."

Page 17, Line 32: The "classical" water masses, do they have a name?

Page 18: Refer to figures.

Page 18, Line 1: Add "... larger range than the near-surface (upper 6.5 m) in situ measurements ..."

Page 18, Line 7: A recognisable river front or a river plume front? Define.

Page 18; Line 8: 145 E? How do you define the sea edge? Is it a specific sea ice concentration limit?

Page 18, Line 11: You write "... captured under the ice and exposed back." How or when? What about melting sea ice from below?

Page 18, Lines 21-22: Show place names on map in Figure 1.

Page 18, Lines 24-25: There is also a high density within the range 22.5-30 and 3-4 °C.

Discussion and conclusion

This section provides little discussion or conclusions, mostly summary. There is more discussion in between the presentation of the results in the Results section.

Page 19, Lines 11-12: You write "The fresh waters displacement was associated with the Ekman transport." How? This is not well presented or described in the results.

Page 19, Lines 15-16: You write ". . . and there is no evidence that the sea ice melting itself can create such a considerable layer of freshwater." How or where have you presented this lack of evidence?

Page 19, Lines 18-20: What about river water melting sea ice from below? Are there any $\delta$18O data (or other tracers) to evaluate this?

Page 19, Line 24: Add "Calculated monthly Ekman pumping indicates the area of most intense mixing processes induced by wind."?

Page 19, Lines 25-26: Why is this included? This is not used in the discussion. Are there any conclusions from this?

Appendix

Page 20, Lines 7-10: What about Ekman transport to the south in September piling up water on the shelf and toward the coast in the south? This might induce downwelling (see Figure 9).

Figures

Figures with more than one panel should be labelled with a), b), c) etc. and then properly referred to in the figure captions and the text when the figures are described. Also use same font size in the figures with several panels. Remove titles on figures and add this text in the figure caption instead.

Figure 1: What is the definition of the sea ice edges? Add all place names mentioned in the text on the map.

Figure 2 and 3: Last sentences in the figure captions: this information should be provided in the data and method section.

Figure 4: Missing numbers on y-axis in lower left panel. Use white isolines below 0 degree Celsius/28 PSS? It is hard to see black lines on dark background. This is also the case for Figures 6 and 8.

Figures 5 and 7: Where is the data from? What bathymetry data is used?

Figure 8: Why do you show conservative temperature and absolute salinity in this figure and not in

Figure 6? At least use proper units when referring to Figure 8 in the text.

Figure 9: Increase the fonts. The two upper rows are not described in the text. What are the arrows in the lowest row, and why is the resolution higher than in the row above? Write units in the figure caption. Where is the data from?

Figure 10: Add the freezing point temperature line. Remove the figure title.

Figure 11: What are the boxes? Are the lines the freezing point temperature line?

Figure 12: Put the dates inside the plots instead.

Figure A1: Remove the titles from the figures. Write units in the figure caption. Increase the fonts.

Tables

Table 1: Add more information to the table caption.

---

## Referee Comment (RC3) · Anonymous Referee #3 · 10 Feb 2020

The presented observations of surface freshwater distribution is from an interesting area of the Arctic Ocean. The observations are presented in a nice manner. And there is plenty of figures, with many detailed results. This is all fine.

There is not much available knowledge on how the river-water spreads north along the shallow Siberian shelves, so this paper is potentially a significant contribution in that regard. Beside some issues already noted by the other reviewers, like language, I have two larger issue that made me tick the "major" box here.

My "major" science concern is the contribution from sea-ice melt. I think you need to do a somewhat better job at addressing this possible contribution. It is difficult, but it

can potentially explain much of the freshwater available. If delta-18 O samples where available, one could differ between these to sources perhaps, although I think much of the river-water from previous summers probably freeze-up, and some of the newly melted sea ice could thus just be old river water.

I also find that you mix Results and Discussion in the 3. Results section. This may be OK, but then you need to call this 3. Results and Discussion. And then the sections 4.X are somewhat also Results and discussion. And then you cannot have "Discussion and Conclusion" in section 6. See? A re-struction is needed. There is also no general conclusion drawn on what actually spreads the freshwater to the north. Is this wind driven mostly? Or not?

I also have a general suggestions: Provide the spatial mean vertical profile down to 100m for T and S, and use that to describe the mean stratification. THEN – AFTER-WARDS, you can present, the spatial and temporal changes from this mean profile.

Minor issues:

Abstract: Your main explanation for how the river water is transported out from the river mouth area should be lifted up into the abstrat. Is this all wind-driven?

We use the term "ice-free" not "free of ice" in general.

Page 3, Line 7: Add what products is "validated".

Page 5. Last line. "Ice sheet" is used for the large inland thick glaciated areas in Antarctica and Greenland. You probably mean "sea-ice" here??? Page 10 and other places. Is it ok to use PSS as salinity unit? Should it not be absolute salinity, or unit less?

Figure 5: Please provide the section in a map.

Page 16: 4.1.2. This section is really about "Wind Forcing" and nothing else. So it should have this name, and not "Mean Monthly Observations". The Ekman equations

[Figure]

should be in the method section.

Table 1: Use other "names" than 1-6. Cold and Warm, Fresh, Salty and Medium Salinity? SO 1=WF, 2=WMS, for example... (Numbers are more difficult to remember than names. . ..)

Page 18. Line 20: Why? Mixing with saltier water below? Or sea ice formation. When is the first onset of freezing? And what is the "normal/mean" for this freeze-up?

Page 19. Line 15-16. While there is "no evidence that "sea-ice melting can create such a layer of freshwater" – is there evidence that it can not? This is my major point #1.

Conclusion: The evaluation is OK, and described nicely. But what is the main message? What is learned of the river water flow? This still needs to be described. Main point#2.

––––––––––––––––––––––––––––

---

## Author Comment (AC4) · 16 May 2020

The presented observations of surface freshwater distribution is from an interesting area of the Arctic Ocean. The observations are presented in a nice manner. And there is plenty of figures, with many detailed results. This is all fine. There is not much available knowledge on how the river-water spreads north along the shallow Siberian shelves, so this paper is potentially a significant contribution in that regard. Beside some issues already noted by the other reviewers, like language, I have two larger issues that made me tick the "major" box here. My "major" science concern is the contribution from sea-ice melt. I think you

need to do a somewhat better job at addressing this possible contribution. It is difficult, but it can potentially explain much of the freshwater available. If delta-18 O samples were available, one could differ between these two sources perhaps, although I think much of the river-water from previous summers probably freeze-up, and some of the newly melted sea ice could thus just be old river water. I also find that you mix Results and Discussion in the 3. Results section. This may be OK, but then you need to call this 3. Results and Discussion. And then the sections 4.X are somewhat also Results and discussion. And then you cannot have "Discussion and Conclusion" in section 6. See? A re-struction is needed. There is also no general conclusion drawn on what actually spreads the freshwater to the north. Is this wind driven mostly? Or not?

Answer:

Thank you very much for your recommendations! We discuss the delta-O18point below. We reorganised the paper and moved a part of the "Results" section to the "Data" section, as it was suggested by the reviewer 2. We also added some information and citations to the Introduction part, and tried to separate the Discussion from the Conclusion, to bring forward the main message of the paper (yes, in our opinion, the river waters were driven northwards by the winds). The CTD cast at the most northern point of 126°E section was done at the beginning of September, so we suggest that the wind conditions in August resulted in this arrival of river water to the north of the Laptev Sea).

I also have a general suggestion: Provide the spatial mean vertical profile down to 100 m for T and S, and use that to describe the mean stratification. THEN – AFTERWARDS, you can present the spatial and temporal changes from this mean profile.

Answer:

Below, we provide the two requestions profiles (temperature and salinity; the shaded area represents the median +/- STD values). As you can see, the "mean stratification" is not really representative for the Laptev and East-Siberian Seas, because the physical characteristics of the upper-layer water masses vary a lot at the end of the summer season. Additionally, we did a lot of measurements in very shallow areas where the depths were between 30 and 50 m, so the calculated mean (or median) vertical profile is a composite of "shallow" and "deep" vertical profiles. Finally, we have only 45 measurements at 2 m depth, and most of the vertical profiles (141) start at 5 m depth (Fig.2). The vertical profiles from th point-measurements (CTD stations) are very useful, but cannot be used to discuss the temporal changes from these CTD profiles, as stations were not repeated. We presented the spatial and temporal changes using satellite (surface) data, as they had better coverage, both in time and space Nevertheless, we added these figures to the manuscript along with a figure with our calculations of MLD for each profile (Fig.1), please, see an answer to the RC2 and the revised manuscript for further details.

The full caption of Figure 1: (Left column)Vertical profiles of conservative temperature (a) and practical salinity (b) from CTD measurements in the upper 50 meters. Red stars indicate the mixed layer depth, calculated using (de Boyer Montégut et al. (2004) method (see details in the text). Colored profiles show the cases when the MLD is deeper than 7 m and gray profiles indicate when the MLD is shallower than 7 m. (Right column) : Vertical profiles of median conservative temperature (a) and practical salinity (b) with their STD (calculated for 146 CTD stations in both shallow and deep areas).

Minor issues:

Abstract: Your main explanation for how the river water is transported out from the river mouth area should be lifted up into the abstract. Is this all wind-driven?

Answer:

We suggest thatat this mainly is wind-driven. The abstract was corrected to: "... The surface gradients and mixing of river and sea water in the ice free and ice covered areas were described with a special attention to the marginal ice zone. The Ekman transport was calculated to better understand the pathway of surface water displacement. We

suggest that the fresh water was pushed northward, close to the MIZ and under the sea ice, which was confirmed by the oxygen isotope analysis. T-S diagrams using surface satellite estimates were used to investigate the transformation of the surface water masses at synoptic scales and reveal the presence of river waters on the shelf of the East-Siberian Sea."

We use the term "ice-free" not "free of ice" in general.

Answer: Corrected

Page 3, Line 7: Add what products are "validated".

Answer: Corrected to "SST DMI and SSS SMOS"

Page 5. Last line. "Ice sheet" is used for the large inland thick glaciated areas in Antarctica and Greenland. You probably mean "sea-ice" here???

Answer: Yes, corrected to "the number of measurements is low due to the presence of the coast and islands even without the sea ice".

Page 10 and other places. Is it ok to use PSS as salinity unit? Should it not be absolute salinity, or unit less?

Answer: Corrected to unitless for salinity, as all reviewers advised

Figure 5: Please provide the section in a map.

Answer: Added the maps to the figures 5 and 7 with section positions

Page 16: 4.1.2. This section is really about "Wind Forcing" and nothing else. So it should have this name, and not "Mean Monthly Observations". The Ekman equations should be in the method section.

Answer: Corrected

Table 1: Use other "names" than 1-6. Cold and Warm, Fresh, Salty and Medium Salinity? SO 1=WF, 2=WMS, for example... (Numbers are more difficult to remember

than names. . ..)

Answer: Corrected, thank you for your suggestion!

Page 18. Line 20: Why? Mixing with saltier water below? Or sea ice formation. When is the first onset of freezing? And what is the "normal/mean" for this freeze-up?

Answer: " By September 13, the SST and SSS variability slows down." We can hypothesize that the vertical mixing, a weaker river discharge (Fig. 3), and a continuously decreasing radiative income (Fig. 4) impact the variability of surface water characteristics. The second yearly maximum of river discharge occurs at the beginning of August. The warm and fresh river water is redistributed and transformed in the surface layer of the Laptev Sea during the month of August, but after several cyclones at the beginning of September, there is no additional important source of heat and fresh water that would maintain the variability of water masses. Over all, in September, the water mass CS (ex 6) progressively occupy the ice-free surface of the Laptev sea instead of other ("transformed") water masses observed there in August. The ice formation starts at the end of September but is achieved only by November, so it seems that freezing will begin only after the heat accumulated during the summer season is released to the atmosphere and the water temperature at the surface drops to the freezing point (Fig.5).

The full caption for Fig. 5: "Ice chart for the Arctic Ocean for September 3 - September 11, 2018. Colors indicate total sea ice concentration according to the analysis done at AARI (Sept 11), Canadian Ice Service (Sept 3), National Slow and Ice Data Center (Sept 6). Colored lines show the occurrence of sea ice edge for September 11-15 over the period 1979-2012 with SSMR-SSM/I-SSMIS data (NASATEAM algorithm). From http://wdc.aari.ru/datasets/d0042/2018/aari_20180903-20180911.pdf"

Page 19. Line 15-16. While there is "no evidence that "sea-ice melting can create such a layer of freshwater" – is there evidence that it can not? This is my major point #1.

Answer:

Yes, there is. We added a section on the use of the delta18 oxygen isotopic data to the Results and Discussion. Using a three-component simple model (marine water / river water (meteoric water) / sea ice melt water) described in Bauch and Chernyavskaya, 2018 and the isotopic analysis from surface water samples, we calculated the fractions of each source (Fig. 6). The isotopic analysis revealed that the most important fraction of river waters was brought to the shelf and continental edge of the East-Siberian Sea. At the same time, the water samples at the northern part of the 126 E section consist of 10-15 % of river water and only 0-5% by the sea ice melt.

The full caption for the Figure 6: Fractions of river water (a), sea ice melt water (b) and marine water (c), calculated using d-O18 measurements and Bauch andCherniavskaia (2018) 3-components model of freshwater balance. A thin black line shows the position of sea ice edge on August 31, 2018,when the northern stations of the meridional (5) section along 126 E were done in the MIZ, and the blue line shows the sea ice edge onSeptember 16, 2020, when the ARKTIKA-2018 expedition was working in the MIZ of the East-Siberian Sea. Please, note that the colorbar scale is different for each water fraction

Conclusion: The evaluation is OK, and described nicely. But what is the main message? What is learned of the river water flow? This still needs to be described. Main point#2.

Answer:

For the first time, we followed how the river water input was distributed and where it was stored in the Laptev and the East-Siberian Sea at synoptic scale. This became possible, first of all, due to a new satellite-derived salinity field in this region, a vast range of in situ measurements and also results of geochemical analysis. The shelf area of the Laptev and the East-Siberian Seas was described as a substantial region of sea ice production for the central Arctic by, e.g. Ricker et al., 2016, so the fresh water

pathways in the Arctic should be understood better. The transformation of fresh river water input occurs very quickly during the Arctic summer and disappears as well on the order of 1-2 weeks. Some part of the fresh water was clearly mixed over the shelf of the Laptev Sea under the wind-driven mixing, but a very important part was brought northward and to the East-Siberian Sea, under the ice. This result is different from a concept that fresh river water propagates mainly eastward, following the coastline under the Coriolis force. It is also different from the suggestion of Morison et al.2012, where the displacement from the eastern shelf Seas is northward (to the Central Arctic) with a low Arctic Oscillation Index (AO) and eastward with a high AO. In 2018 the mean AO index was high , but we showed that an important part of the river water was transported to the central basin. To better evaluate the freshwater budget, we suggest that future models assimilate the estimates of river discharge, a new satellite-derived sea surface salinity, and winds.

Please, see a new version of Discussion and conclusions.
* * *
**Fig. 1.** Vertical profiles of median conservative temperature (a) and practical salinity (b) with their STD (calculated for 146 CTD stations in both shallow and deep areas), and T and S with calculated MLD

[Figure]

[Figure]

[Figure]

[Figure]

**Only stations where 2m measurements are present:**

**Fig. 2.** The upper 10 meters of T, S, density from vertical CTD profiles

[Figure]

[Figure]

**Fig. 3.** The Lena River discharge in summer months 2018, data from Arctic GRO dataset,
(https://www.arcticrivers.org/data)

[Figure]

**Fig. 4.** Radiative balance components (Watt/m2) measured from RV Akademik Tryoshnikov during ARKTIKA-2018 expedition: magenta - shortwave radiation; blue, longwave radiation; black, total (done by A.Pashkov)

[Figure]

**Fig. 5.** Ice chart for the Arctic Ocean for September 3 - September 11, 2018. Colors indicate total sea ice concentration according to the analysis done at AARI (Sept 11), Canadian Ice Service (Sept 3),

[Figure]

**Fig. 6.** Fractions of river water (a), sea ice melt water (b) and marine water (c), calculated using d-O18 measurements and Bauch andCherniavskaia (2018) 3-components model of freshwater balance.

---

## Author Response (AR1)

Overview.

This manuscript uses a variety of resources, including in situ measurements, satellite measurements and reanalysis products, to describe the evolution of the Laptev and East Siberian Seas in late summer of 2018. The manuscript begins with a correct statement: that the region of interest is understudied. I think that the manuscript should be publishable, but only after significant attention has been directed (1) at the introduction (for context) and conclusions (for the meaning of the results in context), and (2) at Ekman fluxes and Ekman pumping. For the Ekman point, I believe the calculations to be invalid in shallow and coastal waters; see specific point 10 below. More on point (1)

[Figure]

follows, because at present, I think that the manuscript reads more like a report than a paper.

The manuscript introduction has some weaknesses. It tells the reader some of what happens in the region, but it does not really say why the region is important, why it matters. If I were making a case as to why a reader should care about the region, I would be looking at freshwater fluxes, specifically the role of the Siberian shelves in modulating the inflow to the Arctic Ocean of the great Siberian rivers, either those that discharge directly into the Laptev and East Siberian Seas, or those west of Severnaya Zemlya, some of whose freshwater runoff enters the region through the Vilkitskiy Strait. The most recent (and very good) review of such issues is the Eddy Carmack paper (JGR 2016).

Also, if the region is understudied (and I agree that it is) then the introduction needs to describe whatever of importance has been published on the region. Item 4 below gives three references led by Tom Armitage. They are pan-Arctic remote sensing papers that have quite a bit to say about the Siberian Shelves, and they are not mentioned. Plus I could add Johnson & Polyakov (GRL 2001), Semiletov et al. (GRL 2005), Lenn et al. (GRL 2009 and JPO 2011), and the Lenn papers remind me that the manuscript talks about currents but does not mention tides, which are important in the shelf seas. There may be more papers, these are just a few that come to mind. I am not confident that the authors have thoroughly reviewed the literature.

Then concerning the final section called Discussion and conclusions; this section is really not much more than a brief restatement of the work reported in the preceding sections. If the introduction does not tell the reader why the region is important, then the conclusions cannot then tell the reader why the new results matter.

Specific comments are in order of occurrence.

1. The first paragraph on pp 1-2 describes the region. Reference should be added to the map of Figure 1; all locations in the text should be labelled, so please add text

"Severnaya Zemlya" to the map. Later you refer to Arkticheskiy Cape, add this as well.

2. In the same first paragraph, it is correctly stated that the region is little-studied, but I think there should be a sentence stating why the authors think the region is important. At the end of para. 1 on the top of p 2, add a sentence "The region is important because ...".

3. When talking about salinity, it is not appropriate to use "PSS". If using the new Absolute Salinity, then you can say parts per thousand, ppt, or use the "per mille" symbol. If not, then just "salinity of nn.n" or "nn.n in salinity" is correct.

4. Recent papers led by Armitage using Envisat, Cryosat and GRACE between 2003-2014 seem not to have been looked at: JGR 2016 is about sea surface height variability, with discussion of Siberian shelf seas; Cryosphere 2017 is about surface geostrophic circulation; GRL 2018 is about sea level & surface circulation response to the Arctic Oscillation. Some the material in these papers is directly relevant, and this omission should be corrected.

5. All figures with multiple panels, please label them a, b, c, etc. and refer to the panels as such in the manuscript text. All captions must state all plotted quantities and their units.

6. Section 3.1.2 on salinity, and cf SMOS text in 2.2.2. I doubt that the spatial resolution is as high as it appears in Figure 3. I understood SMOS to resolve at about 100 km, in 2.2.2 the authors mention sampling at 15 km resolution, but adjacent points are surely not independent. How does this affect their statistics?

7. Section 4 first & second lines, your temperature accuracy cannot be 1 m$^{\circ}$C, so why are you quoting temperature values to three decimal places?

8. Figures 5 and 7, Hovmoller plots: you are inconsistent. Figure 5, longitude (x-axis), days (y-axis); figure 7, vice-versa. Pick one orientation and stick to it.

9. Figure 8, temperature sections. Improve your presentation, please. Viewing the

[Figure]

PDF, about half of the figure is just black and any temperature structure is obscured. The simplest solution would be to plot contours in white – not black!

10. Section 4.1.2. Please improve your terminology. "Transport" is typically a volume transport, units mˆ3/s. In your eq. 1, stress (N/m2) divided by [ density (kg/m3) * Coriolis (sˆ-1) ] has units m2/s. These units appear in tiny (almost unreadable) notation in figure 9. This is neither a velocity nor a transport. Importantly, though: are your Ekman calculations valid in shallow water? Ekman's assumptions included (1) no boundaries (remote from coasts), and (2) deep water (typically >200 m). What is the Ekman layer depth? Is it not more likely that upwelling / downwelling are dominated by sea surface height changes in shallow water? For instance, a wind from the west will cause surface water movement to the right (the south) in the northern hemisphere. Water "piles up" against the coast, and that induces downwelling (at the coast): see papers by Steven Lentz, for example. Is this in accord with your calculated vertical velocities in figure 9? I would say not.

———————————————————

[Figure]

Ocean Sci. Discuss.,
https://doi.org/10.5194/os-2019-60-AC2, 2020

[Figure]

General comments: "The manuscript introduction <...> does not really say why the region is important, why it matters <...>: freshwater fluxes, specifically the role of the Siberian shelves in modulating the inflow to the Arctic Ocean of the great Siberian rivers, either those that discharge directly into the Laptev and East Siberian Seas, or those west of Severnaya Zemlya, some of whose freshwater runoff enters the region through the Vilkitskiy Strait. The most recent (and very good) review of such issues is the Eddy Carmack paper (JGR 2016). Item 4 below gives three references led by Tom Armitage. They are pan-Arctic remote sensing papers that have quite a bit to say

about the Siberian Shelves, and they are not mentioned. Plus I could add Johnson & Polyakov (GRL 2001), Semiletov et al. (GRL 2005), Lenn et al. (GRL 2009 and JPO 2011), and the Lenn papers remind me that the manuscript talks about currents but does not mention tides, which are important in the shelf seas." (+ commentary 2) "In the same first paragraph, it is correctly stated that the region is little-studied, but I think there should be a sentence stating why the authors think the region is important. At the end of para. 1 on the top of p 2, add a sentence "The region is important because ...". " (+commentary 4) " Recent papers led by Armitage using Envisat, Cryosat and GRACE between 2003- 2014 seem not to have been looked at: JGR 2016 is about sea surface height variability, with discussion of Siberian shelf seas; Cryosphere 2017 is about surface geostrophic circulation; GRL 2018 is about sea level & surface circulation response to the Arctic Oscillation. Some the material in these papers is directly relevant, and this omission should be corrected.

Answer: Thank-you, we have rewritten introduction, conclusion and annex A to take into account these general and specific comments, and correct these omissions.

Specific comments: 1. The first paragraph on pp 1-2 describes the region. Reference should be added to the map of Figure 1; all locations in the text should be labelled, so please add text "Severnaya Zemlya" to the map. Later you refer to Arkticheskiy Cape, add this as well.

Answer: Corrected, please see a new version of Fig.1

The full version of caption of Fig.1: Legs and stations of the ARKTIKA-2018 expedition overlayed on the bathymetry from ETOPO1 "1 Arc-Minute Global Relief Model" " (Amante and Eakins (2009)). CTD stations are shown with white dots. The color indicates the number of days since August 1, 2018. The sea ice edge position is indicated with a red dashed line for the beginning (August 21) and with the purple dashed line for the end of the expedition (September 21). The ice edge is based on the sea ice mask provided in the SST DMI product. Numbers indicate positions of 10 oceanographic

transects discussed below. The black triangle in the north of the Komsomolets Island shows the Arkticheskiy Cape. The Severnaya Zemlya Archipelago consists mainly of the Komsomolets, the October Revolution, and the Bolshevik Islands (with smaller islands not shown here). The black box indicates the Shokalskiy Strait between the October Revolution and the Bolshevik Islands. The Yana River estuary is situated southward the Yanskiy Bay (out of the map).

2. In the same first paragraph, it is correctly stated that the region is little-studied, but I think there should be a sentence stating why the authors think the region is important. At the end of para. 1 on the top of p 2, add a sentence "The region is important because ...".

Answer:

Please, see a new version of the introduction.

3. When talking about salinity, it is not appropriate to use "PSS". If using the new Absolute Salinity, then you can say parts per thousand, ppt, or use the "per mille" symbol. If not, then just "salinity of nn.n" or "nn.n in salinity" is correct.

Answer:

Agree that the use of PSS is not appropriate. Nevertheless, the community of satellite-derived salinity widely uses the psu, practical salinity unit, to quantitatively describe the salinity; "in situ" practical salinity is computed from CTD measurements of conductivity, and also uses the "psu" scale. For the validation of SSS we use practical salinity. Absolute salinity was used only to calculate water density. We would prefer to change the units from pss to psu, but agree to make it unitless.

4. Recent papers led by Armitage using Envisat, Cryosat and GRACE between 2003-2014 seem not to have been looked at: JGR 2016 is about sea surface height variability, with discussion of Siberian shelf seas; Cryosphere 2017 is about surface geostrophic circulation; GRL 2018 is about sea level & surface circulation response to the Arctic

[Figure]

Oscillation. Some of the material in these papers is directly relevant, and this omission should be corrected.

Answer:

Corrected, please, see a new version of Appendix A.

5. All figures with multiple panels, please label them a, b, c, etc. and refer to the panels as such in the manuscript text. All captions must state all plotted quantities and their units.

Answer:

Corrected.

6. Section 3.1.2 on salinity, and cf SMOS text in 2.2.2. I doubt that the spatial resolution is as high as it appears in Figure 3. I understood SMOS to resolve at about 100 km, in 2.2.2 the authors mention sampling at 15 km resolution, but adjacent points are surely not independent. How does this affect their statistics?

Answer:

The "initial" SMOS instrument (radiometric) resolution is 50 km (which we meant to explain in 2.2.2, line 17-18), but the SMOS SSS product distributed by ESA is already sampled in the ISEA grid with a resolution of 15 km. In other words, the spatial resolution of SMOS SSS Level 2 v662 product is 15 km, we just resampled all satellite products at the same grid for convenience. This "oversampling" of SMOS SSS at 15 km is practical for two reasons. First, to retain the real salinity gradients observed with in situ measurements and not to smooth them to 50 km when comparing with SMOS SSS. The spatial resolution of ship measurements depends on the ship speed (8 knots $\sim$ 3 m/s), pumping speed (16 l/s) and the CTD measurement frequency (24 Hz), and is of order (o) 1 m. After processing the raw data, its resolution is (o)250 m. A 7.5-km in situ measurement average corresponds to 30 minutes of TSG measurements, as line 6, p.3.1.1 (and a 15-km pixel represents one hour of measurements). Second, this

retains SSS on the same grid as the rather high resolution SST for further calculations, e.g., density.

7. Section 4 first & second lines, your temperature accuracy cannot be 1 degree C, so why are you quoting temperature values to three decimal places?

Answer:

Corrected, and temperature reported to 2 decimals

8. Figures 5 and 7, Hovmoller plots: you are inconsistent. Figure 5, longitude (x-axis), days (y-axis); figure 7, vice-versa. Pick one orientation and stick to it.

Answer:

The orientation of the Hovmoller plots was chosen to have a better geographical representation: meridional section has longitude in y-axis, and thus, days in x-axis, and zonal section have latitude in x-axis, and days in y-axis. We added the maps with the positions of these virtual sections to illustrate it (Fig.2, 3).

The full caption to figures 2-3 are the following: Fig. 2. "Hovmoller diagram of DMI SST (a), SMOS SSS "A" (b), ASCAT wind speed (c) and ERA5 SLP (d) for the virtual meridional transect at 126 degree E. Sea ice concentration (AMSR2) is indicated with a blue color, see Fig. [SSS validation] for the color scale. The bathymetry along the transect (e) is extracted from "1 Arc-Minute Global Relief Model" " (Amante and Eakins (2009)). The position of a virtual transect is shown on SST SMI and SMOS SSS "A" maps for August 26, 2018 (f, g)."

Fig. 3 "Hovmöller diagram of DMI SST (a), SMOS SSS "A" (b), ASCAT wind speed (c), and ERA5 sea level pressure (d) for the zonaltransect at 78 degree N. Small circles at SST and SSS diagrams show in situ measurements of temperature and salinity (first CTD or TSG at 6.5m). Sea ice concentration (AMSR2) is indicated with a blue color, see Fig.5 for the color scale. The bathymetry along the virtual transect (e)is extracted from "1 Arc-Minute Global Relief Model" (Amante and Eakins (2009)). The position of

[Figure]

a virtual transect is shown at SST SMIand SMOS SSS "A" maps for August 26, 2018 (f, g)"

Figure 8, temperature sections. Improve your presentation, please. Viewing the PDF, about half of the figure is just black and any temperature structure is obscured. The simplest solution would be to plot contours in white – not black!

Answer:

Corrected, please see Fig. 4 here. The full title of this figure is: "Temperature, degree C, (a, e, i, first column), salinity, (b, f, j, second column), water density, kg/m3 (c, g, k, third column) and buoyancy frequency, 1/s, (d, h, l, fourth column) obtained from CTD measurements in the upper 50 m for section 1 northward of Arkticheskiy Cape (upper row), section 10 across the Shokalskiy Strait (second row), and section 4 across the Vilkitskiy Strait (lower row). See Fig.1 for the section's positions. The zero km is always placed at the southern point of each section "

10. Section 4.1.2. Please improve your terminology. "Transport" is typically a volume transport, units mËĘ3/s. In your eq. 1, stress (N/m2) divided by [ density (kg/m3) * Coriolis (sËĘ-1) ] has units m2/s. These units appear in tiny (almost unreadable) notation in figure 9. This is neither a velocity nor a transport. Importantly, though: are your Ekman calculations valid in shallow water? Ekman's assumptions included (1) no boundaries (remote from coasts), and (2) deep water (typically >200 m). What is the Ekman layer depth? Is it not more likely that upwelling / downwelling are dominated by sea surface height changes in shallow water? For instance, a wind from the west will cause surface water movement to the right (the south) in the northern hemisphere. Water "piles up" against the coast, and that induces downwelling (at the coast): see papers by Steven Lentz, for example. Is this in accord with your calculated vertical velocities in figure 9? I would say not.

Answer:

[Figure]

The assumption with Ekman transport is exploited as the region of study has a very strong vertical stratification during the survey, which isolates the surface Ekman layer from the bottom. It was used as an illustration of possible mixing mechanisms, although we agree that over the most shallow areas this concept is not realistic. Please, see a new version of calculated horizontal Ekman transport et pumping (Fig. 5, 6)

The conclusions cannot then tell the reader why the new results matter.

Answer:

For the first time, we followed how the river water input was distributed and where it was stored in the Laptev and the East-Siberian Sea at synoptic scale. It became possible, first of all, due to a new satellite-derived salinity field in this region, a vast range of in situ measurements and also results of geochemical analysis. The shelf area of the Laptev and the East-Siberian Seas was described as a substantial region of sea ice production for the central Arctic by, e.g. Ricker et al., 2016, so the fresh water pathways in the Arctic should be understood better. The transformation of fresh river water input occurs and diminish very quickly during the Arctic summer, on the order of 1-2 weeks. Part of the fresh water was clearly mixed over the shelf of the Laptev Sea by wind-driven mixing, but a very important part was transported northward and to the East-Siberian Sea, under the ice. This result is different from a concept that fresh river water propagates mainly eastward, following the coastline under the Coriolis force. It is also different from the suggestion of Morisson et al.2012, where the displacement from the Eastern shelf Seas is northward (to the Central Arctic) with a low Arctic Oscillation Index (AO) and eastward with a high AO. In 2018 the mean AO index was high , but we showed that an important part of the river water was transported to the central basin. To better evaluate the freshwater budget, we suggest that future models assimilate the estimates of river discharge, a new satellite-derived sea surface salinity, and winds.

[Figure]

**Fig. 1.** Legs and stations of the ARKTIKA-2018 expedition overlaid on the bathymetry

[Figure]

**Fig. 2.** Hovmoller diagram of DMI SST (a), SMOS SSS "A" (b), ASCAT wind speed (c), and ERA5 sea level pressure (d) for the meridional transect at 126 E

[Figure]

[Figure]

**Fig. 3.** Hovmoller diagram of DMI SST (a), SMOS SSS "A" (b), ASCAT wind speed (c), and ERA5 sea level pressure (d) for the zonaltransect at 78 N. Small circles at SST and SSS diagrams show in situ measurements

[Figure]

**Fig. 4.** Temperature, salinity, density obtained from CTD measurements in the upper 50 m for sections 1, 10, and 4

[Figure]

**Fig. 5.** Mean monthly Ekman transport and pumping together with wind speed in August 2018

[Figure]

**Fig. 6.** Mean monthly Ekman transport and pumping together with wind speed in September 2018

[Figure]

Ocean Sci. Discuss.,
https://doi.org/10.5194/os-2019-60-RC2, 2020

[Figure]

The submitted manuscript "Surface waters properties in the Laptev and the East-Siberian Seas in summer 2018 from in situ and satellite data" by Tarasenko et al., presents and validates DMI SST L4 and SMOS SSS products (including errors) in a novel way against in situ temperature and salinity data (CTD profiles from the upper 6.5 m and TSG data at 6.5 m depth) obtained during the ARKTIKA-2018 expedition in the Laptev Sea and the East–Siberian Sea in August and September 2018. Since this region is highly under sampled, this study is of high relevance for the scientific community focusing on air-ice-ocean interaction processes in the Arctic Ocean. In the study, they follow the north-eastward spreading of warm and fresh river water from the Lena River Delta towards the central Laptev Sea and further towards and under the sea ice

in the East-Siberian Sea. Also, river water inflow from the Kara Sea to the Laptev Sea is followed. By combining SST and SSS data products with satellite-derived products of sea ice concentration, wind speed (and direction), and ERA5 reanalysis fields of sea level pressure, they investigate wind-induced horizontal displacements of the river water plume due to Ekman transport and change in the strength of the stratification across different shelf-slope transects due to Ekman pumping (downwelling/upwelling). Based on the satellite-derived SST and SSS data, they also suggest different types of surface water masses for the area, and further present their distribution and presence during the studied period.

Even though this study is of high relevance, the submitted manuscript needs a major revision before suited for publication in Ocean Science. First of all, the English language needs to be improved and corrected. There are simply too many grammatical and syntax mistakes and typos for me to point out, so I will only mention some of the most general mistakes below. Secondly, I spent too many hours reading the manuscript due to an unfinished appearance with in some places lack of background information or a logical flow. Hence, I found it a bit hard to get the main objectives, the main results, and the main conclusions from the study. More specific comments follow below.

General on language:

Use definite (the) or indefinite (a/an) articles where suitable.

Remember to use capitol letter S in named seas: Laptev Sea, Kara Sea, East-Siberian Sea etc. This is not used consistently in the text. Same with named rivers, use capitol letter R (Lena River etc.) Same with named archipelagos, use capitol letter A (Severnaya Zemlya Archipelago etc.) Write the Arctic Ocean with capitol letter O. Use capitol letter W in named water masses: Atlantic Water, Arctic Water, etc.

Be consistent with use of present (is) or past (was). Use past and not present when referring to others work.

[Figure]

Avoid plural in surface water, river water, etc. if you don't have several named surface or river waters.

"Ice free areas" instead of "free of ice areas"?

In the north/south/east/west, not on the north/south/east/west.

Practical salinity scale (PSS) or practical salinity unit (PSU) is termed salinity with no unit, while absolute salinity is termed absolute salinity with unit g/kg.

Title

Perhaps change the title to "Properties of surface water masses ..." To use surface water in plural seems a bit strange to me. You are suggesting six different water masses in the surface, not several surface waters, or?

Introduction

Generally, the introduction lacks objectives and proper background information of the study area. What is known in this region already, what studies have been done in this under-sampled area of the Arctic Ocean, what about river discharges into the study area (the river water plume is the main research focus of the study), and why is this area important?

Page 1, Line 19: "In the Arctic region, a strong seasonal warming and cooling with sea ice melting and freezing modify ..."

Page 2, Line 5: "... the upper ocean water displacement ..." Not plural.

Page 2, Line 9: " via the energy vertical distribution ..." What energy?

Page 2, Lines 14-19: Rewrite this paragraph and improve language, for instance:

Line 14: Rewrite to i.e. "... proposes to use the term "surface layer" instead of "mixed layer" for the Arctic Ocean, because the water ..."

Line 15: What is meant with "the water horizon lying between the sea surface and the

[Figure]

Arctic main halocline"? Use another word than horizon?

Line 17: "m depth in the Eastern Arctic Ocean (Dmitrenko et al., 2012) and at 100-200 m depth in the Western Arctic Ocean ..."

Page 2, Line 23: Rewrite to i.e. "... the saline waters, which was considered the mean Arctic Ocean salinity at that time."

Page 2, Line 25: Reorder sentence to "Cherniavskaia et al. (2018) reported an overall salinity in the upper 5-50 m layer to lie within the range from 30.8 to 33 based on in situ data in the Laptev Sea during 1950-1993 and 2007-2012.

Page 2, Line 31: Remember unit on density (kg m-3)

Page 3, Line 8: Define L-band satellites.

Data and Methods

All information regarding the data and methods (presentation of the in situ hydrographic sections, processing and analysis methods) that are described and included in the results section should be moved to the data and method section. More details are given later. It would also be useful to include a link or doi to the downloaded data products (DMI SST L4, SMOS SSS L2 (you are making weekly averages of these, are you using any other SSS products?), AMSR2 SIC (and specify all SIC products used), ASCAT C-2015 L3 for wind speed and direction, ERA5 reanalysis for SLP and air temperature). Several data products are mentioned and utilised in this study, and to better clarify which ones that are actually used, it would be good to start the description of the used data products, why these ones are chosen, and what has been done with the data products in this study (post-processing). Then, other alternative products used in other studies can be mentioned afterwards for comparison? This goes especially for Section 2.2.2, which now appears a bit messy with a lot of mixed information about others products and work and the products used in this study.

Page 3, Lines 19-21: Put all place names on the map in Figure 1.

[Figure]

Page 3, Line 27: Did you find a typical error estimate from the comparison with CTD measurements?

Page 4, Line 2: What interpolation methods were used?

Page 4, Line 6: Define AVHRR, MODIS and VIIRS

Page 4, Line 8: Define AMSR2

Page 4, Line 11: Your write "(hereafter referred to as "SST DMI")", then make sure you do. Several names are used on this product later in the manuscript (DMI SST L4, the SST field from DMI L4 product, blended DMI product?, temperature estimates provided by DMI, and others).

Page 4, Line 11: Define Level 4.

Page 5, Line 7: You write "100-km averaged ship SSS . . ." Is this TSG?

Page 5, Lines 12-13: Define SMAP CAP/JLP and SMOS BEC L3, REMSS

Page 5, Line 16: Define ESA.

Page 5, Line 17: Be consistent with naming of products. Here you use L2 SSS and not SMOS SSS L2.

Page 5, Line 18: What do you mean with individual SMOS SSS? Explain how the SMOS SSS are sampled over an Icosahedral Snyder Equal Area grid at 15 km resolution. Is this the interpolation?

Page 5, Line 23: Define ECMWF.

Page 5, Lines 26-29: Specify the correction with numbers. Also, give the criterion on the ACARD parameter. And what are the typical errors in the weekly SMOS SSS and the individual SMOS SSS?

Page 5, Lines 32-34: Where is the river plume? Define and introduce it. How does this affect the weekly SSS error?

[Figure]

Page 6, Line 4: Define all the ice masks provided.

Page 6, Line 10: Define ASCAT.

Results

Move all method and technical descriptions to Data and Methods. Only results should be presented here. The results section should have more focus on the SST and SSS distribution, the hydrographic sections, and highlight results to be discussed, not so much on technical details, error analysis and discussion, which I recommend you to move to the data and method section. In general when presenting and discussing the results, please be consistent and distinguishing between the parameters SST/SSS and in situ temperature/salinity. Don't use temperature or salinity on both.

Page 6, Line 22: Please define the river waters. Use plural form if there are several types of river water, it not, use singular form (river water).

Page 6, Lines 24-25: Move to Data and Method.

Page 7, Lines 1-7: This paragraph belongs in data and methods.

Page 7, Line 2: What are "basic statistics"?

Page 7, Lines 4-5 & 8-9: How much does the temperature and salinity change in the upper meters? Perhaps show a mean +/- std profile of the upper 10 meters?

Page 7, Lines 10-11: And how is it in ice free conditions?

Page 7, Line 15: In-situ surface layer temperature is then the upper 6.5 m (but not including the uppermost 2? meters typically)?

Page 7, Lines 17-19: This is a complicated sentence, so please rewrite. Can this be shown?

Page 7, Line 21: What grid? Please describe in data and method, and refer to it.

Page 7, Lines 28-29: I guess the 15 km SMOS resolution is the interpolated resolution?

[Figure]

Please be more precise when naming the different SMOS SSS products, which should all be clearly defined and stated in the data and method section.

Page 7, Line 31: Use either ARKTIKA-2018 expedition or Akademik Tryoshnikov measurements. Be consistent with the naming of data both in the text and in the figures.

Page 7, Line 33: Again, be consistent with naming of data products, I guess the vessel SSS is the CTD and TSG data from the upper 6.5 m from the ARKTIKA-2018 expedition?

Page 7, Lines 33-34: This belongs in the method description.

Page 8, Line 1: "SMOS post-processed SSS", what is this product compared to the other named products?

Page 9, Lines 2-3: Is the precision so high that you can use three decimals in SST?

Page 9, Lines 2-4: Put place names on the map in Figure 1.

Page 9, Lines 9-10: Please define river water somewhere (in data and method?). What is the definition of the river water plume?

Page 9, Line 10: Standard deviation of what?

Page 10, Line 1: What about the Katanga River Estuary?

Page 10, Line 2: Define the thermal fronts.

Page 10, Line 15: "at 125 E, . . ." what latitude?

Page 10, Line 20: Distribution of freshwater input, is that water with S = 0? Please define the characteristics of this freshwater.

Page 10, Line 21: Please refer to Figure 1. What Section number or longitude/latitude limits?

Page 10, Line 21: Temporal evolution of what? Introduce Figure 6 before Figure 7.

[Figure]

Page 10, Line 25: "shelf break" instead of "edge of the shelf"?

Page 10, Line 29: What was the wind direction during these series of cyclone passages?

Page 10, Line 30: Change to ". . . salinity increased by 1 . . ."

Page 10, Line 34: Change to ". . . mixing event induced by the wind."

Page 10, Lines 30-34: How does this relate to any river discharge data from the same period? Are there any model data to compare with?

Section 4.1.1

Change to something like "Water from the Kara Sea". You have not defined Kara Sea waters anywhere in the text.

Page 11, Lines 6-7: The temperature is decreasing?

Page 11, Line 7: Where and when has the exchange with the atmosphere taken place? Is the atmosphere colder here than in the central Laptev Sea?

Page 13, Line 1: You write ". . . significantly greater than . . ." How can you see relative amount of freshwater from one snap-shot and with different timing on each section occupation?

Page 13, Lines 2-5: Show place names on map in Figure 1. Also, define the hydrographic sections in data and methods with time, wind condition, and air temperature during their occupations. Then stick to the named Sections (Section numbers) later in the text.

Page 13, Line 7: 1024-1027 kg m-3?

Page 13, Line 14: What happens during cyclone passages? Clouds? Have you discussed this effect in Data and Methods? If yes, refer to it.

Page 13, Line 15: Can you provide any T and S limits?

[Figure]

Page 13, Line 17: By depression passage you mean low-pressure or cyclone passage?

Page 13, Lines 18-19 & Page 14 Lines 1-4: What about wind direction? Wind speed alone will not give the complete forcing pattern. Perhaps make some cross-correlation maps between wind speed/direction and SLP with SST and SSS? SLP and wind direction should be highly correlated.

Page 14, Lines 5-7: This belongs in Data and Methods.

Page 14, Line 9: Absolute salinity and conservative temperature are shown in Figure 8. Either change the text to absolute salinity and unit g/kg or change to practical salinity in figure.

Page 14, Line 11: You refer to different observations at specific latitudes in the figure, but the figure is presented with distance in km, so please refer to km as well. This applies for the other sections as well.

Page 14, Lines 12-16: What about melting under the sea ice due to the presence of warmer (above freezing point temperature) river water? This will cool the river water and still keep the water under the ice relatively fresh. Are there any $\delta$18O data (or other tracers) to be able to identify the source of this water?

Page 14, Line 15: You write: "The heat exchange with the sea ice might be more effective than with the atmosphere, . . .." Why?

Page 14, Line 19: "Below the pycnoclines, . . .." to what depth?

Page 15, Line 1: The 34.5 isohaline is not shown in the figure.

Page 15, Lines 2-3: What are the T and S characteristics of AW is this region? I don't agree that AW is seen at 100-120 m depth in Figure 8? Why is it instability in the AW layer?

Page 15, Line 5: Define all hydrographic sections in Data and Method (see earlier comment on this).

[Figure]

Page 15: Please remember to refer to figures.

Page 15, Line 11: You write: ". . . which is clearly seen in temperature signal that is negative even close to the surface." Is the temperature still above the freezing point, i.e. any melting under the ice?

Page 15, Lines 13-14: What kind of mixing?

Page 15, Line 19: How can you see an efficient mixing from these data?

Page 15, Line 20: Are the temperatures still above freezing?

Page 15, Lines 25-26: The river discharge information should be introduced in the introduction.

Page 15, Lines 27-28: Was the atmosphere colder or warmer than the river water or the mixed upper layer water? What about melting of sea ice from below? Is the river water warmer than the freezing point temperature?

Section 4.1.2

Some more background information is needed in the beginning of this section. It is also an analysis method and should be moved to the data and method section. The surface fronts in question should be defined. To get the Ekman transport, you need to integrate over the Ekman depth, what is the Ekman depth in this region? Assumptions made by Ekman were no boundaries, infinitely deep water, and no geostrophic flow. How are these assumptions met in this region? What happens when you have boundaries (coastlines)?

Page 16, Line 8: Give the reference to TEOS-2010 and show the formula for the drag coefficient CD.

Page 16, Line 10: Use Ekman pumping instead of vertical Ekman speed?

Page 17, Line 12: You write ". . . mixing of different water masses, . . ." What are these

water masses? You should introduce the water masses in the region in the introduction section.

Section 5

Perhaps start the results with this section?

Page 17, Line 17: And below 200 m?

Page 17, Line 23: Specify which of the Arctic Ocean waters that quickly change.

Page 17 Line 24: Add "..., and the synoptic satellite data provide ..."

Page 17, Line 32: The "classical" water masses, do they have a name?

Page 18: Refer to figures.

Page 18, Line 1: Add "... larger range than the near-surface (upper 6.5 m) in situ measurements ..."

Page 18, Line 7: A recognisable river front or a river plume front? Define.

Page 18; Line 8: 145 E? How do you define the sea edge? Is it a specific sea ice concentration limit?

Page 18, Line 11: You write "... captured under the ice and exposed back." How or when? What about melting sea ice from below?

Page 18, Lines 21-22: Show place names on map in Figure 1.

Page 18, Lines 24-25: There is also a high density within the range 22.5-30 and 3-4 °C.

Discussion and conclusion

This section provides little discussion or conclusions, mostly summary. There is more discussion in between the presentation of the results in the Results section.

[Figure]

Page 19, Lines 11-12: You write "The fresh waters displacement was associated with the Ekman transport." How? This is not well presented or described in the results.

Page 19, Lines 15-16: You write "... and there is no evidence that the sea ice melting itself can create such a considerable layer of freshwater." How or where have you presented this lack of evidence?

Page 19, Lines 18-20: What about river water melting sea ice from below? Are there any $\delta$18O data (or other tracers) to evaluate this?

Page 19, Line 24: Add "Calculated monthly Ekman pumping indicates the area of most intense mixing processes induced by wind."?

Page 19, Lines 25-26: Why is this included? This is not used in the discussion. Are there any conclusions from this?

Appendix

Page 20, Lines 7-10: What about Ekman transport to the south in September piling up water on the shelf and toward the coast in the south? This might induce downwelling (see Figure 9).

Figures

Figures with more than one panel should be labelled with a), b), c) etc. and then properly referred to in the figure captions and the text when the figures are described. Also use same font size in the figures with several panels. Remove titles on figures and add this text in the figure caption instead.

Figure 1: What is the definition of the sea ice edges? Add all place names mentioned in the text on the map.

Figure 2 and 3: Last sentences in the figure captions: this information should be provided in the data and method section.

[Figure]

Figure 4: Missing numbers on y-axis in lower left panel. Use white isolines below 0 degree Celsius/28 PSS? It is hard to see black lines on dark background. This is also the case for Figures 6 and 8.

Figures 5 and 7: Where is the data from? What bathymetry data is used?

Figure 8: Why do you show conservative temperature and absolute salinity in this figure and not in

Figure 6? At least use proper units when referring to Figure 8 in the text.

Figure 9: Increase the fonts. The two upper rows are not described in the text. What are the arrows in the lowest row, and why is the resolution higher than in the row above? Write units in the figure caption. Where is the data from?

Figure 10: Add the freezing point temperature line. Remove the figure title.

Figure 11: What are the boxes? Are the lines the freezing point temperature line?

Figure 12: Put the dates inside the plots instead.

Figure A1: Remove the titles from the figures. Write units in the figure caption. Increase the fonts.

Tables

Table 1: Add more information to the table caption.
* * *
[Figure]

Ocean Sci. Discuss.,
https://doi.org/10.5194/os-2019-60-AC3, 2020

[Figure]
Use definite (the) or indefinite (a/an) articles where suitable. Remember to use capital letter S in named seas: Laptev Sea, Kara Sea, East-Siberian Sea etc. This is not used consistently in the text. Same with named rivers, use capital letter R (Lena River etc.) Same with named archipelagos, use capital letter A (Severnaya Zemlya Archipelago etc.) Write the Arctic Ocean with capital letter O. Use capital letter W in named water masses: Atlantic Water, Arctic Water, etc. Be consistent with use of present (is) or past (was). Use the past and not present when referring to others' work. Avoid plural in

surface water, river water, etc. if you don't have several named surface or river waters. "Ice free areas" instead of "free of ice areas"? In the north/south/east/west, not on the north/south/east/west. Practical salinity scale (PSS) or practical salinity unit (PSU) is termed salinity with no unit, while absolute salinity is termed absolute salinity with unit g/kg.

Answer:

Thank you very much. We have thoroughly edited the paper to follow your recommendations.

Title. Perhaps change the title to "Properties of surface water masses ..." To use surface water in plural seems a bit strange to me. You are suggesting six different water masses in the surface, not several surface waters, or?

Answer:

We will follow this recommendation and change the title of the paper.

Introduction. Generally, the introduction lacks objectives and proper background information of the study area. What is known in this region already, what studies have been done in this under-sampled area of the Arctic Ocean, what about river discharges into the study area (the river water plume is the main research focus of the study), and why is this area important?

Answer:

We have rewritten the introduction with more focus of the undersampling of this region and on the most important river discharges in this area. We have also added the following figure for the Lena river discharge in the Data and Methods section (Fig. 1).

Page 1, Line 19: "In the Arctic region, a strong seasonal warming and cooling with sea ice melting and freezing modify . . ." Answer: Corrected

Page 2, Line 5: ". . . the upper ocean water displacement . . ." Not plural. Answer:

[Figure]

Corrected

Page 2, Line 9: " via the energy vertical distribution . . ." What energy? Answer: We meant the thermal energy

Page 2, Lines 14-19: Rewrite this paragraph and improve language, for instance: Line 14: Rewrite to i.e. ". . . proposes to use the term "surface layer" instead of "mixed layer" for the Arctic Ocean, because the water . . ." Answer: Corrected

Line 15: What is meant with "the water horizon lying between the sea surface and the Arctic main halocline"? Use another word than horizon? Answer: Corrected for "layer"

Line 17: "m depth in the Eastern Arctic Ocean (Dmitrenko et al., 2012) and at 100-200 m depth in the Western Arctic Ocean . . ." Answer: Corrected

Page 2, Line 23: Rewrite to i.e. ". . . the saline waters, which was considered the mean Arctic Ocean salinity at that time." Answer: Corrected

Page 2, Line 25: Reorder sentence to "Cherniavskaia et al. (2018) reported an overall salinity in the upper 5-50 m layer to lie within the range from 30.8 to 33 based on in situ data in the Laptev Sea during 1950-1993 and 2007-2012. Answer: Corrected

Page 2, Line 31: Remember unit on density (kg m-3) Answer: Corrected

Page 3, Line 8: Define L-band satellites. Answer: Corrected, 1.43 GHz

Data and Methods. All information regarding the data and methods (presentation of the in situ hydrographic sections, processing and analysis methods) that are described and included in the results section should be moved to the data and method section. More details are given later. Answer: Thank you for your recommendation. We reorganized some sections and added information and one figure on CTD and in situ measurements on Page 3, Line 27 (see below).

It would also be useful to include a link or doi to the downloaded data products (DMI SST L4, SMOS SSS L2 (you are making weekly averages of these, are you using any

[Figure]

other SSS products?), AMSR2 SIC (and specify all SIC products used), ASCAT C-2015 L3 for wind speed and direction, ERA5 reanalysis for SLP and air temperature). Several data products are mentioned and utilised in this study, and to better clarify which ones that are actually used, it would be good to start the description of the used data products, why these ones are chosen, and what has been done with the data products in this study (post-processing). Then, other alternative products used in other studies can be mentioned afterwards for comparison? This goes especially for Section 2.2.2, which now appears a bit messy with a lot of mixed information about others products and work and the products used in this study.

Answer:

Corrected where possible. We tried to unify the products names used in the paper and clarified the representation and processing.

Page 3, Lines 19-21: Put all place names on the map in Figure 1.

Answer:

Corrected, please see a new version of the figure (Fig. 2 here). The full version of caption of Fig.1: Legs and stations of the ARKTIKA-2018 expedi- tion overlayed on the bathymetry from ETOPO1 "1 Arc-Minute Global Relief Model" " (Amante and Eakins (2009)). CTD stations are shown with white dots. The color indi- cates the number of days since August 1, 2018. The sea ice edge position is indicated with a red dashed line for the beginning (August 21) and with the purple dashed line for the end of the expedition (September 21). The ice edge is based on the sea ice mask provided in the SST DMI product. Numbers indicate positions of 10 oceanographic transects discussed below. The black triangle in the north of the Komsomolets Island shows the Arkticheskiy Cape. The Severnaya Zemlya Archipelago consists mainly of the Komsomolets, the October Revolution, and the Bolshevik Islands (with smaller islands not shown here). The black box indicates the Shokalskiy Strait between the October Revolution and the Bolshevik Islands. The Yana River estuary is situated

southward the Yanskiy Bay (out of the map).

Page 3, Line 27: Did you find a typical error estimate from the comparison with CTD measurements? Answer: Yes, we did it. We added it as well as further information on the vertical profiles in this section to clarify the applicability of the CTD data for satellite validation studies : When calculating a linear regression between the CTD measurements at 6.5 meters depth and the TSG measurements, we obtained a good correlation for both temperature and salinity (correlation coefficients equal to 0.9789 and 0.9656, respectively). The standard error was 0.0231 for temperature and 0.0251 for salinity, and the standard deviation for the difference of measurements (CTD-TSG) was STD_temp = 0.4134, and STD_sal = 0.4296. We applied the obtained linear regression equation to TSG data to obtain adjusted temperature and salinity data.

Finally, to investigate if the TSG measurements can be used to study the surface layer in a highly stratified Laptev sea, we estimlated a summer mixed layer depth following de Boyer Montégut et al. (2004) method based on density and temperature gradient thresholds (Fig.2). The MLD base is found at a depth of the first maximum temperature gradient exceeding (or equal to) a defined (by a threshold) gradient (see de Boyer Montégut et al. (2004)). Using a similar approach, we also computed MLD independently with density, temperature and salinity vertical profiles. The thresholds chosen for the gradients were the following: 0.3 kg/m3 per 1 m for density, 0.2 degrees per 1 m for conservative temperature and 0.2 psu per 1 m for practical salinity gradient. For the MLD calculated from salinity (MLDsal), most (75.17%) vertical profiles had a MLDsal larger than 7 m , with a median value of MLDsal of 11.99 m. As for the temperature-based MLD (MLDtemp), 81.37% of profiles had the MLD larger than 7 m, with a median value of MLDtemp = 13.50 m. Thus, we conclude that in most cases the upper 12 m of the surface layer was homogeneous, and our CTD and TSG measurements can be used for the validation of satellite data.

The full caption for Figure 3: Vertical profiles of conservative temperature (a, b) and practical salinity (c, d) from CTD measurements in the upper ocean layer. Figures (a)

[Figure]

and (c) show all vertical profiles in the upper 50 m, where red stars indicate the mixed layer depth, calculated using de Boyer Montégut et al. (2004) method (see details in the text), colored profiles show the cases, when the MLD is below 7 m depth and gray profiles indicate when the MLD is above 7 m depth. Figures (b) and (c) show the median vertical profiles of temperature and salinity in the 5-100 m layer, respectively, where the shaded area shows the associated STD.

Page 4, Line 2: What interpolation methods were used? Answer: Linear interpolation, added to the text.

Page 4, Line 6: Define AVHRR, MODIS and VIIRS Answer: Corrected

Page 4, Line 8: Define AMSR2 Answer: Corrected

Page 4, Line 11: Your write "(hereafter referred to as "SST DMI")", then make sure you do. Several names are used on this product later in the manuscript (DMI SST L4, the SST field from DMI L4 product, blended DMI product?, temperature estimates provided by DMI, and others). Answer: Corrected

Page 4, Line 11: Define Level 4. Answer: "Level 4 product" means that several swath measurements were interpolated to achieve a regular resolution in time and space

Page 5, Line 7: You write "100-km averaged ship SSS . . ." Is this TSG? Answer: Yes, corrected to clarify to "100-km averaged TSG surface salinity measurements"

Page 5, Lines 12-13: Define SMAP CAP/JLP and SMOS BEC L3, REMSS Answer: Corrected. "However, existing L3 ("Level 3" means a product resampled at a uniform time-spatial grid, different from swath) SSS products: SMAP CAP/JPL (Soil Moisture Active Passive satellite, a product created using the Combined Active Passive algorithm by Jet Propulsion Laboratory) or SMOS BEC (Barcelona Expert Center), are spatially averaged from 60 km to more than 100 km." REMSS (Remote Sensing Systems)

Page 5, Line 16: Define ESA. Answer: ESA (European Space Agency)

[Figure]

Page 5, Line 17: Be consistent with naming of products. Here you use L2 SSS and not SMOS SSS L2. Answer: Corrected. The paragraph describing postprocessing of SMOS SSS L2 data from ESA to the used SMOS SSS L3 product was re-written to clarify the naming (see a new part 2.2.2)

Page 5, Line 18: What do you mean with individual SMOS SSS? Explain how the SMOS SSS are sampled over an Icosahedral Snyder Equal Area grid at 15 km resolution. Is this the interpolation?

Individual SMOS SSS" was mentioned to refer to the radiometric resolution of L1 (Level-1 product, thus brightness temperatures at swath grid) and is an initial resolution for all L2 products, Your text hereindependent from their producer. The SMOS SSS L2 product mentioned in the text is interpolated at ISEA grid by ESA and distributed on this grid. We changed the text as following: "The mean spatial (radiometric) resolution of SMOS product is close to 50 km, but SSS SMOS ESA L2 products are distributed resampled over an Icosahedral Snyder Equal Area (ISEA) grid at 15 km resolution."

Page 5, Line 23: Define ECMWF. Answer: (European Centre for Medium-Range Weather Forecasts)

Page 5, Lines 26-29: Specify the correction with numbers. Also, give the criterion on the ACARD parameter. And what are the typical errors in the weekly SMOS SSS and the individual SMOS SSS? Answer: We add the formula of the correction in the paper. The criterion on the ACARD parameter is : we consider only SSS measurements with an ACARD value over 45. Typically, theoretical error of weekly SSS measurements may reach values under 0.5 pss in the center of the Laptev Sea and reach values higher than 2 pss close from sea ice and in coastal areas. For the individual SMOS SSS the theoretical error is very variable and may reach value higher than 10 pss.

Page 5, Lines 32-34: Where is the river plume? Define and introduce it. How does this affect the weekly SSS error? Answer: The definition of river plume will be introduced later. We changed the sentence to: Concerning the way on how a river plume may

influence the SMOS SSS "A" error : here, a river plume is associated with an increase of SST that induces a theoretical decrease of SMOS SSS "A" error. Nevertheless, river plumes are closer to the coast and SST (used as prior during SMOS SSS retrieval) has a higher error in these areas : these will increase the SMOS SSS "A" error.

Page 6, Line 4: Define all the ice masks provided. Answer: The AMSR2 ice masks were used in addition to the masks provided with every satellite product discussed (SST DMI, SMOS SSS "A", ASCAT (Advanced SCATterometer) winds L3).

Page 6, Line 10: Define ASCAT. Answer: ASCAT (Advanced SCATterometer)

Results. Move all method and technical descriptions to Data and Methods. Only results should be presented here. The results section should have more focus on the SST and SSS distribution, the hydrographic sections, and highlight results to be discussed, not so much on technical details, error analysis and discussion, which I recommend you to move to the data and method section. In general when presenting and discussing the results, please be consistent and distinguishing between the parameters SST/SSS and in situ temperature/salinity. Don't use temperature or salinity on both. Page 6, Lines 24-25: Move to Data and Method. Page 7, Lines 1-7: This paragraph belongs in data and methods. Page 7, Lines 33-34: This belongs in the method description. Answer: Thank-you very much. We have taken these recommendations into account. The whole part 3.1 was moved to section 2: the validation of SST and SSS satellite productsis now presented after the description of the product.

Page 6, Line 22: Please define the river waters. Use plural form if there are several types of river water, if not, use singular form (river water). Answer: Corrected for the singular form. The definition of river water cannot be givenat this point, as we are only later defining the water masses.

Page 7, Line 2: What are "basic statistics"? Answer: Linear regression equation, correlation coefficients, STD. . . Excluded this part from the text as unnecessary.
[Figure]

Page 7, Lines 4-5 & 8-9: How much does the temperature and salinity change in the upper meters? Perhaps show a mean +/- std profile of the upper 10 meters? Answer: The vertical profiles of in situ temperature and salinity were added in the "data" section (new Figure 2). A "mean" vertical profile of temperature or salinity is not representative for the areas where surface temperature varies from +6 to -1.8°C and salinity, from 24 to 34.5.

Page 7, Line 15: In-situ surface layer temperature is then the upper 6.5 m (but not including the uppermost 2? meters typically)? Answer: In-situ surface layer temperatureincludes all measurements from 0 to 6.5 m depth, thus "upper 6.5 m layer". Indeed, the "uppermost 2 meters" are typically not measured, due to how a CTD cast is done and data processing. The postprocessing of the CTD data usually eliminates the top 1-3 meters, because the Rosette with CTD is by itself 1.5 m tall itself and also because surface waves and the close-by ship hull affect the near-top ends of the profiles.

Page 7, Lines 17-19: This is a complicated sentence, so please rewrite. Can this be shown? Answer: Please, see the Figure 3, where the surface CTD measurements are shown. "This value seems to be realistic, and the in situ data justifies it. According to CTD measurements, in average, the 0-3 m water layer is 0.3 degree warmer than the 3-6.5 m layer..."

Page 7, Line 21: What grid? Please describe in data and method, and refer to it. Answer: All products were interpolated on a regular polar stereographic grid (15 km)

Page 7, Lines 28-29: I guess the 15 km SMOS resolution is the interpolated resolution? Please be more precise when naming the different SMOS SSS products, which should all be clearly defined and stated in the data and method section. Answer: Yes, we meant the spatial resolution of interpolated at 15 km ISEA grid, SMOS SSS "A" product. Corrected.

Page 7, Line 31: Use either ARKTIKA-2018 expedition or Akademik Tryoshnikov measurements. Be consistent with the naming of data both in the text and in the figures.

[Figure]

Answer: Corrected to "in situ measurements" here, using ARKTIKA-2018 expedition further

Page 7, Line 33: Again, be consistent with naming of data products, I guess the vessel SSS is the CTD and TSG data from the upper 6.5 m from the ARKTIKA-2018 expedition? Answer: Yes, corrected to "in situ measurements"

Page 7, Lines 33-34: This belongs in the method description. Answer: Corrected

Page 8, Line 1: "SMOS post-processed SSS", what is this product compared to the other named products? Answer: In the revised draft we write (and weearlier meant it) about SMOS SSS "A" - it is always the same product.

Page 9, Lines 2-3: Is the precision so high that you can use three decimals in SST? Answer: The indicated accuracy is arbitrary. It can be reduced to two decimals as in typical SST-validation reports, e.g. http://www.osi-saf.org/sites/default/files/dynamic/page_with_files/file/validation_report_sentinel3_slstr_sst_calval_v1p0.pdf

Page 9, Lines 2-4: Put place names on the map in Figure 1. Answer: Added

Page 9, Lines 9-10: Please define river water somewhere (in data and method?). What is the definition of the river water plume? Answer: We implicitly did it in section 5 and Table 1 when discussing the surface water masses in the Laptev Sea. There was no particular definition of the river plume in this study, as collected in situ data was not covering the freshest waters sufficiently close to river discharge deltas. In general, we meant the furthest (from river deltas) salinity gradient position as a front of "river plume" . It roughly corresponds to the 29 isohaline position (Fig.4). We added the definition of "river plume" in the last paragraph of the Introduction section.

Page 9, Line 10: Standard deviation of what? Answer: "Standard deviation of SST", corrected

Page 10, Line 1: What about the Katanga River Estuary? Answer: Indeed, changed to: "Standard deviation of SST in Fig.\ref{fig: sst-sss-mean-and-std} is the largest in the

[Figure]

Olenekskiy Bay (over 2.5°C), along the coastline close to the Khatanga estuary (2.5-3°C), the Lena River delta (about 4°C) and in marginal ice zone (mostly over 1.5°C)."

Page 10, Line 2: Define the thermal fronts. Answer: Thermal fronts are the areas with largest gradients of temperature.

Page 10, Line 15: "at 125 E, . . ." what latitude? Answer: "At 78-80° N 125°E, free-floating patches of broken ice detached from compact sea ice edge were observed during several weeks in August-September 2018"

Page 10, Line 20: Distribution of freshwater input, is that water with S = 0? Please define the characteristics of this freshwater. Answer: Freshwater is defined in comparison to the "marine water", eg. less than 34.92 as in the studies of (Bauch and Chernyavskaya, 2018) or 34.80, as defined by Aagard et al., 1989. As the 0-salinity river water quickly mixes with saltier marine water, in reality the "freshwater" we consider is more "brackish" than "fresh". Nevertheless, for the simplicity assuming a river plume front at 29 isohaline, the "freshwater" corresponds to all water masses with a salinity lower than 29.

Page 10, Line 21: Please refer to Figure 1. What Section number or longitude/latitude limits? Answer: Virtual meridional section corresponds to section 5, and the virtual zonal section does not correspond to any real oceanographic sections. The maps of the virtual sections positions were added to both Hovmöller diagrams.

Page 10, Line 21: Temporal evolution of what? Introduce Figure 6 before Figure 7. Answer: Corrected to : "To evaluate the distribution of freshwater input in the Laptev Sea in August-September 2018, we considered virtual zonal and meridional transects along 78° N and 126° E, respectively, and plotted the temporal evolution of SST DMI, SSS SMOS "A", wind speed and SLP in Hovmöller diagrams"

Page 10, Line 25: "shelf break" instead of "edge of the shelf"? Answer: Corrected

Page 10, Line 29: What was the wind direction during these series of cyclone pas-

sages? Answer: Below, there is an initial version of Fig. 5 with the wind speed and wind directions instead of SLP. We found it too complex to present in paper, so have chosen the pressure for clarity. The wind directions had 0-90° azimuth (north-east in "oceanographic convention").

Page 10, Line 30: Change to ". . . salinity increased by 1 . . ." Answer: Corrected

Page 10, Line 34: Change to ". . . mixing event induced by the wind." Answer: Corrected

Page 10, Lines 30-34: How does this relate to any river discharge data from the same period? Are there any model data to compare with? Answer: It is possible that a small peak observed in the Lena River discharge in the very beginning of September (Fig. 1) contributed to an additional portion of fresh water that arrived at 78 N latitude in several weeks.

Section 4.1.1 Change to something like "Water from the Kara Sea". You have not defined Kara Sea waters anywhere in the text. Answer: Corrected the title. The Kara Sea receives the two other large Siberian Rivers, the Ob' and the Yenisey, and thus presents a low salinity compared to the central Arctic Basin. In the absence of significant river sources on the Severnaya Zemlya Archipelago, we considered that the freshwater input close to the Vilkitskiy and the Shokalskiy Straits, arrives from the Kara Sea.

Page 11, Lines 6-7: The temperature is decreasing? Answer: We suppose so, but the SST of the Kara Sea should be additionally studied.

Page 11, Line 7: Where and when has the exchange with the atmosphere taken place? Is the atmosphere colder here than in the central Laptev Sea? Answer: The comparison of the Kara and the Laptev seas summer conditions should be additionally studied and can become a subject of another paper. Nevertheless, we can suppose that a passage through the Vilkitskiy Strait diminish the temperature of arriving waters. The

[Figure]

Severnaya Zemlya Archipelago is known for its icebergs (thus, the Kara Water heat might be lost in their vicinity). There is also a system of countercurrents in the Vilkitskiy Stait. Together with tidal currents and steep topography it creates turbulent instability, and a loss of energy.

Page 13, Line 1: You write ". . . significantly greater than . . ." How can you see the relative amount of freshwater from one snapshot and with different timing on each section occupation? Answer: There is no other possibility to have simultaneous measurements of freshwater amount than moorings or a dedicated campaign with several ships. The only mooring installed in the studies in this section area, is situated northward of the Bolshevik Island. Another solution is numerical modeling, but it is out of scope of this study, and in any case, without good initial and boundary conditions (meaning good measurements of all river discharges, estimated of glaciers melts, etc), the model will have some difficulties to reproduce the freshwater budget very accurately. Indeed, in this paragraph we compare the quantities measured in situ with a time difference of 4 weeks between section 1 (northward of the Arkticheskiy Cape) and section 10 in the Shokalskiy Strait. Nevertheless, the satellite images as the only source of continuous information on the surface layer, do not show any significant inflow of freshwater northward of the Severnaya Zemlya Archipelago.

Page 13, Lines 2-5: Show place names on map in Figure 1. Also, define the hydrographic sections in data and methods with time, wind condition, and air temperature during their occupations. Then stick to the named Sections (Section numbers) later in the text. Answer: The names of hydrographic sections were indicated both in Fig.1 and Fig.6 of the previous version of the manuscript. In the paragraph that you mention, we mention their geographical distribution. As for the conditions during every section, it can be found in the cruise report on the NABOS web-site https://uaf-iarc.org/wp-content/uploads/2019/09/NABOS-2018_report.pdf The wind speed and the air temperature during the cruise are shown in Fig. 6.

Page 13, Line 7: 1024-1027 kg m-3? Answer: Yes, corrected

[Figure]

Page 13, Line 14: What happens during cyclone passages? Clouds? Have you discussed this effect in Data and Methods? If yes, refer to it. Answer: Yes, as it was mentioned in the DMI SST-product description, clouds are opaque for the IR measurements, and the DMI SST dataset is blended from the IR measurements.

Page 13, Line 15: Can you provide any T and S limits? Answer: Yes, corrected Page 13, Line 17: By depression passage you mean low-pressure or cyclone passage? Answer: We call a low-pressure system a cyclone, and a high-pressure system an anticyclone

Page 14, Lines 5-7: This belongs in Data and Methods. Answer: "Oceanographic sections are used to estimate a thickness of the freshwater layer and how far the river water propagates under the ice. Section 5 provides complementary information to the meridional Hovmöller diagram (Fig.8, upper row) as it was done along the same 126E parallel from 76 to 81.4 N on September 1-4 2018." We do not provide the figures for all 10 oceanographic sections but do it to illustrate some particular processes or phenomena, so we introduce the largest meridional section in the discussion of meridional Hovmöller section.

Page 14, Line 9: Absolute salinity and conservative temperature are shown in Figure 8. Either change the text to absolute salinity and unit g/kg or change to practical salinity in figure. Answer: Corrected to practical salinity

Page 14, Line 11: You refer to different observations at specific latitudes in the figure, but the figure is presented with distance in km, so please refer to km as well. This applies for the other sections as well. Answer: Corrected the figure: added the latitude axis.

Page 14, Lines 12-16: What about melting under the sea ice due to the presence of warmer (above freezing point temperature) river water? This will cool the river water and still keep the water under the ice relatively fresh. Are there any $\delta 18O$ data (or other tracers) to be able to identify the source of this water? Answer: Yes, there is. We added

a section on the isotopes of the oxygen 18 to the Discussion part. The salinity/delta-18O scatter plot for the surface water samples is presented in Fig. 7. Using a three-component model (marine water / river water (meteoric water) / sea ice melt water) described in Bauch and Chernyavskaya, 2018 and the results of isotopes analysis from surface water samples, we calculated the fractions of each water. The isotopes analysis revealed that the most important fraction of river waters was transported to the shelf and continental edge of the East-Siberian Sea. At the same time, the water samples at the northern part of the 126°E section consist of 10-15 % of river water and only 0-5% by the sea ice melt. (Fig.7)

The full caption of Fig.7: Fractions of river water (a), sea ice melt water (b) and marine water (c), calculated using d-O18 measurements and Bauch andCherniavskaia (2018) 3-components model of freshwater balance. A thin black line shows the position of sea ice edge on August 31, 2018,when the northern stations of the meridional (5) section along 126E were done in the MIZ, and the blue line shows the sea ice edge onSeptember 16, 2020, when the ARKTIKA-2018 expedition was working in the MIZ of the East-Siberian Sea. Please, note that the colorbar scale is different for each water fraction

Page 14, Line 15: You write: "The heat exchange with the sea ice might be more effective than with the atmosphere, . . ." Why? Answer: "The heat exchange with the sea ice might be more effective than with the atmosphere," because it is a transfer of energy from a liquid to a solid body, which is more effective than to the gas. At the same time, it depends on thermal conductivity in the ice, and its initial temperature profile, so this question needs a special attention.

Page 14, Line 19: "Below the pycnoclines, . . ." to what depth? Answer: To the depth of 25 meters, which comes from p.14, lines 18-19

Page 15, Line 1: The 34.5 isohaline is not shown in the figure. Answer: Corrected

Page 15, Lines 2-3: What are the T and S characteristics of AW in this region? I don't

agree that AW is seen at 100-120 m depth in Figure 8? Why is there instability in the AW layer? Answer: AW's main characteristics in this region is its positive temperature (Pnyushkov et al.2018, Section 2.1). In oceanographic section 5, the 0°C isotherm is situated at the depth 100-150 m, so we conclude that AW is situated below. An apparent weak "instability" is due to the colored representation.

Page 15, Line 5: Define all hydrographic sections in Data and Method (see earlier comment on this). Answer: The names of the different hydrographic sections were indicated in Fig.1 of the previous version of the manuscript.

Page 15, Line 11: You write: ". . . which is clearly seen in temperature signal that is negative even close to the surface." Is the temperature still above the freezing point, i.e. any melting under the ice? The temperature is above the freezing point in most of cases. To obtain a more detailed information on the sea ice formation/melting during the ARKTIKA-2018 expedition, please see the Fig. 7.

Page 15, Lines 25-26: The river discharge information should be introduced in the introduction. Answer: Corrected

Section 4.1.2 Some more background information is needed in the beginning of this section. It is also an analysis method and should be moved to the data and method section. The surface fronts in question should be defined. To get the Ekman transport, you need to integrate over the Ekman depth, what is the Ekman depth in this region? Assumptions made by Ekman were no boundaries, infinitely deep water, and no geostrophic flow. How are these assumptions met in this region? What happens when you have boundaries (coastlines)?

For the Ekman transport we assume that there is a layer with no vertical momentum flux, and that it is not the sea bottom (no friction at the ocean floor). Of course, coastlines are discontinuities, where the approach will not work, and close to them (and in the very shallow areas), the assumption of no bottom friction does not hold either.

[Figure]

Page 16, Line 8: Give the reference to TEOS-2010 and show the formula for the drag coefficient CD. For the wind speed below 10 m/s : u* = 0.051*Uw - 0.14 for stronger winds, u* = 0.051*(Uw-8) + 0.27, where Uw is a wind speed (from FOREMAN AND EMEIS, 2014 http://www.atmo.arizona.edu/students/courselinks/fall10/atmo551a/DragCoef_2010jpo4420%252E1.pdf)

Page 16, Line 10: Use Ekman pumping instead of vertical Ekman speed? Answer: Corrected

Page 17, Line 12: You write ". . . mixing of different water masses, . . ." What are these water masses? You should introduce the water masses in the region in the introduction section. The common names of the surface water masses do not exist, as there was no previous study at this temporal scale to define them. The most well-known water masses are defined in the new version of the Introduction.

Section 5 Perhaps start the results with this section? Answer: We aggree, so this section was moved to the beginning of Results part

Page 17, Line 17: And below 200 m? Answer: We discuss only the upper 200 m in this Figure.

Page 17, Line 23: Specify which of the Arctic Ocean waters that quickly change. Answer: Corrected to "surface water"

Page 17 Line 24: Add ". . ., and the synoptic satellite data provide . . ." Answer: Corrected

Page 17, Line 32: The "classical" water masses, do they have a name? Answer: Corrected to the "main surface water masses"

Page 18, Line 1: Add "... larger range than the near-surface (upper 6.5 m) in situ measurements . . ." Answer: Corrected

Page 18, Line 7: A recognisable river front or a river plume front? Define. Answer:

[Figure]

Corrected to "river plume front"

Page 18; Line 8: 145 E? How do you define the sea edge? Is it a specific sea ice concentration limit? Answer: It is a typo. Corrected to "sea ice edge", which is defined with 1% sea ice concentration from AMRS2 SIC product.

Page 18, Line 11: You write ". . . captured under the ice and exposed back." How or when? What about melting sea ice from below? Answer: We added a section on oxygen isotopes that clarifies this point.

Page 18, Lines 21-22: Show place names on map in Figure 1. Answer: Corrected

Page 18, Lines 24-25: There is also a high density within the range 22.5-30 and 3-4 Answer: Corrected

Discussion and conclusion. This section provides little discussion or conclusions, mostly summary. There is more discussion in between the presentation of the results in the Results section. Answer: Corrected, please see an improved version of this section.

Page 19, Lines 11-12: You write "The fresh waters displacement was associated with the Ekman transport." How? This is not well presented or described in the results. Answer: Corrected. Please see a new paragraph on "wind forcing".

Page 19, Lines 15-16: You write ". . . and there is no evidence that the sea ice melting itself can create such a considerable layer of freshwater." How or where have you presented this lack of evidence? Answer: We added a section on oxygen isotopes that clarifies this point.

Page 19, Lines 18-20: What about river water melting sea ice from below? Are there any $\delta$18O data (or other tracers) to evaluate this? Answer: Please, see a new section on oxygen isotopes that clarifies this point.

Page 19, Line 24: Add "Calculated monthly Ekman pumping indicates the area of most

intense mixing processes induced by wind."? Answer: Corrected

Figures with more than one panel should be labelled with a), b), c) etc. and then properly referred to in the figure captions and the text when the figures are described. Also use the same font size in the figures with several panels. Remove titles on figures and add this text in the figure caption instead.

Figure 1: What is the definition of the sea ice edges? Add all place names mentioned in the text on the map. Answer: Corrected. Here the ice edge is based on the sea ice mask provided in the SST DMI product

Figure 2 and 3: Last sentences in the figure captions: this information should be provided in the data and method section. Answer: Corrected.

Figure 4: Missing numbers on y-axis in lower left panel. Use white isolines below 0 degree Celsius/28 PSS? It is hard to see black lines in the dark background. This is also the case for Figures 6 and 8.

Figures 5 and 7: Where is the data from? What bathymetry data is used? Answer: Corrected to "Hovmöller diagram of SST (a), SSS (b), wind speed (c), and sea level pressure (d) for the zonal transect at 78N. Small circles at SST and SSS diagrams show in situ measurements of temperature and salinity (first CTD of TSG at 6.5 m). Sea ice concentration (AMSR2) is indicated with a blue color, see Fig.5 for the color scale. The depth profile along the transect (e) is extracted from "1 Arc-Minute Global Relief Model" (Amante and Eakins (2009). The position of a virtual transect is shown at SST SMI and SSS SMOS "A" maps for August 26, 2018 (f, g)."

Figure 8: Why do you show conservative temperature and absolute salinity in this figure and not in Figure 6? At least use proper units when referring to Figure 8 in the text. Answer: Corrected to practical salinity. These two figures explain transformation of different water masses. Fig.6 illustrated the processes of Kara and Laptev waters mixing in the western part of the Laptev Sea, and Fig. 8 shows the propagation of river

[Figure]

waters above the shelf and the shelf break.

Figure 9: Increase the fonts. The two upper rows are not described in the text. What are the arrows in the lowest row, and why is the resolution higher than in the row above? Write units in the figure caption. Where is the data from?

Figure 10: Add the freezing point temperature line. Remove the figure title.

Figure 11: What are the boxes? Are the lines the freezing point temperature line? Answer: The boxes show the cores of 6 water masses described in text. Red line show the freezing point.

Figure 12: Put the dates inside the plots instead.

Figure A1: Remove the titles from the figures. Write units in the figure caption. Increase the fonts.

Answer: Corrected in most of cases.

Tables. Table 1: Add more information to the table caption. Answer: Corrected
* * *
[Figure]

[Figure]

**Fig. 1.** The Lena River discharge in 2018, data from Arctic GRO dataset (https://www.arcticrivers.org/data)
**Fig. 2.** Legs and stations of the ARKTIKA-2018 expedition overlaid on the bathymetry

[Figure]

**Fig. 3.** Vertical profiles of temperature (a) and salinity (b) from CTD measurements in the upper 50 meters. The red stars indicate MLD, colored profiles have MLD larger than 7 m, gray - smaller than 7 m.

[Figure]

SMOS-L3A-2018-08-31-v9-acard-weekly-large-swath.nc

**Fig. 4.** Gradient of salinity for August 31, 2018 calculated from SSS SMOS "A" (see the description of product in a new version of manuscript), [psu/km]

[Figure]

[Figure]

**Fig. 5.** Hovmoller diagram for zonal virtual section along 78 N: SST DMI (upper left), SSS SMOS "A" (lower left), ASCAT wind speed (upper right) and direction (lower right).

[Figure]

a)

[Figure]

b)

**Fig. 6.** Wind speed and air temperature measured in the ARKTIKA-2018 expedition (from August 17 to 26 of September).

[Figure]

[Figure]

**Fig. 7.** Fractions of river water (a), sea ice melt water (b) and marine water (c), calculated using d-O18 measurements and Bauch andCherniavskaia (2018) 3-components model of freshwater balance

[Figure]

Ocean Sci. Discuss.,
https://doi.org/10.5194/os-2019-60-RC3, 2020

[Figure]

The presented observations of surface freshwater distribution is from an interesting area of the Arctic Ocean. The observations are presented in a nice manner. And there is plenty of figures, with many detailed results. This is all fine.

There is not much available knowledge on how the river-water spreads north along the shallow Siberian shelves, so this paper is potentially a significant contribution in that regard. Beside some issues already noted by the other reviewers, like language, I have two larger issue that made me tick the "major" box here.

My "major" science concern is the contribution from sea-ice melt. I think you need to do a somewhat better job at addressing this possible contribution. It is difficult, but it

can potentially explain much of the freshwater available. If delta-18 O samples where available, one could differ between these to sources perhaps, although I think much of the river-water from previous summers probably freeze-up, and some of the newly melted sea ice could thus just be old river water.

I also find that you mix Results and Discussion in the 3. Results section. This may be OK, but then you need to call this 3. Results and Discussion. And then the sections 4.X are somewhat also Results and discussion. And then you cannot have "Discussion and Conclusion" in section 6. See? A re-struction is needed. There is also no general conclusion drawn on what actually spreads the freshwater to the north. Is this wind driven mostly? Or not?

I also have a general suggestions: Provide the spatial mean vertical profile down to 100m for T and S, and use that to describe the mean stratification. THEN – AFTER-WARDS, you can present, the spatial and temporal changes from this mean profile.

Minor issues:

Abstract: Your main explanation for how the river water is transported out from the river mouth area should be lifted up into the abstrat. Is this all wind-driven?

We use the term "ice-free" not "free of ice" in general.

Page 3, Line 7: Add what products is "validated".

Page 5. Last line. "Ice sheet" is used for the large inland thick glaciated areas in Antarctica and Greenland. You probably mean "sea-ice" here??? Page 10 and other places. Is it ok to use PSS as salinity unit? Should it not be absolute salinity, or unit less?

Figure 5: Please provide the section in a map.

Page 16: 4.1.2. This section is really about "Wind Forcing" and nothing else. So it should have this name, and not "Mean Monthly Observations". The Ekman equations

[Figure]

should be in the method section.

Table 1: Use other "names" than 1-6. Cold and Warm, Fresh, Salty and Medium Salinity? SO 1=WF, 2=WMS, for example... (Numbers are more difficult to remember than names. . ..)

Page 18. Line 20: Why? Mixing with saltier water below? Or sea ice formation. When is the first onset of freezing? And what is the "normal/mean" for this freeze-up?

Page 19. Line 15-16. While there is "no evidence that "sea-ice melting can create such a layer of freshwater" – is there evidence that it can not? This is my major point #1.

Conclusion: The evaluation is OK, and described nicely. But what is the main message? What is learned of the river water flow? This still needs to be described. Main point#2.
* * *
[Figure]

Ocean Sci. Discuss.,
https://doi.org/10.5194/os-2019-60-SC1, 2019

[Figure]

Review of "Surface waters properties in the Laptev and the East-Siberian Seas in summer 2018 from in situ and satellite data" by Anastasiia Tarasenko et al. Submitted to: Ocean Science Manuscript number: 2019-60

Summary: In this paper the authors focus on distribution of surface water masses based on satellite salinity and temperature and in situ measurements at the shelf of the Laptev and East-Siberian seas. They report several features registered by satellite and in situ data in the surface layer, namely, variability of surface salinity and temperature in August-September 2018, inflow of freshwater from the Kara Sea to the Laptev Sea,

seasonal cooling of surface water in autumn. The paper is interesting insofar as the authors focus on the Arctic seas which hydrological structure and dynamics remain largely unstudied. Thus, the topic addressed in this manuscript is of great scientific and practical interest. Despite my enthusiasm for the topic, I don't think this article is ready to be published in Ocean Science due to several drawbacks of this work. Generally, this article seems like a cruise report, it describes structure of SSS and SST in the Kara Sea to the Laptev Sea, but lacks scientific novelty and new insights into processes in the study area. I recommend the authors to improve their study by providing more thorough analysis of in situ data including vertical profiles and to focus on certain processes that occur at the shelf of the Laptev and East-Siberian seas, rather that providing brief description of multiple processes.

General comments: 1. The authors define the plume as water mass with salinity less than 30 (e.g., page 10, line 28). However, the majority of works that deal with river plumes in different World coastal areas define river plumes as relatively shallow surface-advected water masses bounded with large salinity gradient at their border with ambient sea. Existence of this salinity gradient determines significantly different dynamics of river plumes (governed by buoyancy force), as compared to ambient sea, which is the main reason to distinguish river plumes as individual water masses. River plumes formed at the shelf of the Laptev and East-Siberian seas generally have sharp salinity gradient at isohalines of 15-25, while water masses with greater salinity are regarded as ambient shelf water. Thus, I recommend the authors to determine salinity border of the Lena plume based on maximal salinity gradient and to distinguish wind-driven dynamics and variability of river plumes and more "typical ocean dynamics" of shelf water mass. 2. In this study you use SSS data from SMOS satellite which spatial resolution is 50 km (page 5, line 18). However, you deal with salinity maps with 15 km spatial resolution (page 5, line 18). Did you reduce spatial resolution only by reprojection? 3. In this study you deal with in situ thermohaline data obtained from the depth of 6.5 m (page 7, line 3). However, salinity at this depth can be significantly different from surface salinity (even more than several units) especially within the river

plumes. Thus, your usage of this data to compare and validate satellite data require additional proof, e.g., based on vertical thermohaline measurements. 4. The results of validation of satellite SSS and in situ salinity obtained from the depth of 6.5 m does not seem convincing, especially at the areas influenced by freshwater discharge (Section 3.1.2 and Figure 3). We see underestimation of salinity by several units for almost all measurements. I recommend authors to deal with salinity gradients rather than absolute salinity values, e.g., to show that satellite SSS data reproduces well shows relative salinity differences if it really does. 5. Ranges of temperature and salinity values used to determine different water masses at the study area are heuristic and are not based on any precise idea (Section 5 and Table 1). What is the reason to select T = 3 °C and S = 25 and 29 as borders between water masses? Why you determine 6 water masses? Why not to determine 5 or 7? 6. Reconstructed circulation in the Laptev and East-Siberian sea based on Ekman theory does not seem convincing, especially the presented patchy distribution of upwelling and downwelling areas (Section 4.1.2 and Figure 9e). These results have to be supported by in situ measurements and/or numerical modelling. 7. Propagation of freshened water from the Kara Sea and its presumed missing with Lena plume in the Olenekskiy bay requires additional proof by in situ measurements and/or numerical modelling (page 19, lines 6-9). The role of the Khatanga plume in this process (as well as the plume of the Olenyok River surprisingly not mentioned here) also should be supported by additional data.

———————————————

[Figure]

Ocean Sci. Discuss.,
https://doi.org/10.5194/os-2019-60-AC1, 2020

1. The authors define the plume as water mass with salinity less than 30 (e.g., page 10, line 28). However, the majority of works that deal with river plumes in different World coastal areas define river plumes as relatively shallow surface-advected water masses bounded with large salinity gradient at their border with ambient sea. Existence of this salinity gradient determines significantly different dynamics of river plumes (governed by buoyancy force), as compared to ambient sea, which is the main reason to distinguish river plumes as individual water masses. River plumes formed at the shelf of the

[Figure]

Laptev and East-Siberian seas generally have sharp salinity gradients at isohalines of 15-25, while water masses with greater salinity are regarded as ambient shelf water. Thus, I recommend the authors to determine salinity border of the Lena plume based on maximal salinity gradient and to distinguish wind- driven dynamics and variability of river plumes and more "typical ocean dynamics" of shelf water mass.

Answer:

When we discuss the river waters (and the "river plume"), we mean some water mass with a river-origin, already transformed to some mix of river and sea waters (not taking into account precipitation and brine from sea ice formation and melting now, as their substantially smaller contribution cannot be estimated with current satellite data). As in other studies that mean that the definition of "river plume" or "river water" becomes arbitrary, depending on the authors of study and the region of interest. Before writing this paper, we regarded the gradients of salinity as well, and found that the highest values of salinity gradients roughly correspond to the position of 29 isohaline. In Fig. 1, there are high gradients of salinity at 15, 20, 25, and 29 isohalines. Knowing that the freshwater input in this area comes mainly from the rivers, as we demonstrate it later with the oxygen isotope analysis (please, see a new version of manuscript), we have just chosen the furthest position of "river water presence" as a virtual "river plume".

2. In this study you use SSS data from SMOS satellite which spatial resolution is 50 km (page 5, line 18). However, you deal with salinity maps with 15 km spatial resolution (page 5, line 18). Did you reduce spatial resolution only by reprojection?

Answer:

The "initial" SMOS instrument resolution is 50 km (which we meant to explain in 2.2.2, line 17-18), but the SMOS SSS product distributed by ESA is already oversampled in the ISEA grid with a resolution of 15 km. In other words, the spatial resolution of SMOS SSS Level 2 v662 product is 15 km, we just resampled all satellite products at the same grid for convenience. This "oversampling" of SMOS SSS at 15 km is practical

for two reasons. First, to conserve the real salinity gradients observed with in situ measurements and not smooth them to 50 km for the further comparison with SMOS SSS. The spatial resolution of ship measurements depends on the ship speed (8 knots $\sim$ 3 m/s), pumping speed (16 l/s) and the CTD measurement frequency (24 Hz), and is of order O(1) m. After processing the raw data, its resolution is O(250) m. A 7.5-km in situ measurement average corresponds to 30 minutes of TSG measurement, as we mention at line 6, p.3.1.1 (and 15-km pixel represents one hour of measurements). Second, to put SSS on the same grid as rather high-resolution SST for the further calculations, e.g., density.

3. In this study you deal with in situ thermohaline data obtained from the depth of 6.5 m (page 7, line 3). However, salinity at this depth can be significantly different from surface salinity (even more than several units) especially within the river plumes. Thus, your usage of this data to compare and validate satellite data requires additional proof, e.g., based on vertical thermohaline measurements.

Answer: To investigate if the TSG measurements can be used to study the surface layer in a highly stratified Laptev sea, we calculated a summer mixed layer depth following de Boyer Montégut et al. (2004) method based on density and temperature gradient thresholds (Fig.2, 3: Colored profiles show the cases when the MLD is below 7 m depth and gray profiles indicate when the MLD is above 7 m depth). The MLD is found at a depth of the first maximum temperature gradient below a depth of defined (by given threshold) density gradient (see de Boyer Montégut et al. (2004) for details). Using the same logic, we computed MLD also with density and salinity vertical profiles. The threshold chosen for practical density gradient was 0.3, and 0.2 units for conservative temperature and practical salinity gradients. Regarding the MLD calculated from salinity (MLDsal), most of the measured vertical profiles had the MLDsal below 7 m depth 75.17%. The median value of MLDsal was 11.99 m. As for the temperature (MLDtemp), 81.37% of profiles had the MLD below 7 m depth, with a median value of MLDtemp = 13.50 m. Thus, we conclude that in most of the cases the upper 12 m

of the surface layer was homogeneous, and our CTD and TSG measurements can be used for the validation of satellite data.

4. The results of validation of satellite SSS and in situ salinity obtained from the depth of 6.5 m does not seem convincing, especially at the areas influenced by freshwater discharge (Section 3.1.2 and Figure 3). We see underestimation of salinity by several units for almost all measurements. I recommend authors to deal with salinity gradients rather than absolute salinity values, e.g., to show that satellite SSS data reproduces well and shows relative salinity differences if it really does.

Answer: Indeed, both SST and SSS compared to in situ measurements have some biases. Nevertheless, we consider that the bias (or "mean of difference" between SSS and in situ data) is close to be linear, so we extract it from the initial dataset. This study was a case study for a particular period of time and a particular region, and one of the important milestones of this work was a prototype of the "Arctic SSS" product created for it. The quality of this SSS product was the best for the Arctic at the moment of writing the manuscript (we compared it with others, such as other SMOS or SMAP SSS, but didn't present). Later on, we created a first version of SMOS SSS for the whole Arctic (Supply Alexandre, Boutin Jacqueline, Vergely Jean-Luc, Kolodziejczyk Nicolas, Reverdin Gilles, Reul Nicolas, Tarasenko Anastasiia (2020). SMOS ARCTIC SSS L3 V1.0 maps produced by CATDS CEC LOCEAN. SEANOE. https://doi.org/10.17882/71909. It can be check it here: https://www.seanoe.org/data/00607/71909/ ). To convince that the resulting SSS (before additional filtering of MIZ areas) resembles the measurements of TSG, we propose another graphical form of comparison: the timeseries of SSS and TSG-measured salinity in Fig. 4. The largest difference between SSS and in situ salinity is observed when the ship was working in MIZ areas (beginning of September and September 13-17). These "ice" pixels were filtered afterwards.

As for the high error at 74°N 136°E, we suppose, it is the effect of the Stolbovoy Island present there. In the section with SSS description we mention that the closeness

to the coastline deteriorates the quality of SSS estimates. To avoid the "coastline" contamination we applied a 25-km mask around all islands and coasts.

5. Ranges of temperature and salinity values used to determine different water masses at the study area are heuristic and are not based on any precise idea (Section 5 and Table 1). What is the reason to select T = 3°C and S = 25 and 29 as borders between water masses? Why you determine 6 water masses? Why not to determine 5 or 7?

Answer: This section is based on the same principles as a classic T-S analysis, with the main difference that the amount of available points when use SST-SSS diagram is several orders higher than from classical CTD measurements. Identifying the core of each water mass stays heuristic in both cases. A chosen number of water masses corresponds to the number of "water mass cores", defined by the density of points on the SST-SSS diagram.

6. Reconstructed circulation in the Laptev and East-Siberian sea based on Ekman theory does not seem convincing, especially the presented patchy distribution of upwelling and downwelling areas (Section 4.1.2 and Figure 9e). These results have to be supported by in situ measurements and/or numerical modelling. Answer: The Ekman transport was recalculated for the updated version of manuscript using the recommendations of all reviews (Fig. 5, 6). Numerical modeling is out of scope of this study.

7. Propagation of freshened water from the Kara Sea and its presumed missing with Lena plume in the Olenekskiy bay requires additional proof by in situ measurements and/or numerical modelling (page 19, lines 6-9). The role of the Khatanga plume in this process (as well as the plume of the Olenyok River surprisingly not mentioned here) also should be supported by additional data. Answer: Unfortunately, no in situ measurements were carried out in the coastal area during the period of our study neither by the expedition Arktika-2018, nor by any other expedition to the best of our knowledge. The results of numerical modeling should be validated separately, so we consider satellite data the only available source of information. The Kara-origin of the

freshwater appearing in the Vilkitskiy Strait and northward is well-seen on the time series of SSS fields. A special role of the Khatanga, the Anabar, the Olenyok, and the Yana Rivers should be studied additionally. It was not discussed in this paper, because we had neither the appropriate information on their discharge, nor in situ/satellite measurements in the close vicinity of their estuaries/deltas.

[Figure]

SMOS-L3A-2018-08-31-v9-acard-weekly-large-swath.nc

**Fig. 1.** Gradient of salinity for August 31, 2018 calculated from SSS SMOS "A" (see the description of product in a new version of manuscript), [psu/km]

[Figure]

**Upper layer temperature, CTD from ARKTIKA-2018**

**Fig. 2.** Vertical profiles of conservative temperature from CTD measurements in the upper 50 meters. Red stars indicate the mixed layer depth, calculated using de Boyer Montégut et al. (2004) method

[Figure]

**Upper layer salinity, CTD from ARKTIKA-2018**

**Fig. 3.** Vertical profiles of practical salinity (b) from CTD measurements in the upper 50 meters. Red stars indicate the mixed layer depth, calculated using de Boyer Montégut et al. (2004) method

[Figure]

[Figure]

**Fig. 4.** The time series of TSG-measured salinity (black line) and collocated SMOS SSS measurements.

[Figure]

**Fig. 5.** Ekman pumping (in color) and horizontal transport for August 2018 calculated from satellite data

[Figure]

**Fig. 6.** Ekman pumping (in color) and horizontal transport for September 2018 calculated from satellite data

[revised manuscript text omitted]

---

## Author Response (AR2)

*Comments to the Author:*
*Dear Authors*
*Thank-you for your much-revised manuscript. I intend to send it back to the two referees who were willing to review a revised manuscript. However, I have many detailed comments of my own (see below), generally intended to make it clearer for the reader. I think these are best attended to before the referees review it.*
*Yours sincerely*
*John Huthnance (editor)*

Line 2. "then completed" -> "complemented"?
Answer: Corrected

Line 10. "which" -> "as" (only use "which" immediately after the object it refers to)
Answer: Corrected

Line 11. Omit "that it offers"
Answer: Corrected

Line 17. "should be considered with attention" -> "merits attention"?
Answer: Corrected

Lines 19-20. "a decrease in freshwater content of about 180km^3" – from when to when?
Answer: Over 2003-2014 period

Line 2. "500 km^3" – over what time period? Are these freshwater volumes on the basis of salinity = 0?
Answer:
500 km^3 is the volume of exchange between the Laptev and the East-Siberian Seas and central Arctic ocean with anticyclonic atmospheric vorticity conditions, it was calculated from 1920-2005 hydrographic measurements on "quasi-decadal timescales" based on the estimates of freshwater content anomaly and thus, mean salinity values

Lines 3-4. "income in" -> "coming into"
Answer: Corrected

Lines 8-9. "The shelf area of the Laptev and the East-Siberian was described" -> "The Laptev Sea and East-Siberian shelf areas were described"
Answer: Corrected

Line 10. "the freshwater income" -> "the incoming freshwater"
Answer: Corrected

Line 21. "falls under an additional impact" -> "comes under the additional influence"
Answer: Corrected

Line 29. "via" needs to be replaced by some characteristic of the vertical distribution of energy that shows the extent of the mixed layer.
Answer:

The note on the energy was excluded as I do not remember the details, and the book where it comes from has only a paper version and is situated in another country now, so unavailable with coronavirus situation.

Lines 30-31. "open water or under the ice, the Barents Sea, the East-Siberian Sea or the central Arctic"; this list is only useful if you associate different MLD with each region.
Answer:
Added details on the region and time of measurement for all citations

Line 7. "water, which" -> "water; 34.80"
Answer: Corrected

Line 10. "being" -> "to be"
Answer: Corrected

Line 15. "layer between" -> "layer to be between" or "layer as between"
Answer: Corrected

Line 18. Omit "The" at start.
Answer: Corrected

Line 29. Omit first "the".
Answer: Corrected

Line 33. "and" -> "over a"?
Answer: Corrected

Line 1. "net radiative balance at the sea surface changes from 100W/m^2" – in which direction is the 100W/m^2 ?
Answer: Positive values mean from the atmosphere to the ocean and correspond to the warming of sea water in summer

Lines 3-12. This is the place (end of Introduction) to state clearly what your paper will do to improve our knowledge and understanding. That statement is rather hidden by, and mixed up with, too much information about how you will do it. Such information can be in section 2.
Answer:
Thank you for your suggestions! We moved the line 8-12 to the second section, Data and methods.

Line 18. Omit "both,"
Answer: Corrected

Page 5 line 6. Omit "further on"?
Answer: Corrected

Page 6 line 4. "The threshold chosen for practical density gradient was 0.3 degree per 1 m" does not read sensibly. "0.3 degree per 1 m" relates to temperature (your text says so) so how does the density relate to the threshold?
Answer: It is a typo, thank you. 0.3 relates to density, corrected the units to kg/m³ per 1 m.

Page 6 bottom – page 7 top. "the upper 12 m of the surface layer was homogeneous, and our CTD

and TSG measurements can be used for the validation of satellite data." I think 12m homogeneous layer validates the TSG at 6.5m (a referee question) but the satellite data may have a skin effect and/or diurnal variation depending on the weather (you discuss this on pages 9-10).

Answer: These effects are supposed to be filtered out when a product L4 is proposed. We can still confront some non-filtered effects, indeed.

Line 3. ". . Below 40m depth the temperature . .". It may be better to give the temperature at 30m depth; at present you describe the temperature as "0.5∘C . . at 5 m . . slightly rising to -1 ∘C"

Answer:
 I suppose that "40 m" in this question might be a typo. 40 m is the depth of the smallest STD of temperature. Added to the text the following precision:
We observe rather cold (0.5° C) and fresh (30.5) water at 5 m, followed by a smooth thermo- and halocline down to 30 m depth (with a temperature of -1.3° C and salinity of 33.8)

Line 5. Omit "up"
Answer: Corrected

Line 20. "Since" -> "After"
Answer: Corrected

Line 23. "Siberian great" -> "large Siberian"
Answer: Corrected

Line 25. "moment" -> "time".
Answer: Corrected

Line 29. "listed below products" -> "products listed below"
Answer: Corrected

Line 4. "provide" -> "providing"
Answer: Corrected

Line 7. "uses" -> "used"
Answer: Corrected

Lines 18 and 19. "collocated" or "co-located"? I think maybe "collocated" in line 18 (also line 21 and page 9 line 2) but "co-located" in line 19.
It is not clear to me that you "collocated" SST DMI (line 19 and figure 4 caption).
Answer:
Corrected to co-located in line 19 and figure 4 caption.
We meant a stage in data processing when the closest pixel of SST (or SSS) image is found for every in situ measurement of chosen coordinates and date.

Line 25. "embarking" -> "carrying"?
Answer: Corrected

Line 32 to page 11 top line. This sentence is hard to read. You might move all the definitions / explanations to the end.
Answer: Corrected to following:
*"To create SST DMI L4 product, only the observations between 21:00 and 7:00 local time are used (\citet{hoyer2014bias}), thus, local diurnal variations of SST are supposed to be filtered out. Diurnal variation of temperature might be present in real in situ measurements in case of strong diurnal warming events, but no particular observations allowing to investigate this question were done during the cruise."*

Page 11.
Line 4. "at attempting to retrieve" -> "at retrieving". ["aims . . at attempting" is duplication.]
Answer: Corrected

Line 16. "weighted by the estimated error" suggests more weight is given to values with larger estimated error. Maybe "weighted by comparison with the estimated error" or "weighted relative to the estimated error"?
Answer: Corrected to "weighted relative to the estimated error"

Lines 21, 22. I missed definition of "ACARD" that would give meaning to a value 45. Perhaps a reference would be best.
Answer: "Waldteufel, P., Vergely, J. L., Cot, C., "A modified cardioid model for processing multiangular radiometric observations," IEEE Transactions on Geoscience and Remote Sensing, vol. 42, pp. 1059-1063, 2004."

Line 32 and figure 5 caption. I am not sure about "collocated SSS" (c.f. Page 8 comment above).
Answer: Corrected to co-located

Lines 4-5. "a higher confidence" -> "confidence"? ["higher" implies comparison – with what?]
Answer: Corrected

Line 15. "polynya" (spelling)
Answer: Corrected

Line 16. "south of 79∘N" (assuming you mean that it was ice-free at 78∘N, for example).
Answer: Corrected

Page 14.
Equations (1). For Ekman transports I think you should not divide by Ekman depth. The division gives a velocity. Also I think your w_ekm has the wrong dimensions.

Answer:
Thank you for correction. Indeed, we had a velocity instead of horisontal transport in the Equation 1 (corrected the formulae to a version without division by Ekman depth, thus using [m^2/s]). As for the vertical Ekman velocity, which is given in [cm/s], the derivation of this equation is well demonstrated in http://www.geo.cornell.edu/ocean/p_ocean/ppt_notes/16_EkmanTransport.pdf, page 6.

Line 28. I think "freshwater" -> "water" twice (you give the salinity values anyway and it is not actually "fresh"; maybe "fresher"?).
Answer :

Corrected to "Water with salinity … . Additional fresher water from the Kara Sea…"

Page 15.
Figure 6b black line needs explanation.
Answer : Added "The dotted lines in Figure (b) show the position of sea ice edge at different moments of time before and during the ARKTIKA-2018 cruise"

Line 7. "sensibility" -> "sensitivity"?
Answer: Corrected

Page 17 line 3. Omit "back" or replace with a word giving extra meaning.
Answer: Corrected

Table 1 caption. "ans" -> "and"; "explication" -> "explanation"
Answer: Corrected

Line 18. "in a previous section" is too vague; give the section number.
Answer: Corrected

Page 19 Figure 8. I do not see the red freezing line mentioned in the caption.
Answer : At the bottom of every figure in this panel a thin red line shows the freezing temperature for different salinity.

Figure 10 caption line 2. Better "Small coloured circles".
Answer: Corrected

Line 3. Omit ","
Answer: Corrected

Figure 12. I rather agree with referee 1 that these Hovmoeller plots would be better with time as the "y" axis and (increasing) latitude as the "x" axis.
Answer:
Thank you for your suggestion. We discussed this question with co-authors and agreed to keep the same form as presented. There is "geographical" meaning in the presentation with x-axes for time and y-axis for latitude, and we would prefer to show it this way.

Lines 2-4. I think referee 2 had questions about wind direction and mixing. Your point about an error in DMI SST avoids the need to answer those questions but it is not clear why "an error in DMI SST product" should be "due to . . strong winds"

Answer : The error in SST-retrieval are related to the cloud masking and a special correction for the "skin temperature" to bring it to subskin values. This correction is usually uniform for the whole basin and is calculated for the low-wind speed conditions (e.g. 0.17 Kelvin for the Arctic Ocean, see Hoyer et al. 2014, p.203)

Hoyer, J. L., Le Borgne, P., and Eastwood, S. (2014). A bias correction method for arctic satellite sea surface temperature observations. Remote Sensing of Envi- ronment, 146:201–213.

Line 3. Omit "on".
Answer: Corrected

Line 7. "depression" -> "cyclone"?
Answer: Corrected

Line 1. Please explain what "frontal area" you refer to.
Answer:
We mean the surface thermal and salinity fronts, marked as the sharpest gradients.

Lines 4-5. "Higher values . . are expected . . as this is an area exposed to rapid changes." Please explain.
Answer:
As the timeseries of SST and SSS fields is short, the easiest way to see any variability is to observe the area with dramatic changes of SST or SSS (frontal area). This rapid and explicit variability of oceanic characteristics (SST and SSS) could be related to the rapid (usually more rapid by its nature) atmospheric variability in the wind speed or sea level pressure.
We excluded this sentence to avoid confusion.

Lines 13-14. "presented below oceanographic sections" -> "oceanographic sections presented below"
Answer: Corrected

Lines 19-20. Move "(20-30m)" to after "similar" in line 19.
Answer: Corrected

Line 21. "which is clearly seen in temperature signal" -> "as is clearly seen in the temperature"
Answer: Corrected

Line 25. "deepens" -> "moves deeper"
Answer: Corrected

Page 26 line 10. "first (surface)" -> "surface"?
Answer: Corrected

Page 27.
Line 4. "sea edge very slowly" looks like a mistake. Or it needs explaining!
Answer:
Corrected to "It is interesting that the areas with the highest sea ice melt fraction (Fig.\ref{fig: dO18}, c) (5-10%) very slowly follow the sea edge, so they were observed in the central and western part of the Laptev sea and in the MIZ area in the East-Siberian Sea."

Line 12. "In a previous study on" -> "A previous study in"
Answer: Corrected

Lines 3-4. "The cross-validation of satellite DMI SST and SMOS SSS "A" estimates is done with rare in this region continuous TSG measurements and CTD data. " I think you want "Satellite DMI SST and SMOS SSS "A" estimates are cross-validated with continuous TSG measurements and

CTD data (rarely done in this region)."
Answer: Corrected

Line 8. ". . and to discuss . ."
Answer: Corrected

Line 12. I don't understand "and diminishes"
Answer: Deleted "diminishes"

Line 14. "which" -> "as"
Answer: Corrected

Line 3. "is to start in" -> "starts at"?
Answer: Corrected

Line 4. "will be finished by November"? The surface may be completely covered by ice but surely the ice will continue forming under the surface through the winter?
Answer: Yes, definitely. We meant that the sea surface will be covered by ice.

Lines 18-19. "Ekman depth is controlled only by the Coriolis parameter $f$"? Also by viscosity!
Answer:
Indeed, thank you for reminding. Added to the text.

Line 30. "precised" -> "stated"? "Rossby" (spelling)
Answer: Corrected

Line 31. "altimeter along-track resolution of 300 m" seems very fine. Some km "footprint"?
Answer:
Corrected to the "altimeter along-track sampling of 300 m", as it is mentioned in Armitage et al., 2017, p.1773. Indeed, 300 m is the distance between two succeeding measurements, while the "footprint" of, e.g. Sentinel-3 altimeter in Low Resolution Mode is 1.64 km.

Lines 31-32. "at the shallowest areas the highest eddy kinetic energy was reported in the same study." -> "the same study reported largest eddy kinetic energy in the shallowest areas"?
Answer: Corrected

Page 31 line 4. "thank for" – thank who?
Answer: Thank you for your remark, it was a mistake, the "isotopes analysis" acknowledgements are given below.

[revised manuscript text omitted]

---

## Author Response (AR3)

Topic Editor Decision: Reconsider after major revisions (21 Sep 2020) by John M. Huthnance
Comments to the Author:

Dear Authors
Thank-you again for your revised manuscript. I am sorry for the delay in obtaining reviews and getting them on the Editorial system. Both reviews raise serious comments of presentation and there are a few serious aspects of science also for you to address – statistics of results, Ekman transports. I am copying these comments below for your convenience (plus a couple of mine at the end). These comments will be public if or when your manuscript is published, so readers will be able to see whether you have answered them satisfactorily. I may want to send it back to the reviewers again. Please also be sure that your text is clear (Reviewer 4 makes many points). A final manuscript will be copy-edited, but the copy-editor cannot be expected to guess what you mean.
Yours sincerely
John Huthnance (editor)

**Reviewer 1**
Second review of Tarasenko et al., Surface waters properties in the Laptev and the East-Siberian Seas in summer 2018 from in situ and satellite data, submitted to Ocean Science.

In this second review, I refer back to my first review, and to the authors' responses to that review.

In the Overview, I noted that the Introduction section and the Discussion and Conclusions section needed to be improved.

The Introduction is now much better, but there is still an important missing point. The Arctic stores, imports and exports huge volumes of freshwater (as the authors note), but when concentrating on the shallow Siberian shelf seas, they make no comment about their role as conduits for freshwater. Their importance is more that the large river runoff volumes must cross the shelf seas before they reach the Beaufort Gyre / Canadian Basin (and find their eventual fate through export out of the Arctic). How do these freshwaters cross the shelf seas to the deep basins? Our understanding of processes / dynamics / timescales is incomplete, and this is where Armitage (2016) is helpful - it is not just about the small downwards storage trend in the shelf seas, it is also about comparisons of storage variability between deep and shallow basis.

The authors should also consider Schlosser et al. (DSR I, 1994) "Arctic river runoff: mean residence times on the shelf and in the halocline". Has this excellent work been updated? In any event, they show from geochemical measurements that river waters spend two or three years in the shelf seas before leaving offshore. How does this fundamental observation reflect on the authors' findings?

*Answer:*
*Thank you for your suggestions. We modified the Introduction adding the citation of Schloesser et al. (1994) and a more recent study of geotracers in the Arctic (2020), published after we sent our paper to OS in May 2019.*
 *As for the storage variability between deep and shallow basins, I would like to leave this subject for a different study, as it does not concern the small-scale variability presented in this paper. A theory that the Beaufort gyre accumulates most of the fresh water entering the Arctic over the shelf needs to be confirmed for the period after 2014 (so after the study of Armitage 2016), as the sea ice regime has changed significantly over last 6 years.*

*Nevertheless, the reviewer's comment made us think that we should have put more emphasis in the introduction on the interest of satellite sea surface salinity for monitoring the fate of the freshwater on the shelf, so we have added:*

The salinity provides precious information about the fate of the freshwater river input. In this paper, we look at the information accessible with satellite salinity. While this information is restricted to the top sea surface, the regular and synoptic monitoring of sea surface salinity from space allows to document its spatio-temporal variability in great details not accessible with any other means, providing a new tool for analyzing some of the processes at play.

 Regarding the Discussion and Conclusions - I am mainly happy with the revised text but see my point 10 below for a specific (continuing) problem.

I next consider the authors' replies to my specific points.
1, 2, 3, 4, 5. OK, thanks.

6. You state in section 2.2.3 (line 19 in v5) that the SMOS spatial resolution is about 50 km, so that oversampling at 15 km means that each 50 x 50 km2 resolved area contains about ten 15 x 15 km2 points, and oversampling means that those ten points will contain noise around the "true" mean of the 50 km grid cell. So I repeat my question: how does this affect your statistics?
*Answer:*
*This is not what we meant. SMOS SSS are representative of SSS integrated over about 50x50km2 given the footprint of SMOS radiometric measurements involved in the SSS retrievals. The SMOS radiometric measurements are integrated over approximately a 50km x 50km area. We don't make any spatial average. The oversampling on a 15km grid is possible owing to the image reconstruction of the SMOS interferometric data, but in our processing we don't make any spatial average.*

7. OK, thanks.

8. OK, I am happy with your explanation but you need to note in the relevant figure captions that the virtual transects are shown by the red lines.
*Answer:*
*Corrected, thank you*

9. OK, thanks.

10. No problems with Ekman transport, but the Ekman pumping calculation in shallow water is simply not valid. Just because you can solve an equation does not mean that use of the equation is valid. If this is at all unfamiliar, I recommend the nice online textbook by Robert Stewart of Texas A&M (search for "Stewart oceanography text book", it is available for free download) and look at the Ekman sections (in chapters 9, 12). The balance of operative processes is very different in shallow (tens or a few hundreds of metres) versus deep (thousands of metres) waters, which is why I suggested looking at the Lentz papers (JFM 2002, JPO 2004, and Moffat & Lentz, JPO 2012). I also recommend Whitney & Garvine (JGR 2005) as a nice illustration / application of the framework. They describe upwelling and downwelling response to winds and Ekman transports in shallow waters (including buoyant coastal currents). This would provide a credible framework to describe your observations, and not Ekman pumping. You do not necessarily need to do more analytical work, that is not what I am asking for. But you need to remove the Ekman pumping results from your figures, and then interpret the Ekman transports in shallow-water terms.

For example, in your v5 figure 15 panel c, showing an east wind with offshore Ekman transport, that will induce downwelling at the coast and a consequent residual vertical circulation that draws water from offshore onto the shelf below the surface layer. Same figure panel f shows a south wind that induces a mainly along-shore Ekman transport that probably does not induce a residual vertical circulation. This style of interpretation would improve the Discussion and Conclusions.

*Answer:*
*Corrected, please see a new version of "Discussion and Conclusion"*

11. This is an additional point. The manuscript should become publishable subject to the remaining problems being fixed, but the presentation of text in figures (mainly axis and contour labels) is poor. Text is often much too small for sensible legibility. Please check each figure, consider what size the text might appear on the printed page, and make it all a readable size.
*Answer: Thank you, we tried to correct it*

**Reviewer 4**

Summary
In this paper, the authors use a combination of satellite and in-situ data to document the evolution of water masses in the Laptev and East Siberian Sea of the summer of 2018. The focus is on the surface layer, and in particular, water masses deriving from the Lena River and low salinity waters from the Kara Sea.

The authors make use of CTD transects undertaken in the ARKTIKA-2018 expedition to evaluate the accuracy of satellite SST and SSS products. For the latter, the authors themselves present a novel SSS product that they deem to be most appropriate for the geographical region under question. The authors argue that the match between the satellite products and the in situ data is sufficiently strong that the satellite products themselves can be used for much of the remaining analysis in the paper, in particular, defining separate water masses and tracing their evolution over the late summer 2018 period.

Using a TS framework to separate the water masses, the authors resolve spatial changes in the water masses over time. They also focus attention onto particular sections in order to make links between water mass changes and other processes: wind speed and the passing of cyclones. The authors present geochemical analysis from the cruise transects that helps to verify the origins of the surface water masses between the tree end members: river water, sea ice melt, and marine water.

This paper is a useful documentation of the surface ocean in the Laptev and East Siberian Sea in the summer of 2018, and to that extent that the authors successfully make process-based findings, it has implications beyond the time frame during which the measurements were made. The methods are mostly appropriate for the chosen questions, though I have some specific suggestions in terms of robustness of the evaluation of satellite data, and the means of comparison of certain physical data.

Novel oceanographic data and novel SSS satellite retrievals are presented, but it is not quite true that the presentation of water mass properties at a synoptic scale in this region of the Arctic is totally novel (see Osadchiev et al., 2020, Scientific Reports).

*The paper of Osadchiev et al. 2020 was submitted in March 2020 so later than the initial submission of our paper that was available for public review and discussion on 31 July 2019 on Ocean Science Discussion.*

The significance of the work is largely in its implications for the usefulness of satellite SST and SSS in the Arctic, though in my view the authors have more work to do on that point. The communication of the results and methods needs some improvement, both in text and figures.

*Main points*
Evaluation of satellite data.

The authors present co-located surface water (CTD and TSG) measurements and satellite data in scatter plots. The authors also provide a useful error frequency distribution, from which they show that the observed errors are more substantial than that provided by the satellite product (for both SST and SSS). After discussing the comparison to in situ data, the authors simply resolve that the two products agree well with the in situ data and use this as support for using the products for the following analyses. A more robust justification is required, especially given that the differences to the in situ data are not insignificant.

Can the authors use existing conventions in the literature for what constitutes a "good agreement"?
Another approach would be to think about how sensitive the key findings are to random error and biases in the satellite data. Can the authors comment on what threshold of random error or bias the data must be below in order for the results to be robust?

If it can be demonstrated that the results are not sensitive to the present level of error in the measurements, that would be more convincing evidence to the reader that the data are appropriate to be used.

*Answer:*
*We think that part of the reviewer's remark comes from an abuse of language. What was called 'error on DMI estimated SST', 'theoretical error' on SMOS SSS (following the terminology employed in the SMOS measurements documentation) or 'error on SMOS ESA L2 SSS' is in fact an uncertainty; for instance in the case of SMOS, it corresponds to the expected standard deviation between the true SSS value and the SMOS SSS as derived by the SMOS retrieval scheme based on uncertainties on the SMOS brightness temperatures and ancillary measurements used in the retrieval. Hence what is shown on Figure 4d, or 5d is that the statistical distribution of the differences between satellite-in situ SSS is in rather good agreement with the uncertainties of the satellite measurements. However, it is not possible to thoroughly validate these uncertainties in this paper as it would require much more measurements: actually, as shown on Figure 4b and 5b the uncertainties vary much over the area so that a validation of the uncertainty would require large number of measurements for each class of uncertainty value. We have made the vocabulary clearer and we now explain how Fig 4d and Fig 5d show that the order of magnitude of the observed errors is in agreement with the product uncertainties and that these errors are much smaller than the observed spatial gradients shown on Fig 4a and 5a.*

Surface water-mass TS analysis

It is inevitable that the separation of water masses into distinct groups will be based on somewhat arbitrary salinity and temperature values. However, the authors need to do a bit more to explain to the reader why they have chose the definitions they have, and how sensitive their results are to these choices.

It is somewhat concerning that the range of values displayed by the satellite measurements vastly exceeds that of the in situ data. The authors need to explain to what extent this is due to

the in situ data being fewer and spatially limited, and to what extent it is due to error in the satellite data. It is a concern that the classification might be based on erroneous values; it appears that according to the in situ measurements, water-masses 1 and 3 were not sampled. Comparison of the cruise track and Figure 9 indicates that the water mass 1 might never have been properly sampled by the CTD measurements, while it is a surprise that watermass 3 is never sampled in the MIZ on sections 7 and 8.

Can the authors put the cruise sections in context of the identified water masses?

*Answer:*
*The range of variation in SSS and SST values covered by the satellite measurements (first row of Fig 8) is an order of magnitude larger than the one covered by in situ measurements (first column, bottom row of Fig 8). The difference in TS diagram covered by each type of measurement cannot be explained by the errors on satellite measurements (rms difference with respect to in situ measurements of 0.77°C and 0.8 in temperature and salinity respectively) nor by the uncertainties associated with each satellite product ( Fig 4b and 5b). It primarily reflects the much better spatio-temporal monitoring of the different water masses by satellite measurements. This is now explained in the paper. We've chosen arbitrary limits and the number of water masses based on the distribution of T-S density points for the September 4, as it is shown on the figure 8. The T and S ranges of variation for each class has also been chosen to be well above the T and S uncertainties.*
*When we submitted the paper, we did not use any particular clustering method to separate the water masses, only the visual analysis of the potential centers of the water masses. Specially for this answer, I've just tested the dataset of SST&SSS for Sept 4 with the "Elbow method" used for the K-clustering to define the number of clusters centers (please, see details of the method here https://towardsdatascience.com/machine-learning-algorithms-part-9-k-means-example-in-python-f2ad05ed5203):*

[Figure]

*I propose that 6 cluster centers are acceptable for a classification, which main purpose was only to follow the transformation of the water masses. Any further analysis of the sensitivity will induce a number of rather heavy tests in terms of computational cost. A lower number of water masses will bring a less detailed picture of the transformation of the riverine water in the sea water, the sea ice melting and the beginning of freezing.*
*As for water masses 1 and 3 (both less saline than 25) absent in the CTD measurements, it is due to those simple facts, that:*
*1) the Arktika-2018 expedition didn't go far enough to the southern Laptev sea, close to the origin of the riverine water;*

*2) MIZ measurements show the presence of water mass 4, which is close to the WM3, but the WM4 is saltier. Indeed, there might have been 5 water masses, meaning one WM 3&4 together. We separated them mostly to be able to distinguish the water coming from the Kara Sea (seen on Aug 15 and Sept 25 in the west) and the freshwater to the east (in the East-Siberian Sea) exposed after the sea ice melting.*

[Figure]

*Minor points*

P1L20 - make it clear exactly what region was affected by this freshwater decrease
*Answer: Corrected to the Eurasian shelf*

P2L14 - Statement should be revised in light of Osadchiev et al. 2020
*Answer: The paper of Osadchiev was influenced by this paper sent for a review to OS in May 2019, one year before Mr Osadchiev decided that he also has some relevant data to publish on this subject. His comments on this work can be found in the Ocean Science Discussion in 2019. Considering this, this paper does not require to be referred to here, as apparently our "synoptic description of the Laptev Sea surface waters" was done before his publication.*

Fig. 1 - Mark rivers clearly. Dotted lines are used for ice edge, bathymetry and grid markings; this is a bit hard on the eyes. Consider changing some (or all) of these to light, solid lines. Consider choosing a different colour for text that refers to different types of features (eg. rivers in blue, straits in red…)
*Answer: the style of representation was slightly changed. The colour of the figure corresponds only to the time of both the expedition time measurements and the sea ice edge moment. Using different colours for geographical features would be misleading.*

P5L7 - what exactly is the standard error in this context?

$$\sigma_{\bar{x}} \approx \frac{s}{\sqrt{n}}$$

*σ is the standard deviation of the time series (temperature or salinity)*
*n is the size (number of observations) of the time series.*

P6L1 - units can be given for the standard deviations

*Answer: Added « degree Celsius ». Salinity is unitless following the recommendations of earlier reviewers.*

P7L6 - what depth does the satellite sample? This is clearly relevant in the discussion here and would be worth stating explicitly.
*Answer: The section dedicated to the satellite measurements is Section 2.2*
*For the SST blended product, the actual depth measured by the satellite is not very important, as 1) the IR and the MW wavelength are different and penetrate the surface on o(mm): 1 microm for IR and 1 mm for MW; in any case, the effect of skin-layer measured by the SST is supposed to be corrected in the product L4 and the temperature is supposed to be brought to the "foundation temperature" – the temperature of the surface layer not impacted by the diurnal cycle.*
*As for the SSS we talk about 1 cm. The effect of vertical stratification on the SSS validation is described thoroughly in Supply et al.,2020.*

P7L8 - given that the median profile portrays a picture of the water column that is not representative of any given profile, I am not sure it is worth describing it here as if it is physical. For instance, the smoothness of the thermo-/halocline is an artefact of averaging many profiles with sharper halo-/thermoclines at different depths.
*Answer:*
*We agree, but this figure was explicitly requested by one of 5 other reviews during the last 18 months of reviewing process*

P7L14 - use 'is composed of' instead of 'composite' (composite suggests another meaning)
*Answer: Thank you, corrected*

P8L12 - meaning of statement in parentheses is unclear
*Answer: We mean that the IR SST measurements can be obtained only in case of clear sky without clouds. Clouds are opaque for IR waves. Added "without clouds" to the text.*

Fig. 4 - Contour plot looks a bit messy - consider pixel plot without contours. Fig. 4c does clearly show coherent groupings with distinct biases in the satellite data (e.g. measurements from days 45-50 are consistently too cold in satellite data, while in days 25-30 the satellite data are too warm)
*Answer: Agreed. The blue points standing out for the days 45-50 correspond to the measurements in MIZ. Although DMI do their best to develop a special algorithm for the MIZ, it is still not perfect. This commentary is provided in P10L28-30*

P9L6 - is this all data or just those from CTD casts where the mixed layer depth is below 7 m?
*Answer: These are all measurements from all CTD casts.*

P10L21 - 'potential cloudiness' is a bit confusing for a specific date that has already passed. Was it partially cloudy?
*Answer: Yes, corrected*

P10L33 - justification required, as suggested above
*Answer: A "good agreement" is justified by a high correlation coefficient and independency of the error over the ice-free areas (excluding the MIZ).*

P11L7 - provide a reference for this statement? It is not obvious to the reader that this would

be the case, as density of measurements and degraded sensitivity are not obviously compensating.

*Answer: The SSS sensitivity decreases when the SST decreases, but the number of measurements increases when the latitude increases. Repeated measurements over the same area reduce the error when the seven-days running means are calculated (see Supply et al, 2020).*

P11L20 - explicitly state the implications of oversampling for interpretation of the results

*Answer: Each SMOS retrieved salinity is integrated over a 50km x 50km area. We don't make any spatial average. The oversampling on a 15km grid is possible owing to the image reconstruction of the SMOS interferometric data, but in our processing we don't make any spatial average.*

P11L26 - is this an innovation or a convention? Provide a reference if a convention, provide justification if an innovation

*Answer: It is a convention proposed to ignore the SSS with higher uncertainty values due to a too low or too high wind speed (see Supply et al, 2020 for details).*

P13 L2 - DMI SST / SST DMI - be consistent with the naming

*Answer: Thank you, corrected everywhere to DMI SST*

P13L6 - quantitative assessment of what makes this a very good agreement would be valuable (as suggested above)

*Answer: Comparison between the in situ practical salinity and SMOS SSS "A" shows a very good agreement, not yet demonstrated before by any other salinity product in the Laptev Sea: the correlation coefficient is 0.86 with a RMS = 0.86.*

P13L24 - what exactly are 'the ice charts' from AARI? Are they available online?

*Answer:*
*Ice charts show the sea ice concentration and type and are created from satellite data by an ice expert. Regional ice charts prepared in AARI can be accessed from http://www.aari.ru/odata/_d0015.php?lang=0&mod=1&yy=2018 or http://www.aari.ru/odata/_d0004.php?m=Lap&lang=0&mod=0&yy=2018*

P15L9 - can thermal fronts be seen in the daily satellite data?

*Answer: yes, as it is seen in Fig.4*

P16L1 - provide uncertainty

*Answer: The uncertainty for the Laptev sea is 0.0996 and for the East-Siberian is 0.1981*

P16L11 - how might one distinguish between these two sources of variability? Can any attempt to do so be made here? Can a comparison be made with the variability detected in the observational survey?

*Answer: It is rather difficult to estimate the SSS variability with point-CTD measurements. The only "point" close to MIZ, where measurements were done twice is a cross-section of CTD sections 6 and 8 (stations 79 and 108). Both stations are close to 154°E.*

*Station 79 was done on September 8, 2018; station 108 was done on September 17, 2018.*

*In 9 days, there is a difference in surface salinity of 0.5.*
*The edge of the MIZ at Station 79 coordinates (77.5N, 154.18E) was detectable with MODIS images on September 2 (See figure below) and Station 108 was done on the edge of MIZ later as well.*
*The main problem is that these data obtained in MIZ is certainly filtered out from the SMOS SSS measurements, as sea ice pixels elevated brightness temperature contaminates the salinity signal for retrieval.*

[Figure]

September 2, 2018 MODIS visual band: position of future Station 79 is marked with a magenta circle. The MIZ edge is seen below the clouds.

[Figure]

September 15, 2018 MODIS visual band: position of future Station 108 is marked with a magenta circle. The MIZ edge is seen below the clouds, the station is still in MIZ area.

P16L23 - explain origin of the two separate branches
*Answer: The two surface branches are mostly the warmer and low-saline surface waters of the ice-free Laptev Sea and the colder and low-saline waters of the ice-covered East-Siberian Sea.*

*The latter correspond to the measurements from the section 7 and 8 eastward 150E (see the Figure below).*

[Figure]

P16L29 - make it clear to the reader that these diagrams are not analogous to Fig. 7, as they are only for surface values
*Answer: Corrected, thank you*

P16L1 - describe how water mass boundaries were determined (as suggested above)
*Answer: Corrected in the text.*

P18L9 - worth explicitly stating that this water mass is not considered to be melt, but trapped river water, based on the geochemical analysis in 3.3.4
*Answer: Thank you for the suggestion, corrected*

P18L4 - clarification required to make origin of CMS as referred to in L4 and L14 compatible. Could be simply saying that
   - it is comprised of both transformed CF and transformed WF (if this is the case),
 or explicitly saying that
   - there is no transformation route to CMS directly from WF, but it is produced only from CF.

*Answer: Neither of these hypotheses apply.*
   - *CMS can be produced from WF by passing the stage WMS (if it is the "Lena river water" between the Laptev and the East-Siberian Sea.*
   - *CMS can be a transformed river water from the Kara sea. And there we don't know was it first WMS or CF.*

Fig. 9 - use acronyms for the water masses as per main text
*Answer: Corrected*

P23L6 - it is not surprising that there is a weak correlation between these variables; salinity changes will effectively integrate the variability in the wind (see, for instance Osadchiev et al. 2020, Fig. 4). Could the authors try correlating the time derivative of salinity and temperature with the wind speed? This might be more instructive.

P24L5 - is it that the "warm" river water signal is not observed anymore?
*Answer: Yes, this is what we meant. Corrected in the text.*

Fig. 13 - show dashed blue line over entire sections for those done entirely in the MIZ
*Answer: Corrected*

P26L13 - comment on the implications of the findings regarding the B-V frequency?

*Answer:*
*Corrected to:*
*« The maximum value of Brünt-Väisälä frequency are less than for other sections and are observed at 20 m depth and at 55 m depth, following $1024 kg/m_3$ and $1026.5 kg/m_3$ isopycnals, accordingly, showing the maximum stability of water vertical stratification under the ice »*

P27L8 - comment on timing, which is naturally important here
*Answer: Corrected to*
*« The sea ice formation (the negative values of sea ice melt fraction) is found in MIZ and its vicinity at 78-70◦N - 150-160◦E of the East-Siberian Sea, which is expected, as this measurements were done in the second part of September 2018, the beginning of freezing season. The presence of river waters may accelerate the sea ice formation if the air temperature favours it. »*

P29L13 - sentence is ambiguous
*Answer: Corrected*

All figures showing maps should include the traces of the major rivers, and in my view it would be helpful to keep the names of the straits on at least one window in each figure.

**Editorial**
The written text needs some improvement in grammar. A few pointers: Use of 'the' and 'a' needs attention

Use of passive voice and tenses is sometimes confusing/ambiguous. I'd recommend using second person ('we'), active voice, present tense to described things done by you, the authors, in this study.

**Editor comments**

Following up on Reviewer 4 "the" and "a": "the" is the "definite article", "a" is the "indefinite article". That means that "the" is used when referring to something that has already been "defined", either because already mentioned in the previous text or by an adjective, e.g. "the black cat" or "a cat" if you did not previously refer to cats.

Reviewer 1 commented on Ekman transport in shelf seas. There are two points.
   a. Surface Ekman transports are relative to the flow beneath the surface layer. If the wind stress implies upwelling near a coast, for example, the offshore Ekman transport implies onshore flow below and the absolute offshore transport is reduced.

   b. As hinted in a, the nearby coast can induce vertical velocity with uniform wind stress.

[revised manuscript text omitted]